# ABA-independent PP2C-binding in PYLs traces to bacterial origins and persists in land plants

Tianjiao Lu [1,2,5], Qingzhong Li[1,5], Tao Hu[1,3], Wenqi Li[4], Yafei Lu[4], Huiling Huang[1,2] & Yang Zhao [1,2] ✉

Land plants have evolved strategies to survive water deficiency. Among these adaptations, the "drying without dying" strategy evolved in early land plants and is maintained in the desiccated seeds of angiosperms. This process is regulated by a family of ABA receptors known as the PYR/PYL/RCAR (PYL) family, which can bind to clade A protein phosphatases 2Cs (PP2Cs) and suppress their inhibition of water stress responses. ABA-independent PYLs first emerged in an aquatic algal lineage; however, their evolutionary origins and the mechanistic basis of ABA-independent PYL variants in land plants remain poorly characterized. Here, we characterize ABA-independent PYL-like proteins from bacteria, algae, and land plants that retain constitutive PP2C binding but lack ABA-enhanced inhibitory activity, supporting their putative bacterial origin via horizontal gene transfer. We identified a bacterial PYL homolog (PrPYL) in *Paraburkholderia rhynchosiae* with PP2C-binding ability, three PP2C-inhibiting PYL homologs in Zygnematales algae, and ABA-independent PYL-like proteins in seed plants (e.g., AtPYL13 and AtPYL13-like proteins). *AtPYL13*-like genes in dicots exhibited high expression during seed maturation and in desiccated seeds, suggesting a functional shift from canonical ABA receptors to ABA-independent PYL-like proteins following gene family expansion. Two invariant residues underlie ABA dependence in canonical PYL receptors. Transcription factor ABI3 mediates *AtPYL13* expression during the mature seed stage, thereby locally restricting constitutively activated stress signaling. Our findings indicate that ABA-independent *PYL*-like genes likely originated via horizontal transfer from bacteria and function in basal stress signaling in seed plants.

Terrestrialization represents a pivotal event in green plant evolution, requiring adaptive mechanisms to overcome multiple environmental challenges, particularly fluctuating water availability. Plants employ two principal strategies to cope with water deficiency: "drying without dying (desiccation tolerance)" or "avoiding drying (drought avoidance)"[1]. Early land plants are mainly poikilohydric and adopt the "drying without dying" strategy to survive extreme conditions[2]. Vascular land plants retain the "drying without dying" strategy in specific

[1]Key Laboratory of Plant Carbon Capture, Shanghai Center for Plant Stress Biology, CAS Center of Excellence in Molecular Plant Sciences, Chinese Academy of Sciences, Shanghai, China. [2]University of Chinese Academy of Sciences, Beijing, China. [3]State Key Laboratory of Herbage Improvement and Grassland Agro-ecosystems, College of Pastoral Agriculture Science and Technology, Lanzhou University, Lanzhou, China. [4]State Key Laboratory of Bio-membrane and Membrane Biotechnology, Center for Structural Biology, School of Medicine and School of Life Sciences, Tsinghua-Peking Center for Life Sciences, Tsinghua University, Beijing, China. [5]These authors contributed equally: Tianjiao Lu, Qingzhong Li. ✉e-mail: yangzhao@psc.ac.cn

organs, such as seeds and pollen, but mainly utilize the "avoiding drying" strategy for their vegetative stages. These processes both depend on the activation of stress responses by the phytohormone ABA[3,4].

The activation of ABA signaling occurs through a protein phosphorylation cascade, in which SNF1-related protein kinase 2 (SnRK2) kinases have a central role. Upon ABA binding, the PYR/PYL/RCAR (PYL) ABA receptors interact with and inhibit clade A protein phosphatase 2Cs (PP2Cs), releasing SnRK2s from PP2C inhibition and activating downstream stress responses[5,6]. The ABA-binding properties of PYLs vary significantly in seed plants; likewise, their PP2C inhibition mechanisms show distinct ABA-dependence patterns: (i) strictly ABA-dependent (no basal inhibition activity without ABA, e.g., the dimeric subfamily III PYLs in angiosperms)[7]; (ii) ABA-enhanced (basal inhibition activity potentiated by ABA)[8]; or (iii) ABA-independent (not modulated by ABA). The "ABA-independent PYL-like proteins" include: 1) constitutively active PYLs that activate stress responses regardless of ABA[9]; and 2) inactive PYLs incapable of activating stress responses. Overall, ABA signaling plays a central role in the "avoiding drying" strategy via stress-induced ABA accumulation[4,8,10,11] and also regulates seed development and germination via tightly regulated dynamic ABA levels[4,12]. Although it is predictable that ABA-independent PYL-like proteins constitutively activate stress signaling, it remains unclear which plants encode such proteins and whether they contribute to the "drying without dying" strategy in vascular plants.

PYLs existed in the last common ancestor of Zygnematophyceae (green algae) and Embryophyta (land plants)[13,14]. The PYL-homolog ZcPYL8 from the alga *Zygnema circumcarinatum* shows ABA-independent PP2C inhibition, enabling constitutive activation of stress signaling and desiccation tolerance[9]. In the nonvascular liverwort *Marchantia*, MpPYL1 shows ABA-enhanced activity to inhibit PP2Cs, although with high basal activity[9], which triggers high basal stress signaling and actively represses plant growth[15]. Strictly ABA-dependent PYLs with minimal basal activity emerged in Bryophytina mosses and angiosperms[8,16], potentially enabling a broader range of ABA responses and improved balance between growth and stress responses. However, it remains unclear how ABA-independent PYL-like proteins emerged and were subsequently incorporated into land plant lineages.

One hypothesis is that the ancestral *PYLs* were transferred horizontally from soil bacteria to charophytes during their evolution[14]. Here, we analyzed the activity and ABA-dependence of PYLs from bacteria, charophytes, bryophytes, and angiosperms, identifying ABA-independent PYL-like proteins across these groups. The bacterial PYL-homolog from *Paraburkholderia rhynchosiae* possesses PP2C-binding ability, and PYLs from Zygnematales exhibit PP2C inhibitory ability, supporting that *PYLs* in plants might have originated horizontally from soil bacteria. The ABA-independent PYL-like proteins exist in multiple genera in seed plants, including PYL13 and PYL13-like proteins in dicots and monocots. Most dicot AtPYL13-like proteins are highly expressed in mature and desiccated seeds and confer constant activation of stress signaling. The seed-specific expression of *AtPYL13* is controlled by the transcription factor ABI3. These ABA-independent PYL-like proteins differ from the ABA-receptor PYLs in the highly conserved ABA-binding pocket by two amino acid residues – the invariant lysine and leucine residues that are crucial for ABA receptors. Our discoveries suggest that the ABA-independent *PYL*-like genes might have originated horizontally from bacteria and function in basal stress signaling in seed plants.

## Results

### The bacterial PYL-homolog from *Paraburkholderia rhynchosiae* possesses PP2C-binding ability

Phylogenetic analyses propose that ancestral *PYLs* in plants were acquired through horizontal gene transfer from soil bacteria[14]. The *PYL*

genes from plants form a monophyletic group that nests within the diversity of twenty bacterial genes from two phyla: Proteobacteria and Actinobacteria[14]. We selected four bacterial PYLs showing high similarity to PYLs from green algae, bryophytes, and angiosperms for further analyses (Fig. 1a; Supplementary Fig. 1a). However, these bacterial PYLs (from *Paraburkholderia*, *Pseudomonas*, *Frankia*, and *Sphingomonas*) have apparent differences with plant PYLs in the CL2 region, which is critical for interactions with ABA and PP2Cs[17–26].

To test whether PYL homologs in bacteria function as ABA receptors, we evaluated their ABA dependence in the *Arabidopsis* ABA-insensitive *pyl* duodecuple mutant, in which all *AtPYLs* except *AtPYL6* and *AtPYL13* are mutated[4]. As expected, transfection of the *pyl* duodecuple mutant protoplasts with *SnRK2.6* and *ABF2* did not induce expression of the ABA-responsive reporter *RD29B-LUC*, even in the presence of ABA; however, co-transfection of *SnRK2.6*, *ABF2* with an ABA receptor (*AtPYR1*, *MpPYL1*, or *PpPYL1*) resulted in the induction of *RD29B-LUC* in the presence of ABA (Fig. 1b; Supplementary Fig. 1b). PYL homologs from bacteria, including Proteobacteria (*Sphingomonas*, *Pseudomonas*, and *Paraburkholderia*) and Actinobacteria (*Frankia*), failed to induce *RD29B-LUC* expression with or without ABA treatment (Fig. 1b), indicating these bacterial PYLs are ABA-independent PYLs.

Despite lacking ABA dependence, ancestral PYLs should, first and foremost, interact with PP2Cs. Yeast two-hybrid (Y2H) assays were used to evaluate the PP2C-binding ability of these bacterial PYLs with *Arabidopsis* clade A PP2Cs. The *Paraburkholderia rhynchosiae* PYL (PrPYL), but not the bacterial PYL homologs from *Sphingomonas* and *Frankia*, interacted with several PP2Cs, including ABI1, HAB2, HAI3, and PP2CA, in an ABA-independent manner (Fig. 1c, d; Supplementary Fig. 1c, d). We further confirmed physical interactions between PrPYL and AtABI1 using split luciferase (LUC) complementation (LCI) assays. Transient co-expression of *PrPYL-nLUC* with either *cLUC-AtABI1* or *cLUC-AtHAB2* generated weak reconstituted LUC signals in *Nicotiana benthamiana* (*Nb*) leaves (Supplementary Fig. 1e). The weak PrPYL-PP2C interactions may be outcompeted by endogenous proteins (e.g., plant PYLs and SnRK2s), thus generating very weak reconstituted LUC signals *in planta*. These results suggest that PrPYL represents a prototype ABA receptor.

The PP2C inhibitory activity of PrPYL was evaluated using in vitro SnRK2 kinase assays in the presence and absence of ABA (Fig. 1e, f). Recombinant GST-tagged ABI1 and PP2CA inhibited the kinase activity of MBP-SnRK2.6. As a positive control, recombinant His-PYR1 released SnRK2.6 from PP2C-mediated inhibition in an ABA-dependent manner. In contrast, recombinant PrPYL could not release PP2C-mediated inhibition with or without external ABA (Fig. 1e, f). We further assessed PP2C inhibitory activity using the phosphatase substrate *p*-nitrophenyl phosphate (pNPP), and found that PrPYL did not inhibit PP2CA, ABI1, or HAB1 with or without external ABA (Fig. 1g, h; Supplementary Fig. 1f). These results indicated that bacterial PrPYL binds to PP2Cs but lacks PP2C-inhibitory activity.

The structure of PrPYL was predicted using AlphaFold3 and then aligned to the reported ABA-PYL1-ABI1 complex (PDB: 3KDJ)[19]. The PYL1-ABI1 interaction is mainly mediated by the CL loops, especially CL2 in AtPYL1[19]. Ser112 in the CL2 loop of AtPYL1 plays an essential role in the PYL1-ABI1 interaction, with its hydroxyl group and backbone carbonyl oxygen forming hydrogen bonds with the Glu142 and Gly180 residues of ABI1. The predicted structure of PrPYL resembled that of AtPYL1 (Fig. 1i). However, the CL2 loop of AtPYL1 (gold) was closer to ABI1 (cyan) compared to that in the predicted PrPYL structure (purple). While Ser52 in PrPYL occupies a position structurally equivalent to Ser112 in AtPYL1 within the "S/TGLPA" motif (CL2 loop) and is predicted to form a hydrogen bond with Gly180 in ABI1, the distance of 4.5 Å (instead of 3.0 Å and less) (Fig. 1j) could partially explain the weak interaction between plant PP2Cs and PrPYL. In addition, the ligand-binding pocket of AtPYL1 (gold) accommodates the ABA molecule (green and red sticks) (Fig. 1k), but the ligand-binding pocket of PrPYL

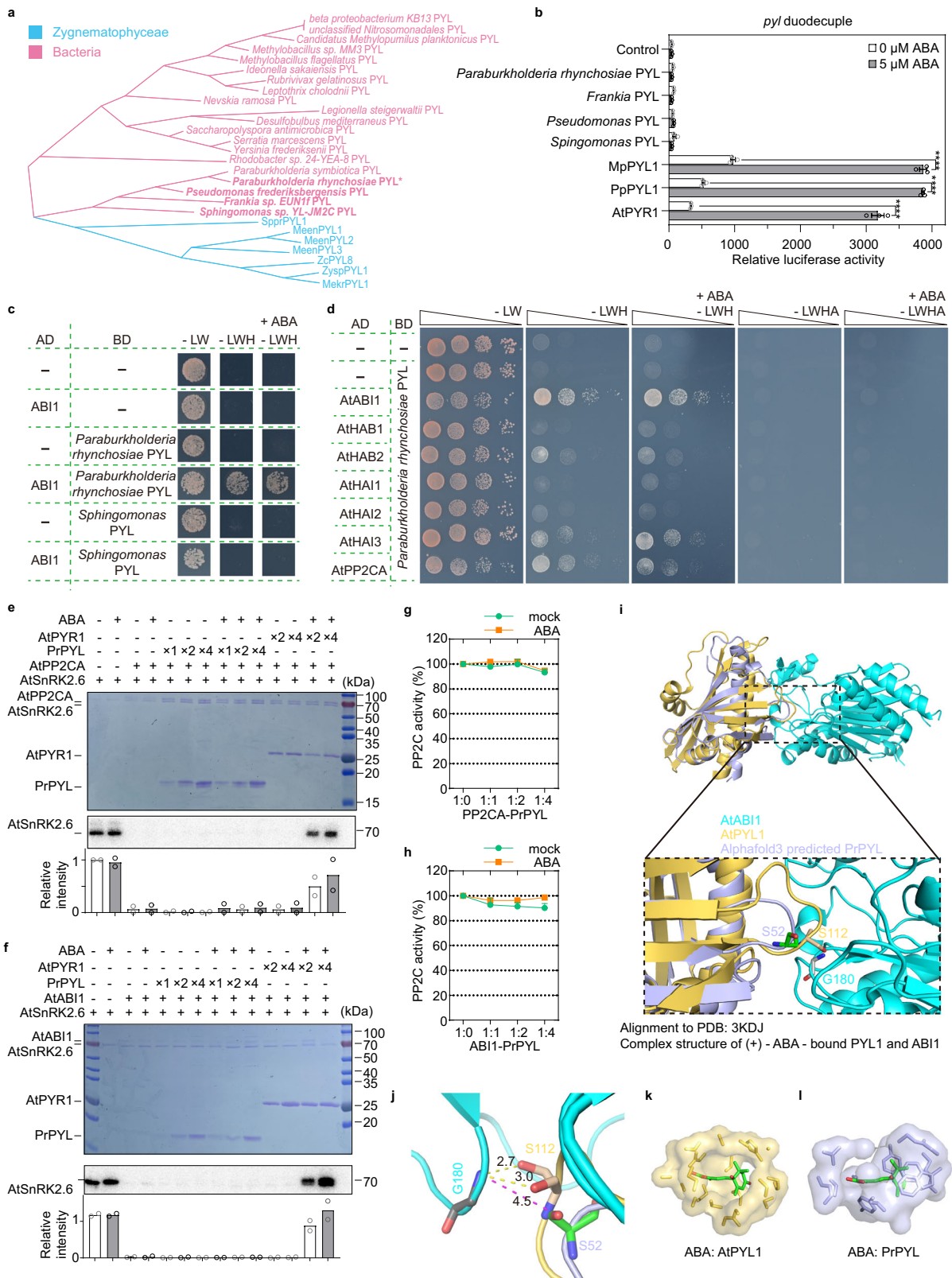

was too small to hold ABA (Fig. 1l), explaining its inability to respond to ABA. We also analyzed the binding affinity between PrPYL and ABA using isothermal titration calorimetry (ITC) and found that PrPYL has no binding affinity for ABA (Supplementary Fig. 1g). Our results support the hypothesis that *PYLs* originated via horizontal gene transfer from bacteria, but acquired their PP2C-inhibitory and ABA-binding functions in plants.

## Zygnematales PYLs exhibit PP2C inhibitory activity in an ABA-independent manner

PYLs in plant lineages appear to have emerged before the last common ancestor of Zygnematophyceae and land plants[13,14]. The algal PYL-homolog ZcPYL8 in *Zygnema circumcarinatum* exhibited an ABA-independent PP2C inhibitory activity in vitro[9]. We mined transcriptome resources from representative land plants and algae and

**Fig. 1 | The bacterial PYL homolog from *Paraburkholderia rhynchosiae* exhibits PP2C-binding ability. a** Phylogenetic analysis of PYL homologs across bacteria and algae. *Sppr, Spirogyra; Meen, Mesotaenium endlicherianum; Mekr, Mesotaenium kramstae; Zc, Zygnema circumcarinatum; Zysp, Zygnemopsis sp.* **b** *RD29Bpro:LUC* expression in *pyl* duodecuple mutant protoplasts transformed with *PYLs*. Transformation with *SnRK2.6, ABF2* and *RD29Bpro:LUC* served as control. Error bars, SEM (*n* = 3 biological repeats). Two-way ANOVA, ****$p$ < 0.0001. Expression data of control and PYR1 were shared by fig. 2b. **c** Interactions between AtABI1 and PYL homologs from *Paraburkholderia* and *Sphingomonas* in yeast two-hybrid (Y2H) assay. Yeast cells were grown on nonselective SD/−Leu−Trp (−LW) and selective SD/−Leu−Trp−His (−LWH) media, without or with 10 μM ABA. **d** Interactions between *Arabidopsis* PP2Cs and *Paraburkholderia rhynchosiae* PYL (PrPYL) in Y2H assay. **e, f** PrPYL did not release SnRK2.6 from inhibition by PP2CA (**e**) or ABI1 (**f**), in the absence or presence of 100 μM ABA, during in vitro kinase assays. SnRK2.6

autophosphorylation was detected by autoradiography (lower panels). **g, h** In PP2C enzyme activity assays using pNPP as the substrate, PrPYL did not inhibit PP2CA (**g**) or ABI1 (**h**), with or without 10 μM ABA. Each reaction contained 0.5 μM PP2C and varying PYL concentrations (0, 0.5, 1.0, or 2.0 μM). Error bars, SD (*n* = 3 biological replicates). **i, j** Overall structure (upper) and a zoomed view (lower) showing interactions between AtABI1 and PYLs (**i**). Structural alignment compares PrPYL (purple, predicted by AlphaFold3) with a reported complex (PDB: 3KDJ) containing ABA, AtPYL1 (gold), and AtABI1 (cyan). The key serine residue in the CL2 loop (S112 in AtPYL1 and S52 in PrPYL) is adjacent to and forms hydrogen bonds with G180 in AtABI1, with hydrogen bond distances as shown in (**j**). **k, l** Close-up view of ABA and surrounding residues within a 5 Å radius. The ligand-binding pocket of AtPYL1 (gold) accommodates ABA (green and red sticks) well (**k**). In contrast, PrPYL (purple) lacks a binding pocket suitable for ABA (**l**). Exact $p$ values for (**b, e, f**) are provided in Source Data.

identified PYL homologs in five species of Zygnematophyceae: *Zygnema circumcarinatum, Zygnemopsis sp., Mesotaenium endlicherianum, Mesotaenium kramstae,* and *Spirogyra sp.,* of which the first four are classified in Zygnematales and the last is classified in Spirogyrales (Fig. 2a and Supplementary Data 1). However, it remains unclear whether PYL homologs from *Mesotaenium, Zygnemopsis* and *Spirogyra* have ABA-dependent PP2C inhibitory activity. To test whether PYL homologs in green algae function as ABA receptors, we evaluated the ABA dependence of PYLs in the ABA-insensitive *pyl* duodecuple mutant. Among PYL homologs from Zygnematales, MeenPYL3 from *Mesotaenium endlicherianum*, MekrPYL1 from *Mesotaenium kramstae*, and ZcPYL8 from *Zygnema circumcarinatum* constitutively activated expression of *RD29B-LUC* in an ABA-independent manner, while ZyspPYL1 from *Zygnemopsis*, SpprPYL1 from *Spirogyra*, and MeenPYL1 and MeenPYL2 failed to induce *RD29B-LUC* expression (Figs. 1a and 2b). A previous study showed that ZcPYL8 inhibits ZcABI1 in an ABA-independent manner and cannot bind ABA[9]. These results indicated that PYL homologs gained ABA-independent PP2C inhibitory activity before the last common ancestor of Zygnematophyceae and land plants.

Consistent with their ability to constitutively activate expression of *RD29B-LUC*, MeenPYL3 and ZcPYL8 strongly interacted with multiple PP2Cs in an ABA-independent manner, as assessed by Y2H assays (Fig. 2c and Supplementary Fig. 2a). Although MeenPYL1 failed to induce *RD29B-LUC* expression (Fig. 2b), it did strongly interact with all clade A PP2Cs except HAI1 and AHG1 (Supplementary Fig. 2a), suggesting that MeenPYL1 has gained PP2C binding ability but not the strong PP2C inhibitory activity. It should be noted that AHG1 is resistant to PYL-mediated inhibition since it lacks the conserved tryptophan for ABA-PYL-PP2C and PP2C-SnRK2 interactions[19,20,22,27-29]. We further confirmed physical interactions between MeenPYL3 and PP2Cs using LCI assays. The transient co-expression of *MeenPYL3-nLUC* and *cLUC-PP2Cs* generated LUC signals in *Nb* leaves (Fig. 2d), suggesting interactions between MeenPYL3 and *Arabidopsis* PP2Cs, including ABI1, HAB1, PP2CA and HAI1.

We further evaluated the PP2C inhibitory activity of ZcPYL8 and MeenPYL3 using in vitro SnRK2 kinase assays (Fig. 2e, f). The recombinant AtHAB1 and AtPP2CA phosphatases repressed the kinase activity of the recombinant SnRK2.6. The recombinant ZcPYL8 partially released SnRK2.6 from HAB1-mediated inhibition, while the recombinant MeenPYL3 partially released SnRK2.6 from PP2CA-mediated inhibition in an ABA-independent manner. We further assessed PP2C inhibitory activity using pNPP as substrate. The recombinant MeenPYL3 partially repressed phosphatase activity of HAB1 and ABI1 on pNPP, while the recombinant ZcPYL8 slightly repressed phosphatase activity of PP2CA on pNPP but not HAB1, in an ABA-independent manner (Fig. 2g–j and Supplementary Fig. 2b). The differential results between the two PP2C activity assays could be due to differences in substrates, since pNPP is not a physiological substrate, and PP2Cs inhibit SnRK2s not only by dephosphorylation but

also by physical interactions[28]. The AlphaFold3-predicted structure of MeenPYL3 was aligned to the ABA-PYL1-ABI1 complex (PDB: 3KDJ)[19] as a template, showing that R63 in MeenPYL (purple, corresponds to K86 in AtPYL1) leads to a clear steric hindrance (<2 Å) that blocks binding of ABA (in green; Fig. 2k). These results further confirm that PYL homologs obtained PP2C inhibition ability before the last common ancestor of Zygnematophyceae and land plants (Fig. 2l).

Structural analysis of PYLs identified four highly conserved loops (CL1–CL4) critical for ABA binding, among which the CL2 region, with a "S/TGLPA" motif, is crucial for the interactions between PYLs and PP2Cs[19]. Interestingly, these algal PYLs have a similar "S/TGL/IPG" motif, while the bacterial PYL homologs are distinguished from PYLs from algae and land plants in the CL2 region (Fig. 2a and Supplementary Fig. 1a). This partially explains the weak interaction of PrPYL-ABI1 and strong interactions between algal PYLs and *Arabidopsis* PP2Cs as well as the constitutive activation of *RD29B-LUC* expression in *pyl* duodecuple mutant protoplasts by algal PYLs (Figs. 1d and 2, and Supplementary Fig. 2a). In ABA-dependent or enhanced PYLs, including AtPYR1, AtPYL1/2/3/9/10, and OsPYL2, the ABA carboxylate forms charge-stabilized hydrogen bonds with the amino group of a conserved lysine in CL1, while the aliphatic portion of ABA interacts with the amino group of a conserved leucine in CL2[17-26]. The key lysine residue that binds to the carboxyl group of ABA is replaced by another basic residue, arginine (R), in the three MeenPYL homologs (Fig. 2a). We also analyzed the binding affinity between MeenPYL3 and ABA using ITC and found that MeenPYL3 has no binding affinity for ABA (Supplementary Fig. 2c). To test whether these sequence variations may explain the ABA insensitivity of MeenPYL3, we co-transfected *SnRK2.6, ABF2* with wild-type and mutated *MeenPYL3* in *pyl* duodecuple mutant protoplasts. Neither MeenPYL3-R63K nor MeenPYL3-R63K/I93L altered the ABA-independent phenotype in the *pyl* duodecuple mutant protoplasts under ABA treatment, and MeenPYL3-R63K even failed to induce *RD29B-LUC* expression (Supplementary Fig. 2d). To our surprise, the AtPYL4-K81R mutant lost its high basal activity in elevating *RD29B-LUC* expression (Supplementary Fig. 2d), suggesting that arginine is a better choice than lysine in CL1 to generate PYL ABA receptors with no basal activity. Since high basal stress signaling would have favored the terrestrialization of land plant progenitors, we propose that this critical arginine was not conserved during evolution to maintain high basal activity of PYLs.

## An ABA-independent PYL-like protein exists in *Arabidopsis*

The ABA-independent PYL-like proteins evolved before the last common ancestor of Zygnematophyceae and land plants, with some lineages subsequently gaining PP2C-inhibition capability. PYLs then acquired ABA-binding and -responsive functions in the last common ancestor of land plants[9,16]. The dimeric subfamily III PYLs in angiosperms exhibit strict ABA dependency[7], and some moss-specific PYLs also display strict ABA dependency despite being monomeric[16]. However, it is unclear whether ABA-independent PYL-like proteins exist and

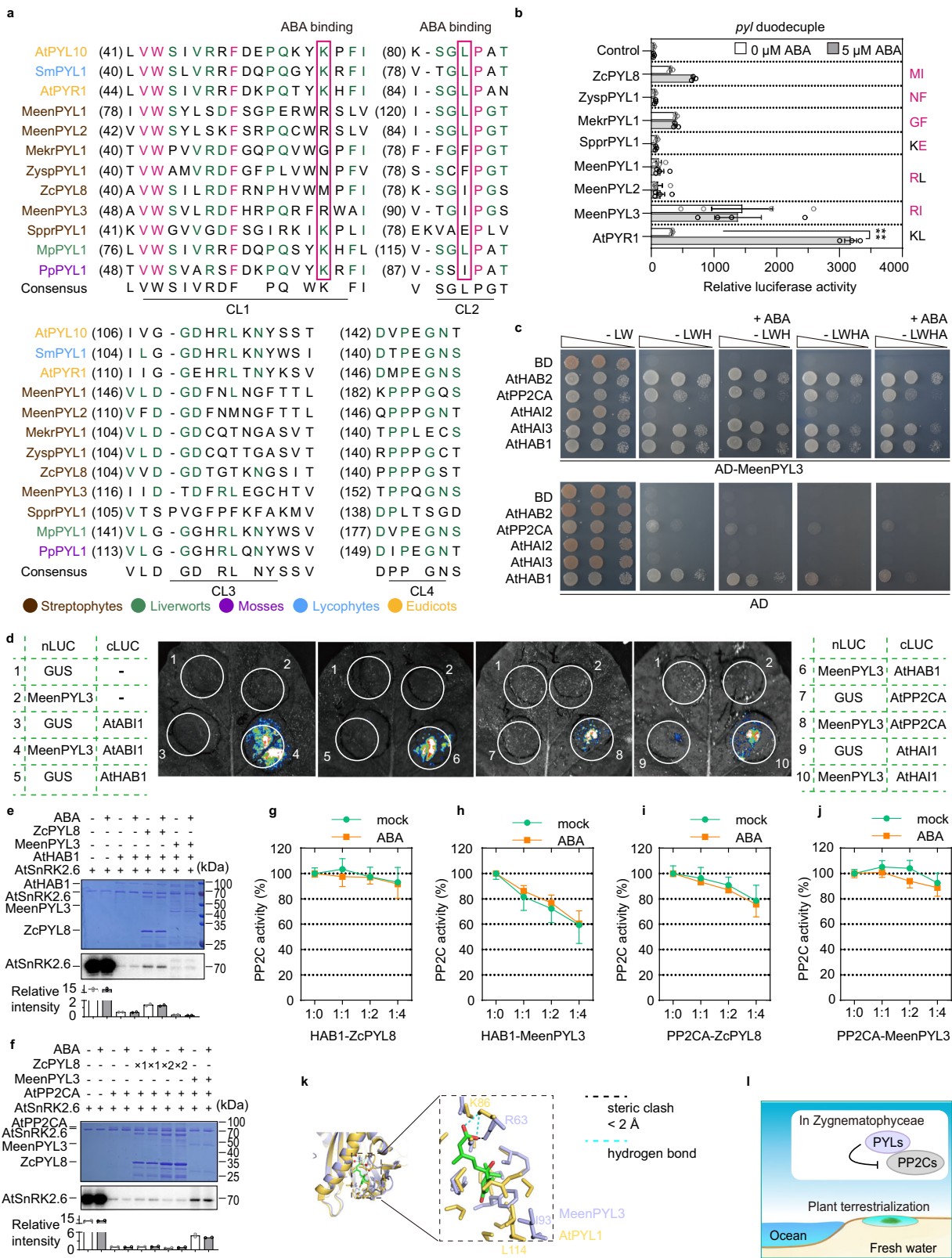

function in angiosperms. Therefore, we first evaluated the ABA dependence of PYLs from the model species *Arabidopsis thaliana* in the ABA-insensitive *pyl* duodecuple mutant (Fig. 3a). Among these ABA-responsive PYLs, the ABA-enhanced AtPYL5-6 and 8–12 exhibited basal activities, leading to the induction of *RD29B-LUC* expression, while the ABA-dependent AtPYR1, AtPYL1 and AtPYL7 could not induce *RD29B-LUC* expression without ABA treatment. In contrast, *AtPYL13*

failed to induce *RD29B-LUC* expression with or without ABA treatment. These results indicated that ABA-independent PYL-like proteins exist in angiosperms.

AtPYL13 differs from the other AtPYLs by two amino acid residues, which correspond to Q38 in CL1 and F71 in CL2 at AtPYL13 (Fig. 3b). The key lysine residue that binds to the carboxyl group of ABA is replaced by glutamine in AtPYL13, and the key leucine residue that binds to the

**Fig. 2 | Three PYL homologs from Zygnematales exhibit PP2C inhibitory activity. a** Sequence alignment of PYL homologs from algae (brown) and land plants. *At, Arabidopsis thaliana; Sm, Selaginella moellendorffii; Meen, Mesotaenium endlicherianum; Mekr, Mesotaenium kuamstae; Zysp, Zygnemopsis sp.; Zc, Zygnema circumcarinatum; Sppr, Spirogyra sp.; Mp, Marchantia polymorpha; Pp, Physcomitrella patens.* Two key residues for ABA binding are indicated by red rectangles. **b** *RD29Bpro:LUC* expression in *pyl* duodecuple mutant protoplasts transformed with ancestral Zygnematophyceae *PYLs*. Transformation with *SnRK2.6, ABF2*, and *RD29Bpro:LUC* served as control. Error bars, SEM (*n* ≥ 3 biological repeats). Two-way ANOVA, ****$p < 0.0001$. Right: the two residues critical for ABA-binding (black) and the corresponding variant residues (red). Expression data of control and PYR1 were shared by fig. 1b. **c** Interactions between MeenPYL3 and *Arabidopsis* PP2Cs in Y2H assay. Yeast cells were grown on nonselective SD/−LW and selective SD/−LWH or SD/−LWHA media, without or with 10 μM ABA. **d** Interactions between MeenPYL3 and *Arabidopsis* PP2Cs in *N. benthamiana* leaves assessed by firefly luciferase complementation imaging (LCI). **e, f** In vitro kinase assays testing whether ZcPYL8

or MeenPYL3 could relieve PP2C-mediated inhibition of SnRK2.6 with or without 100 μM ABA. ZcPYL8, but not MeenPYL3, released SnRK2.6 from HAB1-mediated inhibition (**e**). In contrast, MeenPYL3, but not ZcPYL8, released SnRK2.6 from PP2CA-mediated inhibition (**f**). SnRK2.6 autophosphorylation was detected by autoradiography (upper panels). **g–j** PP2C enzyme activity assays using pNPP as substrate. MeenPYL3 inhibited HAB1 (**h**), whereas ZcPYL8 inhibited PP2CA (**i**), both irrespective of 10 μM ABA. Error bars, SD (*n* ≥ 6 biological repeats). **k** Predicted structure of MeenPYL3 (purple, Alphafold3) aligned with the reported complex (PDB ID: 3KDJ) containing ABA (green/red), AtPYL1 (gold), and AtABI1. Residues within a 5 Å radius of ABA are shown (right). AtPYL1 (K86/L114) and corresponding residues in MeenPYL3 (R63/I93) are labeled. Cyan dashed lines represent hydrogen bonds between ABA and K86 in AtPYL1, while the black dashed line indicates steric hindrance (<2 Å) between ABA and R63 in MeenPYL3. **l** Model depicting that certain Zygnematophyceae PYLs gained PP2C inhibitory activity prior to the last common ancestor of Zygnematophyceae and land plants. Exact *p* values for (**b, e, f**) are provided in Source Data.

---

aliphatic portion of ABA is replaced by phenylalanine in AtPYL13 (Fig. 3b). To test whether these sequence variations may explain the ABA insensitivity of AtPYL13, we co-transfected *SnRK2.6, ABF2* with wild-type and mutated *AtPYL13* in *pyl* duodecuple mutant protoplasts. The wild-type *AtPYL13* could not activate *RD29B-LUC* under ABA treatment, while the expression of *AtPYL13-Q38K/F71L* restored the ABA-dependent activation of *RD29B-LUC* in the *pyl* duodecuple mutant protoplasts (Fig. 3c), indicating that AtPYL13 is otherwise functional and that the two natural sequence variations in the CL1/CL2 domains confer ABA independence to AtPYL13. Phylogenetic analysis of Brassicaceae PYLs revealed a stepwise transition from ABA-enhanced receptors to ABA-independent paralogs after gene family expansion (Supplementary Fig. 3a; steps i–iii). This progression is evidenced by: (i) initial divergence of the *AtPYL11/12/13* clade[30] (node marked with a purple asterisk), which has only been identified in eudicots; (ii) secondary divergence of *AtPYL13* (gold-highlighted branch) from the *AtPYL11/12* subclade; (iii) subsequent acquisition of lineage-specific variations within the *AtPYL13* subclade, including one in *Camelina sativa* and two in the *Arabidopsis* lineage (*A. thaliana, A. lyrata*, and *A. halleri*; variation sites labeled in red). These changes support the hypothesis that ABA-independent PYL-like proteins reemerged in the Brassicaceae family.

There are nine members of clade A PP2Cs in *Arabidopsis*, and the protoplast transient expression assays suggested that AtPYL13 cannot inhibit most, if not all, PP2Cs in the presence of ABA. We thus reconstituted the ABA signaling pathway in vitro, using the recombinant MBP-tagged SnRK2.6, GST-tagged ABI1 and PP2CA, and His-tagged AtPYL13. The SnRK2 activity was evaluated by in vitro kinase assay with (γ-$^{32}$P) ATP (Fig. 3d, e), anti-phospho-S175-SnRK2s antibody (Supplementary Fig. 3b) that recognizes activated SnRK2s[4], and PP2C enzymatic assay using pNPP as substrate (Supplementary Fig. 3c). The incubation of ABI1 or PP2CA with SnRK2.6 repressed SnRK2.6 activity. The repression of SnRK2.6 by PP2CA was released by wild-type AtPYL13 in the absence or presence of ABA, suggesting AtPYL13 constitutively represses PP2CA in an ABA-independent manner (Fig. 3e). In contrast, AtPYL13-Q38K/F71L repressed PP2C phosphatase activity and released PP2C-mediated inhibition of SnRK2 activity in an ABA-enhanced manner (Fig. 3d, e, and Supplementary Fig. 3b, c).

Due to the very weak phenotype of *pyl13* mutants[31], we then evaluated the properties of AtPYL13 in the *pyr1pyl1pyl2pyl4* (*pyr1-pyl124*) quadruple mutant background. We generated transgenic *Arabidopsis* overexpressing either *AtPYL13* or *AtPYL13-Q38K/F71L* driven by the constitutive *35S* cauliflower mosaic virus (CaMV) promoter (Supplementary Fig. 3d). We found that ectopic expression of *AtPYL13-Q38K/F71L* complemented the ABA-hypersensitive phenotype of the *pyl* quadruple mutant during seedling establishment and subsequent growth (Fig. 3f, g). In contrast, ectopic expression of *AtPYL13* did not complement the ABA-hypersensitive phenotype of the *pyl* quadruple

mutant plants. Together with our previous data[31,32], these results indicate that AtPYL13 has ABA-independent PP2C inhibitory activity and that the two natural amino acid variations in AtPYL13 confer ABA insensitivity.

## Two invariant residues are crucial for PYL ABA receptors

To further address the effects of the noncanonical amino acid residues in AtPYLs on ABA dependence, mutated AtPYLs were generated and used in the transient activation assay in *pyl* duodecuple mutant protoplasts. The expression of 12 mutated *AtPYLs*, except *AtPYL8-K61Q/L87F*, did not activate *RD29B-LUC* expression (Fig. 3h). Expression of *AtPYL8-K61Q/L87F* enabled weak ABA-enhanced activation of *RD29B-LUC* expression, likely due to residual ABA dependence. While co-transfection of wild-type *AtPYL8* with *ABI1* enabled ABA-induced *RD29B-LUC* expression, co-transfection of *AtPYL8-K61Q/L87F* with *ABI1* failed to induce expression even with ABA treatment. This suggests AtPYL8-K61Q/L87F lacks strong ABI1-inhibitory capacity. We then analyzed the ABA-binding properties of wild-type PYL9 and its K63Q/L89F mutant using ITC and found that this mutation abolished the ABA-binding affinity of PYL9 (Supplementary Fig. 3e, f). We further evaluated the ABA dependence of wild-type and mutated AtPYL9 with the reconstituted ABA signaling pathway in vitro. The incubation of ABI1 with SnRK2.6 repressed SnRK2.6 activity. The wild-type AtPYL9 blocked the phosphatase activity of ABI1 and HAB1 and released the repression of SnRK2.6 by PP2Cs in an ABA-dependent manner (Fig. 3i and Supplementary Fig. 3g–i). In contrast, AtPYL9-K63Q/L89F exhibited limited or no inhibition of ABI1 or HAB1 in an ABA-independent manner (Fig. 3i and Supplementary Fig. 3g–i). Moreover, the expression of native promoter-driven *AtPYL1-K86Q/L114F* did not complement the ABA-hyposensitive phenotype of the *pyl* quadruple mutant plants (Fig. 3j, k, Supplementary Fig. 3j). As a positive control, the expression of native promoter-driven *AtPYL1* partially complemented the ABA-hyposensitive phenotype of the *pyl* quadruple mutant during seedling establishment (Fig. 3j, k). These results indicate that the invariant lysine and leucine residues are crucial for ABA dependence of plant PYL ABA receptors.

## ABA-independent PYL-like proteins exist in seed plants

We further identified putative ABA-independent PYL-like proteins in seed plants according to natural variations in the key lysine and leucine residues (Fig. 4a and Supplementary Data 1). In addition to AtPYL13, several AtPYL13-like proteins differ from the ABA-dependent PYLs by one or two amino acid residues, which correspond to Q38 in CL1 and/ or F71 in CL2 in AtPYL13 (Fig. 4b). The key lysine residue that binds to the carboxyl group of ABA is replaced by either glutamine in the Brassicaceae family (*Arabidopsis lyrata, Camelina sativa, Capsella rubella*) and in the genus *Oryza*, or by threonine in the genus *Hordeum* (Fig. 4b). Moreover, phenylalanine replaces the key leucine residue

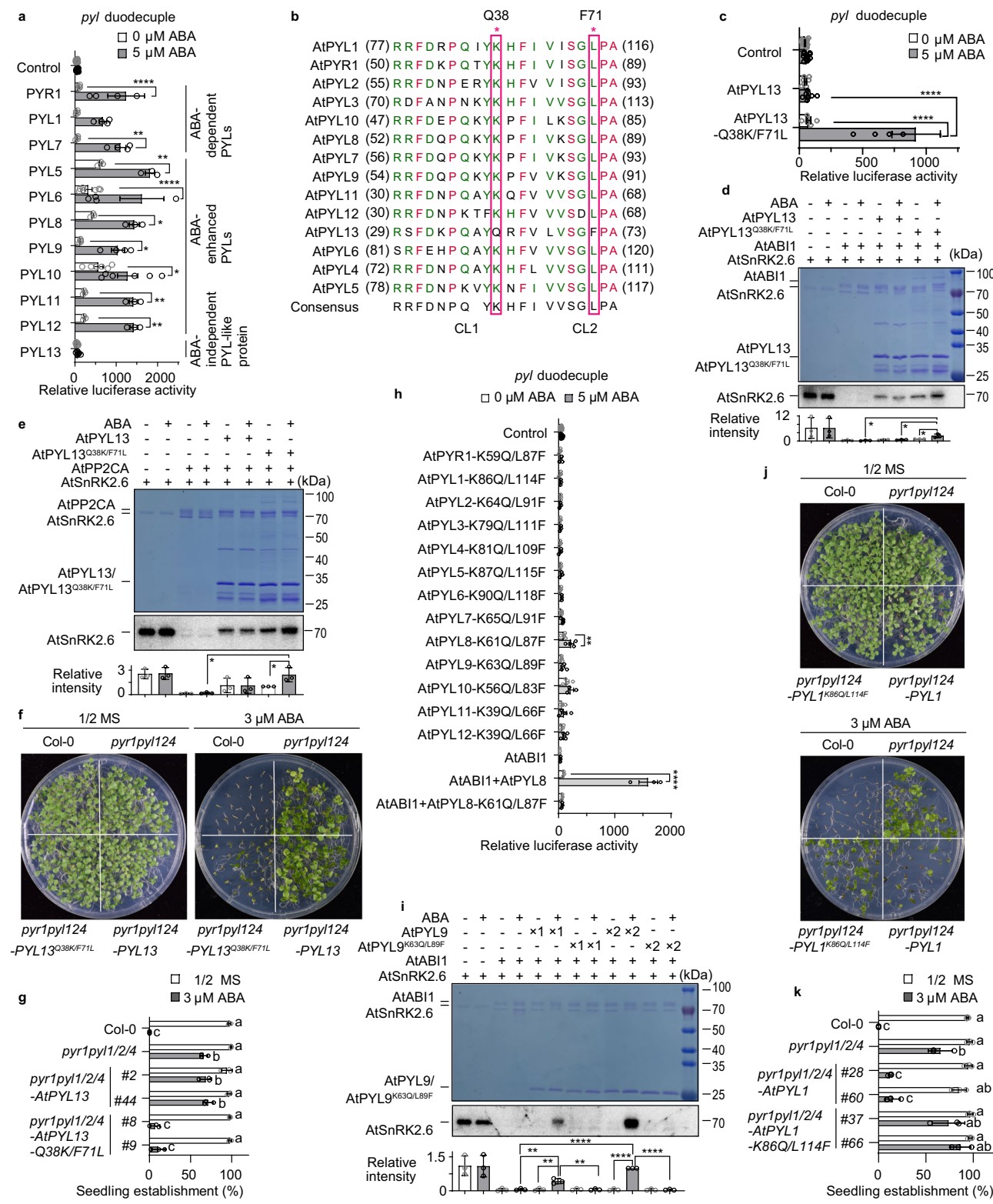

that binds to the aliphatic portion of ABA in PYLs in the genera *Arabidopsis*, *Capsicum*, *Nicotiana*, and *Oryza* (Fig. 4b). We evaluated the ABA dependence of these AtPYL13-like proteins in *pyl* duodecuple mutant protoplasts. Overexpression of *PYLs* from the genera *Camelina* (*CsPYL13L1*), *Capsicum* (*CaPYL13L2*), *Nicotiana* (*NsPYL13L* and *NaPYL13L*), and *Hordeum* (*HvPYL13L1/2*) could not induce expression of *RD29B-LUC*, while overexpression of *PYLs* from the genera *Oryza* (*OsPYL12*) and *Capsella* (*CrPYL13L*) constitutively activated expression of *RD29B-LUC* in an ABA-independent manner (Fig. 4c). These results

indicated that these alterations are associated with ABA dependence of PYL proteins and suggested that the transformation from ABA receptors to ABA-independent PYL-like proteins occurred in seed plants after expansion of the gene family (Supplementary Fig. 3a and Supplementary Data 1).

According to the Klepikova Arabidopsis Atlas eFP browser[33], *AtPYL13* is specifically expressed in dry seeds but not in other tissues (Supplementary Fig. 4a). Next, we investigated the expression pattern of ABA-independent *PYL*-like genes in seed plants. According to the

**Fig. 3 | Two invariant residues are crucial for the ABA dependence of plant PYLs. a, c, h** *RD29Bpro:LUC* expression in *pyl* duodecuple mutant protoplasts transformed with *Arabidopsis PYLs* (**a**), *AtPYL13-Q38K/F71L* (**c**), and variants of *AtPYLs* (**h**). Transformation with *SnRK2.6*, *ABF2* and *RD29Bpro:LUC* served as control. Error bars, SEM, $n \geq 3$ biological repeats in (**a, h**), and $n \geq 6$ biological replicates in (**c**). Two-way ANOVA, $*p < 0.05$, $**p < 0.01$, $****p < 0.0001$. Expression data of control was shared among figs. 3a, c, h and 4c. **b** Sequence alignment of ABA-binding regions in AtPYLs. The two residues in PYL13 that differ from other PYLs are labeled with red asterisks and rectangles. **d, e** In vitro kinase assays assessed whether wild-type or mutated AtPYL13 could relieve PP2C-mediated inhibition of SnRK2.6 with or without 100 μM ABA. SnRK2.6 activity was repressed by AtABI1 (**d**) and AtPP2CA (**e**). SnRK2.6 autophosphorylation (autoradiography, lower panels) and protein loading (Coomassie blue staining, upper panels) are shown. Radioactive intensity of the bands was normalized to that of the AtPYL13-Q38K/F71L sample without ABA. Error bars, SD (n = 3 biological replicates). Two-sided Student's *t* test, $*p < 0.05$. **f, g** The ABA-hyposensitive phenotype of the *pyr1pyl124* quadruple mutant was complemented by *35Spro:AtPYL13-Q38K/F71L* but not by *35Spro:AtPYL13* (**f**). Seedling establishment was quantified (**g**). Error bars, SD (n = 3 biological replicates). Two-way ANOVA followed by a Tukey test. **i** Release of SnRK2.6 from ABI1 inhibition by AtPYL9 and AtPYL9-K63Q/L89F, with or without 100 μM ABA, during in vitro kinase assays. SnRK2.6 autophosphorylation was detected by autoradiography (lower panel). Radioactive intensity of the bands was normalized to that of the AtPYL9 sample (2×) with ABA. Error bars, SD (n = 3 biological replicates). Two-sided Student's *t* test, $**p < 0.01$, $****p < 0.0001$. **j, k** The ABA-hyposensitive phenotype of *pyr1pyl124* was complemented by *AtPYL1-pro:AtPYL1* but not by *AtPYL1pro:AtPYL1-K86Q/L114F* (**j**). Seedling establishment was quantified (**k**). Error bars, SD (n = 3 biological replicates). Two-way ANOVA followed by a Tukey test. Exact *p* values for (**a, c, d, e, g–i**, and **k**) are provided in Source Data. For figures (**d, e**, and **i**), images shown are representative of three independent experiments.

---

eFP browser on the *Nicotiana attenuata* Data Hub[34], *NaPYL13L* (*NIATv7_g33692*) is highly expressed in *Nicotiana attenuata* seeds but not in other tissues (Supplementary Fig. 4b). The expression of *NaPYL13L* was dramatically reduced after seed imbibition and was undetectable after seedling establishment (Fig. 4d). In contrast, the expression of *NaPYL4L* (*NIATv7_g37793*) was undetectable in dry seeds but was elevated after seed imbibition, while *NaPYL8L* (*NIATv7_g17046*) was expressed in dry and imbibed seeds (Fig. 4d). In *Camelina sativa*, *CsPYL13L2* was also highly expressed in seeds but not in other tissues, according to ePlant *Camelina* (Fig. 4e). Moreover, *HvPYL13L* (*HORVU.MOREX.r3.3HG0247520*) had the highest expression level in barley seeds compared to other tissues, according to RNA-Seq of 16 developmental stages of barley (PRJEB14349) (Fig. 4f). In rice, *OsPYL12* was expressed in imbibed rice seeds and seedlings but not in dry seeds, while *OsPYL13* was highly expressed in dry seeds with reduced expression after seed imbibition (Fig. 4g). The F76 variation in OsPYL12 (lavender) causes steric clash (atom distance <2 Å) with ABA, explaining its ABA independence (Fig. 4h). ITC analysis confirmed OsPYL12 cannot bind ABA (Supplementary Fig. 4c). These results indicated that most ABA-independent *PYL*-like genes are expressed in dry seeds, which differs from the expression pattern of ABA-dependent or ABA-enhanced *PYLs* (Fig. 4i).

### ABI3 mediates seed-specific expression of *AtPYL13*

We evaluated the expression of the 14 *Arabidopsis PYLs* during seed development (Fig. 5a). Expression of *AtPYL11, 12*, and *13* was drastically elevated from 12 days after flowering and peaked at 15-18 days after flowering (Fig. 5a and Supplementary Fig. 5). On the other hand, the expression of *PYR1* and *PYL2/5/7/9* gradually elevated during seed maturation, although at levels that were orders of magnitude lower than that of *AtPYL11/12/13*, while expression of *PYL1/3/4/6/8/10* gradually decreased during seed maturation (Fig. 5a). We further confirmed the seed-specific expression of *AtPYL13* using transgenic lines expressing a β-glucuronidase-encoding gene driven by the native *AtPYL13* promoter. GUS staining results confirmed that the *AtPYL13* promoter was highly active in *Arabidopsis* seeds 15 days after flowering (Fig. 5b). These results indicate that *AtPYL13* is a seed-specific ABA-independent *PYL*-like gene.

Analysis of the promoters of *AtPYL11/12/13* and *AtPYL13*-like genes in Brassicaceae identified the presence of RY motifs and ABRE core motifs (Fig. 5c and Supplementary Fig. 5d). The transcription factor ABI3 directly binds to RY motifs via its B3 domain and may also indirectly interact with ABRE motifs[35]. We next investigated whether *ABI3* controls the seed-specific expression of *AtPYL13*. The drastic elevation of *AtPYL11/12/13* expression during seed maturation was nearly blocked in the *abi3-6* mutant, and the gradual increases of *AtPYL2/5/9* were also defective in the *abi3-6* mutant (Fig. 5a and Supplementary Fig. 5). Moreover, the strong GUS staining of *PYL13pro:GUS* seen in dry seeds of the Col-0 WT background was dramatically reduced in the *abi3-6* mutant background (Fig. 5d). To investigate binding between ABI3 and the *AtPYL13* promoter, we performed yeast one-hybrid (Y1H) and electrophoretic mobility shift assay (EMSA) to test whether ABI3 could bind the RY motif-containing region in vitro. The GAL4AD fusion of full-length ABI3 did not support yeast growth on synthetic dropout (SD)-Ura/Leu media, and the GAL4AD fusion of ABI3 or ABI3-B3 did not support yeast growth on SD-Ura/Leu media supplemented with Aureobasidin A (AbA) (Fig. 5e). The R64K/R66K/P69S (KKS) substitutions in ABI3-B3 allow yeast growth in the Y1H system and enhance DNA binding without compromising sequence specificity[36]. These mutations (ABI3-B3-KKS) promoted yeast growth only with the *PYL13* promoter fragment containing a wild-type RY motif but not with a mutated RY motif (Fig. 5e). Recombinant His-tagged full-length ABI3 bound to the Cyanine5 (Cy5)-labeled *PYL13* promoter fragment containing the RY motif (Fig. 5f). Addition of wild-type unlabeled probes, but not unlabeled probes carrying a mutated RY motif, repressed the interaction between His-ABI3 and the Cy5-labeled *PYL13* promoter fragment (Fig. 5f). Chromatin immunoprecipitation (ChIP)-qPCR assays using mature green seeds (about 15 days after flowering) of *OE-ABI3-Flag* plants revealed significant enrichment of fragments with the RY motif (Fig. 5g). These results indicated that the seed-specific expression of *AtPYL13* is transcriptionally regulated by ABI3 (Fig. 5h).

## Discussion

PYL-mediated ABA signaling plays a central role in plant survival under water deficiency[3,4]. Under unstressed conditions, the clade A PP2Cs constantly inhibit the SnRK2s, including SnRK2.2, 2.3 and 2.6[37]. Under drought, plants accumulate ABA and, in turn, trigger the activation of ABA receptor PYLs[38]. The ABA-bound PYLs interact with and inhibit PP2Cs to release SnRK2s and activate stress signaling[5,6]. Although SnRK2s and PP2Cs are evolutionarily conserved key regulators originating in algae, the ABA receptor PYLs are not present until the last common ancestor of land plants[9]. The evolution of ABA receptor PYLs requires at least three critical steps: PP2C binding, PP2C inhibition, and ABA binding and responses. Our results showed that PYL-like proteins are present in bacteria and might have gained PP2C binding ability in some soil bacteria. Plants likely acquired *PYL* genes horizontally from bacteria before the last common ancestor of Zygnematophyceae and land plants, when plant PYLs obtained PP2C inhibition ability (Figs. 1, 2 and 6). *Paraburkholderia* species are the predominant symbionts of *Mimosa* spp. and common bean[39], and further investigation is required to uncover whether *PrPYL* contributes to the host-microbe interaction during symbiosis. PYLs subsequently gained ABA-binding ability and ABA-enhanced PP2C inhibition in the last common ancestor of land plants, while the dimeric PYLs further evolved the completely ABA-dependent PP2C inhibitory activity in the last common ancestor of angiosperms[9]. Independently, the monomeric PYLs also evolved strict ABA dependency in mosses[16]. The ABA-binding properties of

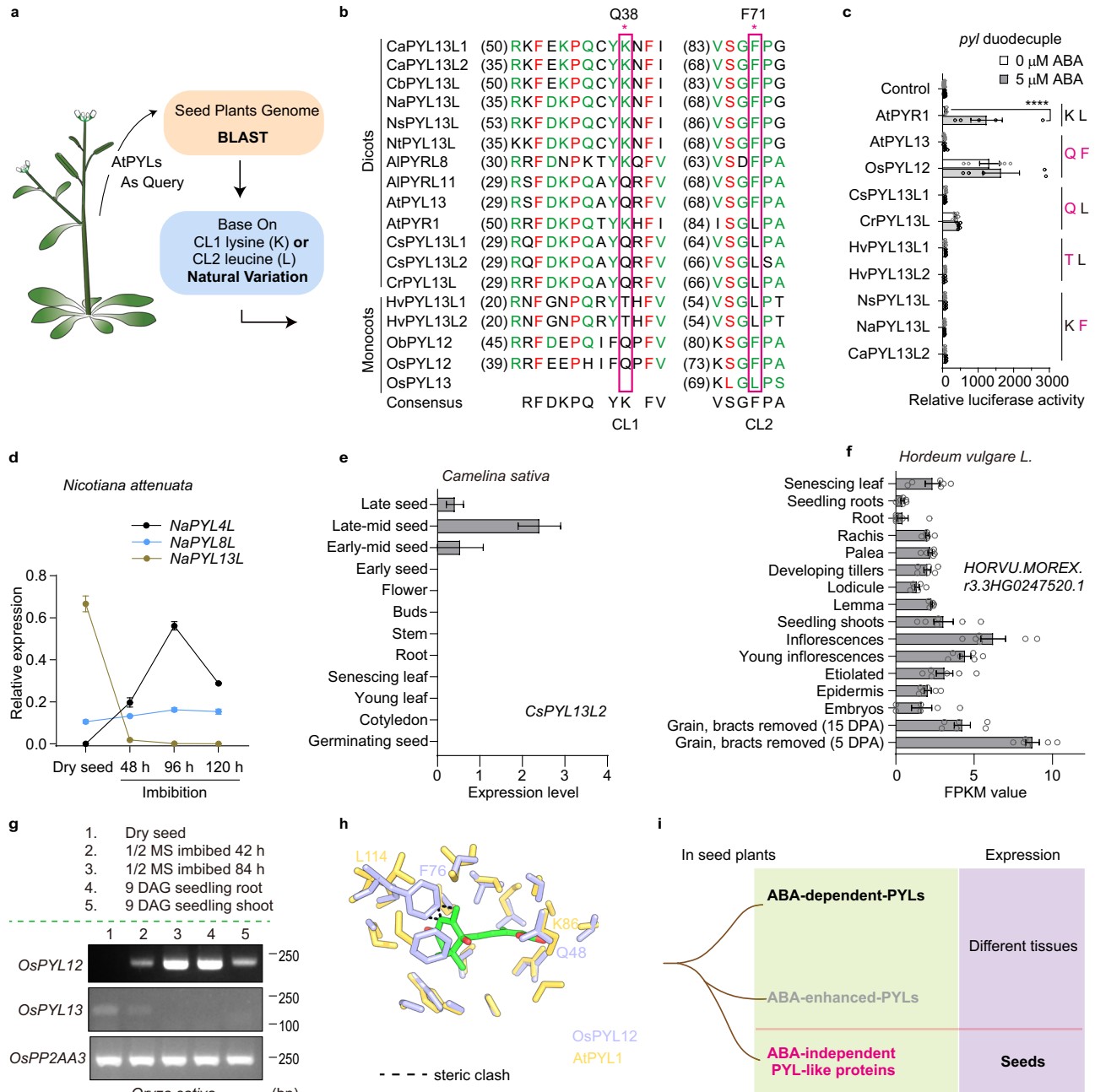

**Fig. 4 | ABA-independent PYL-like proteins exist in seed plants. a** Putative ABA-independent PYL-like proteins in seed plants that differs from ABA receptors at the key lysine and leucine residues were identified. **b** Sequence alignment of the ABA-binding regions of putative ABA-independent PYL-like proteins in seed plants. *Ca, Capsicum annuum*; *Cb, Capsicum baccatum*; *Na, Nicotiana attenuata*; *Ns, Nicotiana sylvestris*; *Nt, Nicotiana tomentosiformis*; *Al, Arabidopsis lyrata*; *Cs, Camelina sativa*; *Cr, Capsella rubella*; *Hv, Hordeum vulgare*; *Ob, Oryza brachyantha*; *Os, Oryza sativa*. The two residues that differ from those in canonical ABA receptors are labeled with asterisks and rectangles. **c** Induction of ABA-responsive gene expression by putative ABA-independent PYL-like proteins. *RD29Bpro:LUC* expression was measured in *pyl* duodecuple mutant protoplasts transformed with *PYLs*. Transformation with *SnRK2.6*, *ABF2* and *RD29Bpro:LUC* served as control. Error bars, SEM (*n* ≥ 5 independent repeats). Two-way ANOVA, ****$p < 0.0001$. Right panel, two residues critical for ABA binding (black) and the corresponding variant residues (red). Expression data of control was shared among figs. 3a, c, h and 4c. **d** Relative expression of ABA-independent *NaPYL13L* in dry seeds, imbibed seeds, and 5-day-

old seedlings of *Nicotiana attenuata*, compared to the expression of the ABA-responsive *NaPYL4L* and *NaPYL8L* genes. *NaIF5α* served as an internal reference. Error bars, SD (*n* = 3 technical repeats). **e** Relative expression levels of *CsPYL13L2* in different tissues and developmental stages of *Camelina sativa*. Data source: ePlant *Camelina*. **f** Relative expression levels of *HORVU.MOREX.r3.3HG0247520.1* in *Hordeum vulgare* tissues from ePlant Barley v3. The *HvPYL13L* gene encodes both HvPYL13L1 and HvPYL13L2 shown in **c**. Error bars, SEM (*n* = 6 independent repeats). **g** Expression of *OsPYL12* and *OsPYL13* in dry seeds, imbibed seeds, and 9-day-old rice seedlings (Nipponbare) was analyzed by RT-PCR. *OsPP2AA3* served as an internal reference. DAG, days after germination. **h** Structural comparison of OsPYL12 (purple, predicted by AlphaFold3) aligned with the ABA–AtPYL1 (gold) complex. ABA and residues within a 5 Å radius are shown. The black dashed line indicates steric hindrance (<2 Å) between ABA and F76 in OsPYL12. **i** A model illustrating seed-specific expression of ABA-independent PYL-like proteins in dicot seed plants. Source data are provided as a Source Data file.

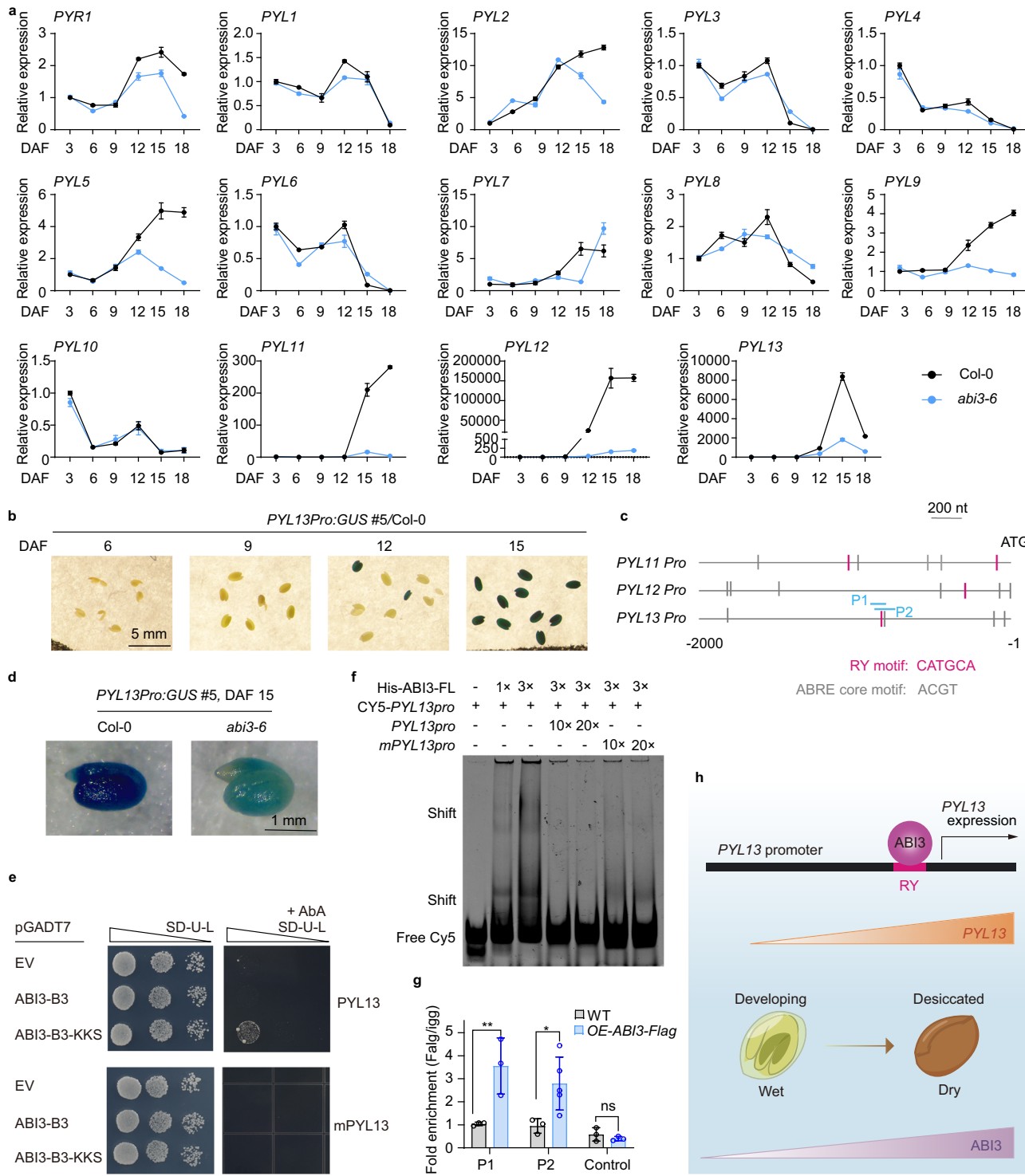

PYLs confer ABA-dependent or ABA-enhanced PP2C-inhibition activity, which enabled land plants to change constitutive activation into the energy-saving conditional activation of stress signaling. Together with previous findings[9,14,16], we outline the evolution of PYLs, while our results support the hypothesis that *PYL*s originated via horizontal gene transfer from soil bacteria (Fig. 6).

Pseudo-receptors, a concept similar to pseudo-enzymes, have long been regarded as evolutionary intermediates that are often overlooked[40]. We reported here that a few pseudo-receptors that share sequence homology with the PYL ABA receptor family lack ABA dependence due to sequence variations in otherwise conserved lysine and leucine residues (Figs. 3 and 4). The evolutionary "jump" that

transforms the pseudo-receptors (e.g., PrPYL, ZcPYL8, ZyspPYL1, MekrPYL1, and MeenPYLs) into ABA receptors occurred in the last common ancestor of Zygnematophyceae and land plants. Interestingly, the ABA-independent PYL-like proteins exist patchily in some land plants (Figs. 3 and 4), indicating that the transformation from ABA receptors to pseudo-receptors happened occasionally during the evolution of land plants after expansion of the gene family (Supplementary Fig. 3a). Most of these ABA-independent *PYL*-like genes are specifically expressed in seeds (Figs. 4 and 5), suggesting that seed-specific expression of *PYL*s relieves corrective selection pressures. Although most of the ABA-independent PYL-like proteins could not induce stress responses in the *pyl* duodecuple mutant protoplasts

**Fig. 5 | ABI3 mediates seed-specific expression of *AtPYL13* in *Arabidopsis.***
**a** Relative expression of *Arabidopsis PYLs* during seed development in Col-0 and *abi3-6*. Transcriptional profiles of *PYLs* were determined by real-time quantitative PCR in developing seeds from 3 to 18 days after flowering (DAF). *PP2AA3* was used as an internal reference. Data were integrated after normalizing the relative expression levels of Col-0 at 3 DAF to 1. Error bars, SD (*n* = 3 technical repeats). **b** *PYL13pro: GUS* expression patterns in developing *Arabidopsis* seeds at 6, 9, 12 and 15 DAF in Col-0 wild type. Scale bar, 5 mm. **c** Schematic illustration of candidate ABI3-binding motifs in the promoters of *AtPYL11/12/13*. nt, nucleotide. **d** *PYL13pro:GUS* expression patterns in dry seeds of Col-0 and *abi3-6* at 15 DAF. Scale bar, 1 mm. **e** Interaction between the *AtPYL13* promoter fragment and the wild-type or R64K/R66K/P69S (KKS) substitutions of the ABI3-B3 domain in yeast one-hybrid (Y1H) assay. The *AtPYL13* promoter fragment contained either the wild-type (upper panels) or a mutated RY motif (mPYL13, lower panels). Dilutions ($10^{-1}$ and $10^{-2}$) of liquid cultures with equal density were spotted onto plates lacking Ura/Leu (−UL) without or with Aureobasidin A (Aba; 1 μg/ml in upper panel, 0.2 μg/ml in lower panel). An empty vector (EV) served as a negative control. **f** Interaction between full-length ABI3 protein and the Cyanine5 (Cy5)-labeled *PYL13* promoter fragment containing wild-type or mutated RY motif in electrophoretic mobility shift assay (EMSA). Excess unlabeled wild-type or mutated *PYL13* promoter fragments were added as competitors. **g** ChIP-qPCR assays were performed using 2 g samples of seeds (-15 DAF) from Col-0 and *OE-ABI3-Flag* plants. Anti-Flag antibody was used to enrich 200–250 bp *PYL13* promoter fragments (P1 and P2, as indicated in **c**) containing the RY motif. A segment lacking the RY motif served as a control. Error bars, SD (*n* ≥ 3 technical repeats, two-way ANOVA, *$p$ < 0.05, **$p$ < 0.01). Exact $p$ values are provided in the Source Data. **h** Model illustrating the transcriptional regulation of seed-specific expression of *AtPYL13* by ABI3 in *Arabidopsis*.

(Fig. 4c), we cannot exclude the possibility that they may repress some clade A PP2C. For example, AtPYL13 suppresses PP2CA but not HAB1 and AHG1[31,32] (Fig. 3e). Therefore, seed-specific expression of these ABA-independent *PYL*-like genes may contribute to high basal stress signaling in dry seeds for seed vigor maintenance, thus relieving selective pressure in vegetative tissues from constitutively active pseudo-receptors. Future investigations are needed to clarify the existence, evolution, and function of pseudo-receptors for other plant hormones or signaling molecules.

Early land plants and seeds mainly adopt the "drying without dying" strategy and require constitutive activation of stress signaling for osmoregulation. ABA-independent *PYL*-like genes function redundantly with ABA receptors to regulate seed viability through SnRK2s and ABFs[4,10,12,41]. During seed maturation, plants accumulate high levels of ABA, and ABA levels gradually decrease during vernalization[42]. Therefore, enhanced basal stress signaling mediated by both ABA-enhanced and ABA-independent PYLs is likely required for maintaining seed viability at stages with low ABA concentrations, e.g., after seed vernalization. In *Arabidopsis*, the seed-specific expression of *AtPYL11*, *AtPYL12*, and *AtPYL13* is regulated by transcription factor ABI3 (Fig. 5). While ABI3 is best known for its high expression in seeds, it has also been implicated in other developmental processes, including bud dormancy[43], lateral root emergence[44], and lamina inclination[45]. Here, our work provides another interesting example of how ABI3 mediates local signaling. Future investigations are needed to clarify if and how seed-specific transcription factors control the seed-specific expression of other ABA-independent *PYL*-like genes (Supplementary Fig. 5d). Taken together, PYLs evolved in a binary pattern in seeds and vegetative tissues, and the ABA-independent *PYL*-like proteins may have critical roles in basal stress signaling during seed desiccation.

## Methods
### Plant materials
The *pyr1pyl124* quadruple and *pyr1pyl12458379101112* duodecuple mutants used in the present study have been described previously (Park et al., 2009; Zhao et al., 2018). To generate *35Spro:AtPYL13* transgenic plants in the *pyr1pyl124* quadruple mutant background, the coding sequences of *AtPYL13* and *AtPYL13-Q38K/F71L* were cloned into a *pGWB517* vector under the control of the *35S* promoter, after which the construct was transformed into the *pyr1pyl124* quadruple mutant. To generate *AtPYL1pro:AtPYL1* transgenic plants in *pyr1pyl124* quadruple mutant background, the coding sequences of *AtPYL1* and *AtPYL1-K86Q/L114F* were fused to the *AtPYL1* promoter and cloned into *pGWB513*, which was then transformed into the *pyr1pyl124* quadruple mutant. To generate *PYL13pro:GUS* transgenic plants, the *GUS* gene was driven by a −2000 to −1 bp fragment of *PYL13* promoter containing RY motif. The *PYL13pro:GUS* transgenic plant in the *abi3-6* mutant background was generated by a genetic cross and subsequent genotyping.

### Phenotypic analysis
Arabidopsis seeds were collected from wild type Col-0, the *pyr1pyl124* quadruple mutant, and T2 or T3 generation transgenic Arabidopsis plants grown under the same well-watered conditions, after which they were allowed to dry for at least 2 weeks, surface sterilized and then sown onto 1/2 MS plates supplemented with or without 3 μM ABA. After stratification in the dark at 4 °C for 3 days, the seeds were grown horizontally in a Percival CU36L4 growth chamber. Seedling establishment was scored as the percentage of seeds with green expanded cotyledons on the 12th day. Seedling establishment images were collected on the 15th day.

### Multiple sequence alignment and phylogenetic analysis
Multiple sequence alignment and phylogenetic analysis were conducted using MEGA12 software, with the ClustalW and maximum likelihood method and the Jones-Taylor-Thornton model of amino acid substitutions.

### Transient expression assay in Arabidopsis protoplasts
The putative ABA-irresponsive PYL homologs from bacteria, algae, and land plants were synthesized and cloned into the *pHBT95* vector. All plasmids were confirmed by sequencing and purified with the QIAGEN Plasmid Maxi Kit.

The *pyr1pyl12458379101112* duodecuple mutant was grown under short-day conditions and high humidity (10 h of light/14 h of darkness, 80% humidity). Rosette leaves of 4-week-old plants were used to generate protoplasts, as previously reported (Fujii et al., 2009; Zhao et al., 2013). Protoplast expression constructs, including *pHBT95-PYL*, *-ABI1*, *-SnRK2.6*, *-ABF2*, *RD29B-LUC*, and *ZmUBQ-GUS* constructs, were the same as previously reported (Fujii et al., 2009; Zhao et al., 2013). All mutants were generated via two-step PCR and verified by plasmid sequencing. Assays for transient expression in protoplasts were performed as described previously (Fujii et al., 2009; Zhao et al., 2013), and all steps were performed at room temperature. For each transform, 6 μg *RD29B-LUC*, 3 μg *ABF2*, 3 μg *SnRK2.6* and 0.5 μg *ZmUBQ-GUS* were added as essential control. Besides, none or 3 μg of different candidate *PYL* was added as an effector to test ABA-receptor identities. After transfection, the protoplasts were washed twice using W5 buffer and incubated in WI solution without ABA or with 5 μM ABA under light for 16 h.

Introduction of *PYL* receptors is sufficient to recover ABA response in the *pyl* duodecuple protoplast transient assay, while addition of *SnRK2* and *ABF* supports robust activation of luciferase signals in the presence of *PYL* receptors. Therefore, we co-expressed SnRK2 and ABFs to evaluate the ABA-responsiveness of PYLs in the *pyl* duodecuple protoplast transient assay.

### Yeast two-hybrid (Y2H) assay
Yeast two-hybrid assays were performed using the Matchmaker GAL4 Two-Hybrid System from Clontech, following the manufacturer's

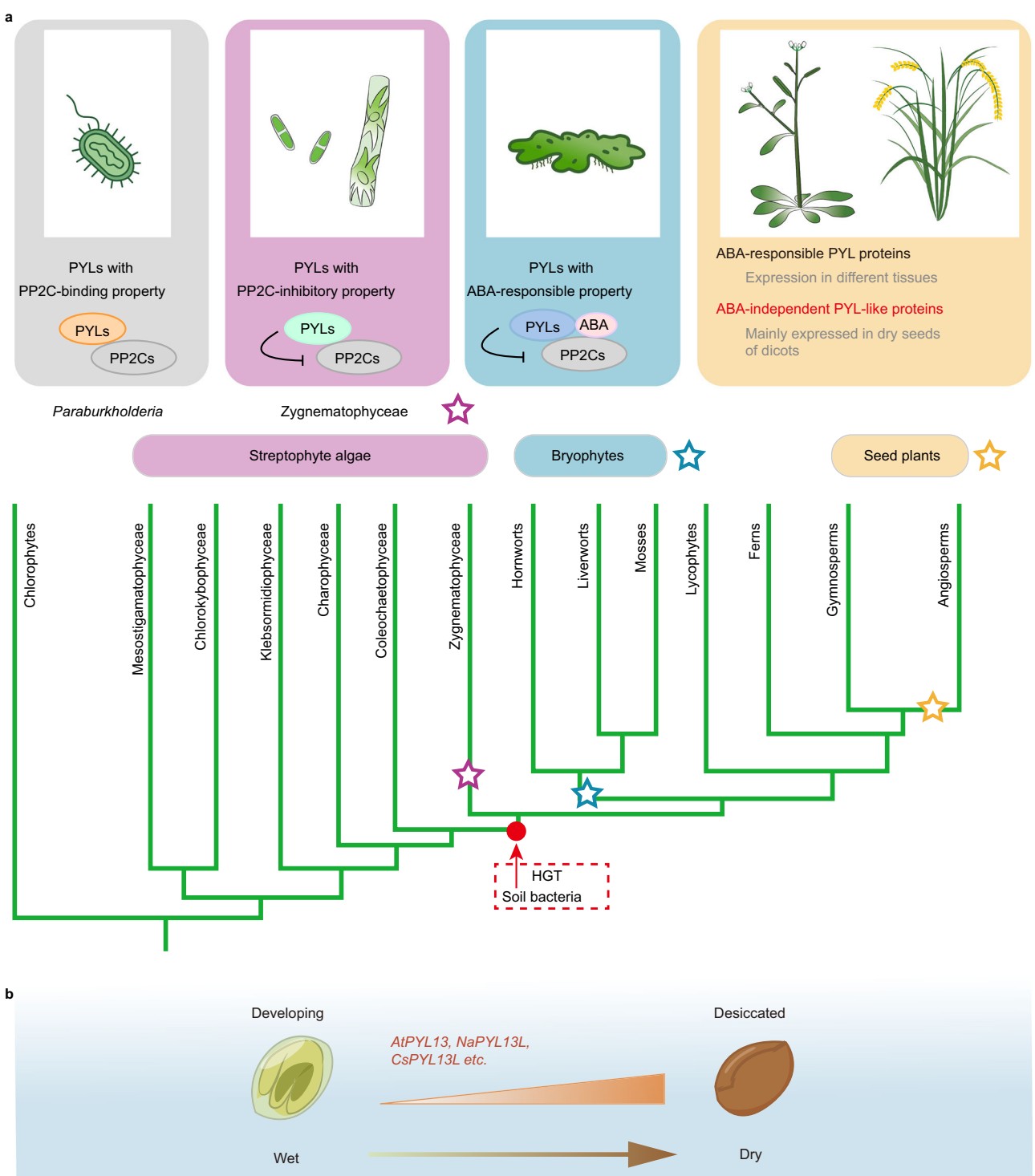

**Fig. 6 | A model illustrating the origination, evolution, and function of ABA-independent PYL-like proteins. a** Proposed stepwise evolution of PYL ABA receptors and ABA-independent PYL-like proteins. The ABA-independent PYL-like proteins originated horizontally from soil bacteria and exist in seeds of land plants. Two invariant lysine and leucine residues are crucial for the ABA responsiveness of PYL ABA receptors. PYL-like proteins gained PP2C-binding ability in some soil bacteria, and the ABA-independent PYL-like proteins obtained PP2C inhibitory activity before the last common ancestor of Zygnematophyceae and land plants.

Although PYLs gained ABA-enhanced PP2C inhibition in bryophytes and evolved ABA-dependent PP2C inhibitory activity in mosses and angiosperms, some ABA-independent PYL-like proteins exist in vascular plants and are mainly expressed in dry seeds in plants such as *Arabidopsis*, *Camelina*, and *Nicotiana attenuata*. **b** Expression pattern of the ABA-independent *PYL*-like genes in dicot seeds. During seed maturation and desiccation, the expression of these ABA-independent *PYL*-like genes is gradually increased, probably functioning in basal stress signaling.

instructions. The yeast strain Y2HGold was co-transformed with paired constructs and then plated on selective dropout nutrition-deficient media containing or lacking 10 μM ABA.

### Firefly luciferase complementation imaging (LCI) assay

The full-length coding sequences of *PrPYL* and *MeenPYL3* were cloned into the *pCAMBIA-35S-nLUC*, while the coding sequences of *ABI1*, *HAI1*, *HAI3*, *HAB2*, and *PP2CA* were amplified by PCR and cloned into the *pCAMBIA-35S-cLUC*. LCI assay was performed by transient expression of paired combinations of constructs in *Nicotiana benthamiana* leaves through Agrobacterium-mediated infiltration. After infiltration, tobacco plants were grown in the dark for one day and under white light for another 2 days. Luciferase activity was detected with a CCD Tanon-5200 by applying firefly D-luciferin (NanoLight).

### Protein expression and purification from *E. coli*

The coding sequences of *PrPYL*, *ZcPYL8*, *MeenPYL3*, *AtPYR1*, *AtPYL9* and *AtPYL13* were amplified and cloned into the pET28a vector to obtain *His-PrPYL*, *His-ZcPYL8*, *His-MeenPYL3*, *His-AtPYR1*, *His-AtPYL9*, and *His-AtPYL13* constructs. All mutants were generated via two-step PCR and verified by plasmid sequencing. The *ABI1*, *PP2CA*, and *HAB1* coding sequences were amplified and cloned into the pGEX-6P-1 vector to obtain the *GST-ABI1*, *GST-PP2CA*, and *GST-HAB1* constructs. *MBP-SnRK2.6* was the same as before[46].

The wild-type or mutated *His-PYLs*, *GST-PP2Cs* and *MBP-SnRK2.6* constructs were transformed into the competent cells of *E. coli* BL21(DE3). Expression of proteins was induced by 1 mM isopropyl β-D-thiogalactoside (IPTG) at 18 °C for 16 h. To purify His-tag-fusion proteins, cells were harvested and sonicated in lysis buffer containing 50 mM NaH$_2$PO$_4$, 300 mM NaCl, pH 8.0 on ice using a Branson SFX250 Sonifier (50% duty, output 5, 2 min, three times). After centrifuging the lysate at 20,000 × *g* at 4 °C for 40 min, the recombinant proteins were purified from the supernatant with the corresponding agarose beads according to the protocols. After washing three times with the washing buffer (50 mM NaH$_2$PO$_4$, 300 mM NaCl, 50 mM imidazole, pH 8.0), the recombinant proteins were eluted with elution buffer containing 50 mM NaH$_2$PO$_4$, 300 mM NaCl, 250 mM imidazole pH 8.0, and frozen at −80 °C. The lysis buffer, wash buffer and elution buffer were altered according to the suggested protocols for MBP- and GST-tagged proteins. It is worth noting that after purification, wild-type or mutated AtPYL13, ZcPYL8, and MeenPYL3 proteins must be used fresh immediately for subsequent assays due to their extreme instability.

### In vitro kinase assays

The purified recombinant wild-type and mutated PYL proteins were incubated with GST-tagged PP2Cs and MBP-SnRK2.6 protein kinase at 30 °C for 2 h in 25 μL of reaction buffer containing 20 mM Tris-HCl, pH 8.0, 1 mM DTT, 5 mM MgCl$_2$, 1 μM ATP and 5 μCi [γ-$^{32}$P]ATP, with or without 100 μM ABA. The reaction was stopped by adding 4× SDS loading buffer, and proteins were separated by a 12.5% SDS-PAGE gel. The phosphorylation signals were detected by a Personal Molecular Imager (Bio-Rad). Coomassie Brilliant Blue (CBB) staining was used as a loading control.

SnRK2 kinase activity was also evaluated using the anti-phospho-S175-SnRK2s antibody[4]. Different combinations of His-PYLs, GST-PP2Cs, and MBP-SnRK2.6 were incubated in 25 μl of reaction buffer (50 mM Tris-HCl, pH 7.5, 20 mM MgCl$_2$, 0.25 mM DTT, 1 μM ATP) for 2 hours. Afterward, samples were boiled with protein SDS loading buffer and separated by a 10%/12.5% gradient SDS-PAGE gel. The anti-phospho-S175-SnRK2s antibody (ABclonal, AP1481, 1:3000 dilution) was used to detect SnRK2 kinase activity. Anti-His antibody (ABclonal, AE086, 1:5000 dilution) was used to detect His-tagged PYL, while anti-MBP antibody (Abmart, M20051M, 1:5000 dilution) was used to quantify MBP-tagged SnRK2.6.

### PYL-mediated PP2C enzyme activity assay

The coding sequences of *PYLs* (*PrPYL*, *ZcPYL8*, *MeenPYL3*, *PYR1*, and *PYL9*) were cloned into *pET28/32a* vectors to generate 6×His-tagged PYLs. The coding sequences of *PP2Cs* (*ABI1* and *PP2CA*) were cloned into *pGEX-6P-1* vector to generate GST-tagged ABI1 and PP2CA. The coding sequence of *HAB1*$^{171-end}$ was cloned into *pMAL* vector to generate MBP-tagged HAB1$^{171-end}$.

Initial reaction velocities for GST-PP2Cs were measured using the synthetic phosphatase substrate pNPP. Reactions contained 0.5 μM PP2Cs (GST-ABI1, GST-PP2CA, or MBP-HAB1$^{171-end}$), with 0, 0.5, 1.0, or 2.0 μM 6×His-PYLs (PrPYL, ZcPYL8, MeenPYL3, PYR1, or PYL9). Reaction buffer consists of 33 mM Tris-OAc (pH 7.9), 66 mM KOAc, 0.1% BSA, 25 mM Mg(OAc)$_2$, and 10 μM (+)-ABA. After 20 minutes, 50 mM pNPP was added to initiate reactions. Hydrolysis of pNPP was monitored at A405 using a SpectraMax® i3x Multi-Mode Microplate Reader (Molecular Devices). PP2C activity in the absence of receptors was defined as 100% activity.

### Structure prediction and alignment

The structures of PrPYL, MeenPYL3, and OsPYL13 with ABA were predicted using alphafold3 (https://alphafoldserver.com/). The ligand (ABA) binding pocket was determined by presenting surrounding residues within a 5 Å radius. The molecular visualization, alignment, and ligand-protein interaction analysis were performed using Pymol (https://pymol.org/2/).

### RNA extraction and quantitative RT-PCR

The total RNA of seeds or seedlings was extracted using FastPure Plant Total RNA Isolation Kit (Polysaccharides& Polyphenolics-rich, Vazyme, RC401-01) according to the manufacturer's protocols. Reverse transcription was carried out using HiScript II Q Select RT SuperMix for qPCR (+gDNA wiper, Vazyme, R233-01). Expression levels were quantified using 2× Universal SYBR Green Fast qPCR Mix (ABclonal, RK21203). The constitutively expressed *AtPP2AA3* or *NaIF5a* was used as an internal inference in *Arabidopsis thaliana* or *Nicotiana attenuate*, respectively.

### Histochemical GUS staining for seeds

GUS reporter transgenic seeds were soaked in water for 15 minutes. The seed coats and endosperm were carefully removed with fine forceps under a stereomicroscope. The embryos were extracted and incubated in a GUS substrate staining solution (O'BioLab, S102) at 37 °C for ~4 hours. Following incubation, the samples were decolorized with 75% and 95% ethanol and water, subsequently at 37 °C and finally imaged by a Stemi 305 stereomicroscope (Zeiss).

### Yeast one-hybrid (Y1H) assay

Yeast one-hybrid assays were conducted using the Yeast One-Hybrid Library Screening System (Clontech), following the manufacturer's instructions. Three tandem copies of the 16nt fragment containing RY or mutated RY motif from the *PYL13* promoter were synthesized and ligated into the pAbAi vector. The RY and mRY reporter yeast strains were generated by homologous integration to Y1HGold and selected by growth on SD/-Ura plates. Subsequently, the minimal inhibitory concentration of Aureobasidin A (AbA) for each bait strain was determined, showing 1 μg/mL for RY or 0.2 μg/mL for mRY reporter yeast strain. The pGADT7-empty, ABI3-B3, and ABI3-B3-KKS plasmids were transformed into the corresponding generated bait-strain competent cells. Dilutions (10$^{-1}$ and 10$^{-2}$) of the same OD liquid suspension of yeast cells were cultured onto the plates and incubated at 30 °C for 2–4 days. The interaction was determined by growth on SD/-Trp/-Leu solid medium with indicated concentrations of AbA.

## Electrophoretic mobility shift assay

The full-length *ABI3* coding sequence with stop codon was amplified and cloned into pET28a, then introduced into *E. coli* BL21(DE3). His-ABI3 protein was expressed and purified following the manufacturer's manual for Ni-NTA Agarose (QIAGEN). The synthetic complementary oligonucleotides of *PYL13* (or *mPYL13* with mutated RY motif) were annealed and cloned into the T-vector to generate the probes. The labeled probes were amplified using Cy5-labeled T7 primer and M13R, while the unlabeled probes were amplified using the normal T7 primer and M13R. The purified proteins were then incubated with the Cy5-labeled probe at room temperature for 20 min in cold EMSA buffer (20 mM Tris-acetate, pH 8.3, 5% glycerol, 0.5 mM EDTA, and 0.05 mg/mL BSA). After incubation, the reaction mixture was electrophoresed on an 8% native polyacrylamide gel with running buffer (40 mM Tris-acetate, pH 8.3) and then imaged with an Amersham Typhoon imager (GE Healthcare). Note that the mutated variants used in the EMSA and Y1H assay differ. Relevant details of oligonucleotides can be checked in the Primer list.

## ChIP-qPCR assays

ChIP assays were performed using 2 g seeds (about 15 days after flowering) from Col-0 and *OE-ABI3-Flag*. Samples were fixed with 1% formaldehyde. Nuclei were isolated using isolation buffer (1 M sucrose, 10 mM HEPES pH 8.0, 5 mM MgCl$_2$, 5 mM KCl, 0.6% Triton X-100, 0.4 mM PMSF, protease inhibitor cocktail), filtered through Miracloth, washed with CHIP buffer (0.25 M sucrose, 10 mM Tris-HCl pH 8.0, 10 mM MgCl$_2$, 1% Triton X-100, 1 mM EDTA, 5 mM β-mercaptoethanol, protease inhibitor cocktail), and pelleted by centrifugation. Nuclei were resuspended in 300 μl lysis buffer (50 mM HEPES pH 7.5, 150 mM NaCl, 1 mM EDTA, 1% SDS, 0.1% Sodium Deoxycholate, 1% Triton X-100, protease inhibitor cocktail) and sonicated with a Bioruptor (Diagenode). The nuclei lysates were incubated overnight with anti-Flag antibody (Sigma-Aldrich, #F1804, 1:100 dilution), followed by 2-h incubation with Protein G beads (Invitrogen). The precipitated protein–DNA mixtures were washed with low salt buffer (20 mM Tris-HCl pH 8.0, 150 mM NaCl, 1% SDS, 1% Triton X-100, 2 mM EDTA), high salt buffer (20 mM Tris-HCl pH 8.0, 500 mM NaCl, 1% SDS, 1% Triton X-100, 2 mM EDTA), LiCl buffer (10 mM Tris-HCl pH 8.0, 250 mM LiCl, 1% NP-40, 1% Sodium Deoxycholate, 1 mM EDTA) and TE buffer (10 mM Tris-HCl pH 8.0, 1 mM EDTA). DNA was eluted from beads with elution buffer (1% SDS, 0.1 M NaHCO$_3$) at 65 °C. The DNA was recovered after reverse cross-linking and proteinase K treatment. Enriched DNA fragments were analyzed by qPCR using the indicated primers (Table S1), with *TUB8* serving as a negative control.

## ITC assays

ITC experiments were performed with an ITC200 Microcalorimeter (MicroCal). All the ligands and proteins were in the same reaction buffer consisting of 20 mM HEPES (pH 7.5) and 150 mM NaCl. The ligand 1 mM (+)-ABA was titrated against 50 μM PrPYL, MeenPYL3, PYL9 or PYL9-K63Q/L89F proteins. The ligand 2 mM (+)-ABA was titrated against 10 μM OsPYL12 proteins. The stoichiometry between the ligands and proteins was set to 1 for all analyses. Data fitting was performed by using the ORIGINTM software supplied with the instrument. Thermodynamic constants were determined by Origin software.

## Statistical analysis

The ImageJ software was used to analyze the relative band intensity during in vitro kinase assays. Statistical analysis was examined using Student's *t* test or two-way ANOVA with Tukey's multiple comparisons test using the GraphPad Prism9 software.

## Reporting summary

Further information on research design is available in the Nature Portfolio Reporting Summary linked to this article.

## Data availability

All data supporting the findings of this study are available in the main text or the supplementary files. The biological materials are available from the corresponding author upon request. Source data are provided with this paper. Genes mentioned in this study can be found in the TAIR database (https://www.arabidopsis.org/) or NCBI Database (https://www.ncbi.nlm.nih.gov/) under the following accession numbers: *Paraburkholderia* PYL (PrPYL), WP_102636708.1, *Paraburkholderia rhynchosiae*; *Frankia* PYL, WP_050795676.1, *Frankia sp. EUN1f*; *Pseudomonas* PYL, WP_096480507.1, *Pseudomonas frederiksbergensis*; *Sphingomonas* PYL, WP_081663793.1, *Sphingomonas sp. YL-JM2C*; AlPYRL8, XP_020877794, *A. lyrata*; AlPYRL11, XP_002870032, *A. lyrata*; CaPYL13L1, PHT90407, *Capsicum annuum*; CaPYL13L2, XP_016558326, *Capsicum annuum*; CbPYL13L, PHT30423, *Capsicum baccatum*; CrPYL13L, XP_006285406, *Capsella rubella*; CsPYL13L1, XP_010449490, *Camelina sativa* (false flax); CsPYL13L2, XP_010434508, *Camelina sativa* (false flax); HvPYL13L1, HORVU3Hr1G040680.1, *Hordeum vulgare* L.; HvPYL13L2, HORVU3Hr1G030210.4, *Hordeum vulgare* L.; NaPYL13L, XP_019234962, *Nicotiana attenuata*; NaPYL4L, NIATv7_g37793, *Nicotiana attenuata*; NaPYL8L, NIATv7_g17046, *Nicotiana attenuate*; NsPYL13L, XP_009802216, *Nicotiana sylvestris*; NtPYL13L, XP_009595006, *Nicotiana tomentosiformis*; ObPYL12, XP_006648510.1, *Oryza brachyantha*; OsPYL12, XP_015624089, *Oryza sativa*. Source data are provided with this paper.

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

## Acknowledgements

We are grateful to Professor Guanzhu Han and Dr. Zhen Gong for their kind assistance and insightful advice from the evolutionary perspective. We thank Professor Chengbin Xiang and Dr. Pingxia Zhao for generously providing the *ABI3-Flag* seeds. We are grateful to Professor Dapeng Li for kindly gifting the *Nicotiana attenuata* seeds. We thank members of the Zhao Lab for helpful discussions. This work was supported by the Biological Breeding-National Science and Technology Major Project (2023ZD040710202 to Y.Z.), the Project of Stable Support for Youth Teams in Basic Research of the Chinese Academy of Sciences (YSBR-119 to Y.Z.), Strategic Priority Research Program of the Chinese Academy of Sciences (Grant no. XDB0630201 to Y.Z.), the Shanghai Center for Plant Stress Biology from the Chinese Academy of Sciences (to Y.Z.), and Beijing Frontier Research Center for Biological Structure, Tsinghua University, Beijing, China (to W.L.).

## Author contributions

Y.Z. conceived and designed the research. T.L., Q.L., T.H., W.L., Y.L., and H.H. performed the experiments. T.L., Q.L., T.H., W.L., Y.L., H.H., and Y.Z. analyzed the results. Y.Z. wrote the manuscript.

## Competing interests

The authors declare no competing interests.
