## [Transparent Peer Review file · Nature Communications]

ABA-independent PP2C-binding in PYLs traces to bacterial origins and persists in land plants

Corresponding Author: Professor Yang Zhao

Version 0:

Reviewer comments:

Reviewer #1

(Remarks to the Author)

This study presents an intriguing hypothesis regarding the evolutionary origins of PYL genes, proposing to provide experimental evidence that suggests a bacterial PYL homolog that was transmitted to the plant lineage. However, the central claims of the paper, particularly as stated in the title, are not adequately supported by the presented data. Most of the experimental work essentially reproduces findings from previous studies rather than providing novel insights. Furthermore, the manuscript suffers from substantial clarity issues, with unsupported or misinterpreted claims in the introduction and a presentation of results that obfuscates key conclusions (see specific examples below).

Major Concerns

Functional Characterization of a Bacterial PYL Homolog

A central premise of this study is the identification of a bacterial PYL homolog with ABA-independent activity. However, the assays used fail to assess this function rigorously. The experimental systems either contain endogenous ABA (as in plant-based systems) or rely on overexpression, which can artificially drive protein-protein interactions independent of ligand binding. Notably, even well-characterized ABA-dependent receptors, such as PYL4, exhibit ABA-independent PP2C binding under similar conditions. A more appropriate *in vitro* approach, where protein concentrations are strictly controlled and ABA is entirely absent, is required to definitively assess ligand dependency.

Horizontal Gene Transfer Hypothesis Lacks Supporting Evidence

The claim of horizontal gene transfer (HGT) is presented without substantive support. Establishing HGT in a system that diverged over 600 million years ago is inherently difficult. Yet, the authors fail to explore/roll out alternative evolutionary scenarios, such as convergent evolution, which remain equally plausible. A robust phylogenetic analysis should provide statistical confidence measures and exclude other evolutionary possibilities. If the phylogenetic tree showed plant sequences embedded within bacterial lineages, it could suggest recent gene transfer, but the current analysis lacks the necessary rigor. The arbitrary nature of the sequence relationships as presented does not substantiate the proposed HGT event. While the reviewer is not a bioinformatician, the absence of statistical validation and failure to test alternative hypotheses significantly weaken this claim.

Specific examples:

In the title "...in the seeds of land plants"

The relations described in the study extend beyond seed plants, as the PYL gene family is conserved across the entire lineage of land plants, not just in seed-bearing species. Also the title should explicitly state which functional aspect of PYL receptors has been retained throughout evolution—specifically, their ability to bind PP2C phosphatases.

Furthermore, the claim of horizontal gene transfer (HGT) remains a hypothesis rather than a demonstrated/conclusion. As such, it should either be removed from the title or clearly framed as a hypothesis rather than a definitive statement. The current phrasing in the title risks misleading readers by presenting an unverified assumption as an established finding.

-Abstract: Has inaccuracies

Examples:

Authors claim: "which requires constitutive activation of stress signaling, was mainly adopted by early land plants and persists in seeds."

This statement is incorrect—ABA levels are crucial for seed dormancy and decrease to allow germination. During dormancy, ABA levels remain high to prevent premature germination. However, for germination to occur, ABA levels must decline (while gibberellins increase).

Authors claim: "Through evolution, the PYR/PYL/RCAR family (PYL) of ABA receptors gained several functions over the course of evolution, namely inhibition of protein phosphatases (PP2Cs), ABA-enhanced inhibition of PP2Cs, and ABA

dependent inhibition of PP2Cs”

The only well-established physiological function of PYL receptors is their ability to inhibit PP2C phosphatases, which are central to ABA signaling. If the authors intended to suggest an additional function, they should clarify it and provide supporting evidence.

Authors claim: “However, it remains unclear how the ABA-irresponsible PYLs originated and whether they exist in land plant.”

This statement is incorrect—it has been shown that PYL receptors in land plants likely originated from algal ancestors, and homologous proteins are present across all land plant lineages. This suggests that the PYL family is not exclusive to land plants but instead evolved from pre-existing proteins in algae before being co-opted into the ABA signaling pathway in terrestrial plants. See doi.org/10.1073/pnas.1914480116

Authors claim: “AtPYL13 and AtPYL13-like genes were highly expressed in dicot seeds, suggesting the transformation from ABA receptors to ABA-irresponsible PYLs after expansion of the gene family”.

This statement misrepresents the actual expression profile of PYL in Arabidopsis seeds. While PYL13 expression can be detected, public datasets (e.g., BAR ePlant) indicate that its expression is approximately 100-fold lower than PYR1, suggesting it plays a minor role in seed germination. Additionally, experimental evidence challenges the significance of PYL13 in this context. For example, activating PYR1 alone, using a specific agonist, was sufficient to inhibit germination. Dimeric multiple mutants exhibit an early germination phenotype, further suggesting that the primary regulators of germination are other PYL receptors. See Park 2009 DOI: [10.1126/science.1173041](https://doi.org/10.1126/science.1173041) and Okamoto 2011 <https://doi.org/10.1073/pnas.1305919110>

Given these findings, if PYL13 has any role in germination, it is likely minimal or redundant, and its functional relevance in this process remains questionable.

-Introduction- Has inaccuracies

Some Examples:

Authors claim: “Terrestrialization is a pivotal event in the evolution of green plants, which requires mechanisms to survive under drought stress.”

The term “drought stress” is not appropriate to describe the primary challenges of terrestrialization (refers to the evolutionary transition from an aquatic to a terrestrial environment). Terrestrialization involves adaptation to a gradual shift from consistently wet conditions to environments with fluctuating water availability rather than exposure to extreme drought. Using “drought stress” oversimplifies the complexity of this transition, which encompasses not only water limitation but also factors such as desiccation tolerance, gas exchange regulation, light intensity, and structural adaptations to terrestrial life.

Authors claim: “Plants mainly utilize two strategies to survive under water deficiency: “drying without dying” or “without drying”.”

These terms are not clear.

Authors claim: “The early land plants are mainly poikilohydric and adopt the “drying without dying” strategy, which requires constitutive activation of stress signaling and enables plants to get through extreme conditions” –

Early land plants possess mechanisms for water regulation and stress adaptation that are independent of vascular systems, as described in the cited paper. A more relevant and evolutionarily conserved mechanism is osmoregulation, which is shared between mosses and seed plants.

Regarding the statement “PP2Cs can be ABA-dependent or ABA-enhanced”, this terminology is unclear. PP2C inhibition occurs through both ABA-dependent and ABA-independent mechanisms, and the distinction between “ABA-enhanced” and “ABA-dependent” needs to be explicitly defined to avoid confusion.

Authors claim: “The entirely ABA dependent dimeric PYLs emerged in angiosperms and have nearly no basal activity”-

This applies to other PYLs as well— not just dimeric forms. This should be considered as it's not the only mechanism of low basal activity (in Arabidopsis and other species), as indicated in DOI: [10.1016/j.cub.2024.12.043](https://doi.org/10.1016/j.cub.2024.12.043).”

Authors claim: “which enables angiosperms to have a wider range of ABA responses and improved balance between growth and stress responses”-

This is an assumption and should be stated as such.

Authors claim: “However, it remains unclear how the ABA-irresponsible PYLs originated.”

The focus should be more specific—there is evidence that land plant lineages and the common ancestor of *Zygnema circumcarinatum* possessed a nonresponsive PYL. What remains unclear is how this protein was ‘adopted’ into the plant lineage.”

-Specific points in the result section:

The section titled: “The bacterial PYL-homolog from *Paraburkholderia rhynchosiae* possesses PP2C-binding ability”

Authors claim: “It is proposed that the ancestral PYLs in plants were gained through horizontal gene transfer from soil bacteria based on phylogenetic analyses (Fig. 1a).” –

The phylogenetic analysis does not support horizontal gene transfer occurring over 600 million years ago.

Authors claim: “Despite lacking ABA responsiveness”

This should be tested in vitro using assays that directly assess the involvement of ABA as a ligand, such as a PP2C inhibition assay, isothermal titration calorimetry, or any equivalent assay.

The authors identified the interaction between bacterial PrPYL and PP2C via a yeast two-hybrid assay. However, this interaction was not reproducible in planta nor when SnRK2 competed for the PP2C interface in an in vitro assay. Based on this, the authors concluded that there is an interaction with PP2C but no PP2C inhibition. As stated, ‘These results suggested that the bacterial PrPYL possesses PP2C-binding but not PP2C inhibitory activity.’ This conclusion is flawed for two reasons: 1. The interaction between PYL and PP2C typically blocks the active site of the PP2C enzyme. Therefore, a

'canonical interaction' should, by default, result also in inhibition. 2. To properly claim this, the interaction and PP2C activity must be evaluated through specific assays, not as a default by a negative result. At this point, it is an unsubstantiated guess. An alternative explanation could be that, in yeast, the protein steady-state concentration is high, allowing for the monitoring of low-affinity interactions. In vitro and in-planta, however, are more challenging, as other proteins could compete for the PP2C interface. Consequently, it is unsurprising that the weak interaction with bacterial PrPYL is outcompeted by SnRK2s (whether from plants or purified protein), resulting in the absence of an interaction signal."

The section titled "ABA-irresponsive PYLs exhibit PP2C inhibitory activity in Zygnematales".

Sun et al. (2019) showed that algal PYL interacts with PP2C in an ABA-independent manner. In that study, Sun also demonstrated that ZcPYL8 cannot bind ABA (by isothermal calorimetry titration). In this current study, the authors aimed to test additional algal PYLs. They used an Arabidopsis PYL mutant with endogenous ABA levels, which were unsuitable for testing ABA-independent activity. The SnRK activity (following release) assay is sensitive to PYLs with basal ABA-independent activity but cannot distinguish if there is a mild ABA effect. These two assays do not support the claim that the 'PYL homologs gained ABA-independent PP2C inhibitory activity during the evolution of Zygnematales.' In vitro assays that directly test ABA binding would be better suited to scratch the surface of this hypothesis. I suggest starting by following the biochemical assays performed by Sun in 2019.

The section titled "An ABA-irresponsive PYL exists in Arabidopsis"

The authors claimed that "the wild-type AtPYL13 could not activate RD29B-LUC with ABA treatment, while the expression of AtPYL13-204 Q38K/F71L restored the ABA-dependent activation of RD29B-LUC in the pyl duodecuple mutant protoplasts (Fig. 3c), indicating that AtPYL13 is otherwise functional and that the two natural sequence variations in the CL1/CL2 domains of AtPYL13 confer ABA irresponsiveness"

A paper from the Grill group addresses PYL13 activity and the specific mutation analyzed here:

<https://doi.org/10.1073/pnas.1322085111>. These mutations were thoroughly tested in that study. I failed to see the added value of repeating them here."

The sections titled "Two invariant residues are crucial for PYL ABA receptors.", "ABA-irresponsive PYLs exist in seed plants" and "ABI3 mediates seed-specific expression of AtPYL13"

I don't understand the rationale behind the ectopic expression of impaired-active mutated PYLs. What can be learned from these experiments? This entire section seems unrelated to the main focus of the study. Furthermore, it is well-established that dimeric PYLs are key players in seed germination biology. This has been tested through agonist activation of specific PYLs and by examining expression levels (actual reads versus the relative expression tested here). In fact, PYL13 expression is barely detectable."

Reviewer #2

(Remarks to the Author)

Lu et al. report the evolutionary trajectory of PYL ABA receptors. Based on phylogenetic and functional analyses, they found that an ABA-irresponsive PYL in soil bacteria was horizontally transferred from soil bacteria, gained PP2C interaction in *Paraburkholderia rhynchosiae*, became capable of inhibiting PP2C in the common ancestor of Zygnematales and land plants, evolved into ABA receptors in liverworts, and then diversified to possess various affinities to ABA, including an irresponsive receptor exclusively expressed in seeds.

The presented work attracts a wide range of biologists who study protein evolution and hormone signaling. The work is comprehensive and based on solid functional analysis. The following are my comments to improve the manuscript:

[1] The title needs to be reconsidered:

(1) "Horizontally transferred" typically refers to genes, not proteins.

(2) I would think that the ABA-irresponsive receptors that was horizontally transferred from soil bacteria and retained functions in seeds are qualitatively distinct. PrPYL is a prototype of ABA receptor, while AtPYL13 is a variant of ABA receptor. The current title sounds continuous ABA-irresponsive receptors.

(3) Related to this, the overall context appears to have two distinct parts. I recommend the authors revise the text to create a cohesive narrative.

[2] Co-evolution of PYL and PP2C:

The authors examined the function of ancient PYLs and their interaction with Arabidopsis PP2Cs. The evolution of PYLs during this period heavily depends on the nature of PP2C. The evolution/conservation of required amino acid residues in PP2Cs should be discussed. Ideally, PP2C residues required for PYL interaction or inhibition can be separately discussed. If possible. This is a core part of the story.

[3] ABA-irresponsive PYLs:

The functionality of ABA receptors or PYLs can be assigned by their ability to inhibit PP2C and their ABA dependency, regardless of whether they act as monomers or dimers. ABA receptors evolved as PYLs with ABA-enhanced PP2C inhibition. The present work demonstrated that ABA-irresponsive PYLs constitutively inhibit some PP2Cs, but their PP2C inhibition activity is not quantitatively compared with other receptors. The ability to induce downstream events (or inhibit

PP2C activity) among PYL members is a critical aspect of their functionality, as they function in the same cells with other ABA-responsive receptors *in vivo*. This makes it difficult for readers to assess their role *in vivo*. The phenotype was primarily assessed by overexpressors. *In vitro* SnRK2 activation was quantified, but this quantification does not provide the degree of strength among family members. The authors discussed, "Although the *pyl* duodecuple mutant is viable and fertile, we cannot get any seeds for the *pyl* quattuordecuple mutant that has further mutated AtPYL6 and AtPYL13, supporting the critical role of AtPYL13 in fertility and seed maturation." The multigenic mutant phenotypes are the most critical data to assess the role of AtPYL13 among family members, especially in seeds where it is exclusively expressed. I encourage the authors to describe the multigenic mutant phenotypes in more detail (I suggest that the overexpressor phenotype be included in the supplemental materials).

Related to the above, I am curious if the two critical residues (Q and F) in ABA-irrespective PYLs are only characteristic conservations of PYL13-like proteins.

Reviewer #3

(Remarks to the Author)

The manuscript by Lu et al. reported an evolutionary model of ABA-irrespective PYL-like proteins and proposed their roles in desiccation tolerance of angiosperms seeds. They demonstrated *in vitro* that a bacterial PYL-like protein can interact with but not inhibit the activity of Arabidopsis ABI1 in an ABA-independent manner, supporting a previous idea that plant PYL family is originated from soil bacteria. They also performed the same experiment set against PYL-like genes from Zygnematomyphyceae, showing that algal PYLs have the binding and inhibitory function against Arabidopsis PP2Cs in an ABA-independent manner. They focused on two important amino acids located at conserved loops 1 and 2 which are not conserved in ABA-irrespective PYLs. The two amino acids were also not conserved in seed-specific ABA-irrespective AtPYL13 and substitution of these amino acids to that of ABA-responsive PYLs were sufficient to confer ABA-responsiveness. They further proposed that AtPYL13 was under the regulation of seed-specific ABI3 transcription factor, which is essential for the acquisition of seed desiccation tolerance.

It is a potentially interesting research paper to give insights into the evolution of PYLs and plant desiccation tolerance; however, the experimental evidence was not sufficient support their claims. Moreover, I recommend the authors should discuss with an expert of evolutionarily research since the manuscript contains many flaws in evolutionary aspect. For example, "PYLs then gained ABA-binding and -responsive activities in liverworts." Liverworts are extant plant group and not ancestor of land plants. My major concern is that there is no *in vivo* experimental data showing that PYL13 is directly regulated by ABI3 and plays an essential role like ABI3 in seed desiccation tolerance. I was also confused about the "ABA-irrespective" PYLs. What are the criteria for this word? For example, MeenPYL1/2 have no capacity but MeenPYL3 does has the capacity to activate RD29B-LUC in *pyl* protoplasts with or without ABA. These are all ABA-irrespective PYLs? Probably it should be, since the authors describe AtPYL13 as ABA-irrespective, which cannot activate RD29B-LUC in *pyl* protoplasts like MeenPYL1/2. Is ABA-irrespective PYL re-evolved in angiosperms? What are the differences between ABA-irrespective algal PYLs and ABA-irrespective angiosperms PYLs.

Comments

L88- and Fig.1a. What is the reason to add SnRK2 and ABF in the *pyl* protoplast transient assay? Was introduction of AtPYL(s) not sufficient to recover ABA response?

L96- This experiment should include a positive control of ABA-dependent PYL.

L121- and Fig. 1k. This is also an extant bacterium and not a doner of PYL, therefor, misleading. The authors did not show that the target of bacterial PYLs is PP2Cs, and Fig. 1k should be deleted or revised.

L122- The Figure layout should be arranged according to the order in the main text. There is no mention about Fig. 2a before Fig.2b.

L155- Fig. 2e,f need quantification.

L162- Is this Ser the same with Ser112 described in L115? The numbers do not match each other.

L178- Reduction of high basal activity should be supported by an appropriate statistical analysis.

L180-182 The description is not clear.

L181- What does "original land plants" mean?

L184- "Zygnematales" are extant plants.

L185- I mentioned above.

L186 and L196 The authors should describe the PYLs subfamilies with their function in the introduction.

L191- There is no data for AtPYL7. What is the criterion of "basal activity"? Is there any significant difference between

AtPYR1 and AtPYL9 autoactivation?

L193- Please describe more about the “stepwise transformation”. For me, it is not evident from the figures. It will be helpful to indicate where transformation occurred on the phylogenetic tree.

L213- Which is not evident from Fig. 3c, as the authors described. Please explain why AtPYL13 failed to activate RD29B promoter. Is PP2CA seed-specific factor?

L217- No PYR1 data in supplementary Fig.3b.

L234-236 Please explain what does this result indicate or suggest.

L249-250 Please clearly explain which are the putative ABA-irresponsive PYLs in Supplementary Data 1.

L280- Why the authors used OsPYL13 for structure prediction? There is no functional data for OsPYL13 (Fig. 4c).

L309-310 Please explain why are these mutations required in this experiment briefly.

Reviewer #4

(Remarks to the Author)

Abcisic Acid (ABA) is arguably one of the most important biomolecules on the planet, as a consequence of its central importance in promoting plant dehydration tolerance: a state essential to the initial plant colonization of land ca. 500Ma that had planetary consequences: the photosynthetic activity of earliest land plants resulted in elevation of the oxygen content of the atmosphere to a level enabling the evolution of large land animals.

ABA, a small molecule derived from the breakdown of carotenoids and ubiquitous in Nature, acts as a ligand to the multimember family of ABA receptors (“PYLs”). These receptors are START-domain proteins that are structurally modified by ABA-binding so that they form molecular complexes with a sub-family of PP2C protein phosphatases blocking their dephosphorylation activity. This activates SnRK2 protein kinases that direct transcription factors to drive the expression of genes encoding a plethora of protective gene products that enables plants to survive dehydration.

This manuscript describes the functional evolution of the PYL receptor family that is thought to have been acquired by horizontal gene transfer from microbe(s) to an ancestral alga from which all land plants subsequently diverged and proliferated.

The paper comprises:

1. Examination of receptor-like functions of bacterial PYL homologs:

(i) Does a bacterial PYL-like gene complement the Arabidopsis 12x pyl- line?

- No: there is no activation of the Rd29b reporter gene either with or without added ABA

(ii) Does a bacterial PYL-like gene bind to an Arabidopsis PP2C

- Yes, but only one of the bacterial proteins tested showed binding in Yeast-2-Hybrid and split luciferase assays.

(Paraburkholderia rhynchosiae “Pr”)

(iii) Does the PrPYL activate SnRK2 phosphorylation in vitro?

- No. The bacteria tested were those identified in Reference 11 (which focused on algal progenitors in land plant evolution, but indicated these species to be taxonomically related potential HGT progenitors of plant PYL genes). Of these Paraburkholderia remains the leading contender.

(iv) Can structural modeling indicate whether PrPYL could interact with a plant PP2C protein.

- Up to a point: there are similarities between the modeled PrPYL structure and that of the crystallographically determined Arabidopsis PYL-ABA-PP2C co-complex, but the predictions indicate structural inconsistencies that would preclude ABA-interaction (not least, the lack of a “gate” motif in the bacterial sequence)

I think that these essentially negative findings are not unexpected, and one would not expect PYL homolog within an extant bacterium that has been independently evolving for the last 500 million years to necessarily be similar to whichever microbial sequence actually might have been passed into the genome of the algal ancestor of today’s land plants. But it is important that this has now been experimentally tested. While comparative genomics has been a powerful tool in developing our understanding of land plant evolution, there will always be some uncertainty in making accurate predictions.

2. Do algal PYL homologs within the Zygnematales have properties similar to angiosperm PYLs in activating ABA-mediated gene expression?

- Yes, but not all the properties of PYLs in modern land plants. PYL function has now been analysed extensively in angiosperms, and also to a lesser extent in early-diverging land plants (Bryophytes: Ref 12 and in a very recently published paper (Zimran et al. 2025. Current Biology 24: 818-830).

The results reported here showing ABA-independent “basal” activity in PYL-PP2C interactions are consistent with these recent papers. Whereas these two papers use direct analysis of PP2C inhibition in vitro, this manuscript nicely reinforces the evidence by showing PYL-binding-specific activation of SnRK2 kinase activity being promoted in an ABA-independent manner.

I suggest that the title of this section be altered to “Zygnematales PYLs exhibit PP2C inhibitory activity in an ABA-independent manner” as the current wording implies that the experiments had all been carried out in algae, rather than analysing the activity of PYLs from algae.

3. Characterizing ABA-independent PYL activity in Arabidopsis

- ABA-independent basal activity has been associated with PYL family members that are monomeric, with the dimeric angiosperm subfamily 3 PYLs being highly ABA-dependent. The authors have carried out complementation of the 12x pyl-mutant to identify Arabidopsis PYLs with basal activity and those without, confirming this.

They also state in Lines 192-194: “Phylogenetic analysis of PYLs from the Brassicaceae family exhibited the stepwise transformation from ABA receptors to ABA-irresponsible PYLs after expansion of the gene family”.

I do not follow this argument. The support for this is supposedly in Supplementary Figure 3a and Supplementary Data 1 but I cannot follow the logic (This may be my fault). I should have expected basal activity alone (ABA-independence) to be the ancestral state with ABA-dependence with some residual ABA-independence to follow and complete ABA dependence to be the latest acquisition. (Essentially this is what is described in the abstract in lines 17-19). I think the authors need to clarify this.

4. Do specific mutations identify residues required for ABA-dependence?

- The residues Q38 and F71 in PYL13 are identified as variant and required for ABA binding, and the PYL13 gene is characterized by basal activity rather than ABA-dependence using site-specific mutations to correct this, and to examine the requirement for Lysine and Leucine at the corresponding positions in other PYL genes.

This then leads on to analysis of PYLs in other plants that carry similar mutations, and that also exhibit principally basal activity in their regulation of SnRK2 phosphorylation.

5. Seed specificity of PYLs

- A major outcome of PYL activation of the ABA response is in the regulation of seed development. The control of ABA-dependent acquisition of desiccation tolerance is one of the most important functions for plant reproduction, enabling the development of resilient reproductive propagules that can be dispersed in space, and as a result of ABA-induced dormancy, dispersed also in time. This property is also of fundamental importance for human social evolution, as the ability to store dry seeds from one year to the next is the basis of agriculture and transformed human societies from hunter-gathering tribes to settled communities leading to the advent of civilisation.

- The authors note that a number of PYLs are specifically expressed during seed development and are thus direct activators of the genetic programme that leads to desiccation tolerance in seeds. They additionally note that AtPYL13 is one of these seed specific PYLs, and that it (and likely the others) is expressed exclusively during this time, the transcription of the gene being directed by the ABI3 transcription factor – itself generally described as being seed-specific in its functions (actually, this is not quite correct: it is also active in bud dormancy).

- However, it is not clear to what extent the AtPYL13 is essential for this process, since seed desiccation tolerance is certainly principally ABA-dependent, and thus necessarily principally reliant on those other seed-expressed PYLs named in this section of the manuscript. I do not imagine that the basal activity of AtPYL13 contributes significantly, by comparison.

- This last section, while interesting, seems somewhat out of place by comparison with the foregoing information, and it might be better published separately in another journal.

Recommendations

This is clearly a well-executed piece of work, but the authors should consider my comments above. I also have a few specific recommendations:

1. The title et seq.

First, I do not think that the term “ABA-irresponsible” is appropriate as a description of the PYL receptors. First, no PYL is either “responsive” or “not responsive” to ABA. PYLs are receptors that become active through interacting with ABA as a ligand in order to enable ABA responsive gene expression.

It is the phosphatase activity and the subsequent pathway of gene activation that is responsive – not the PYL. In the case of PYL homologs that bind to PP2CA phosphatases without binding ABA, - defined as “basal activity” of these PYLs, it remains to be determined how much of a response is elicited.

For example:

Lines 82-83: “we evaluated the ABA responsiveness of PYLs in the Arabidopsis ABA-insensitive pyl duodecuple mutant”

- Actually, this isn't testing the “ABA responsiveness” of the PYL, it's measuring the ABA responsiveness of the Rd29b-Luc reporter when the 12x mutant is complemented by wild-type PYLs from Arabidopsis, Marchantia and Physcomitrella, and so-called bacterial PYLs (or more accurately, genes identified as having some similarity to bona fide PYLs

I recommend the use of ABA-independent, rather than-ABA insensitive, in the title:
“ABA-independent PYL-like proteins...”

...and throughout the manuscript.

2. Minor corrections

Lines 46-7 and 57-8: Use a standard font, not bold text. The use of bold text to emphasise a point is acceptable in a grant proposal, but not in a paper (unless it is a section heading).

Line 49: “encoded by the algae Zygnema...” to: “encoded by the alga Zygnema...”

Lines 192-194: “Phylogenetic analysis of PYLs from the Brassicaceae family exhibited the stepwise transformation from ABA receptors to ABA-insensitive PYLs after expansion of the gene family”

This implies that an ABA receptor with low basal activity arose first, and subsequently PYLs evolved with high basal activity. I'd have thought that basal activity was the ancestral state. Of course, this is being described in a relatively late-evolving plant group, in which there could have been prior loss of members with basal activity prior to the re-emergence of PYLs with high basal activity.

Supplementary dataset 1: Phylogenetic analysis of PYLs.

- The authors have fallen in to the trap of failing to determine whether genes annotated as ABA receptors are indeed bona fide receptors. Thus the set of PYL homologs identified for *P. patens* identifies three gene models that were identified as encoding ABA receptors in a report by Sussmilch et al. (2017) *Plant Signaling & Behavior* 12: e1365210, but that are actually non-functional pseudogenes. These are Pp3c3_660, Pp3c7_4410 and Pp3c15_19950. The first two of these are predicted to contain a short intron that is unsupported by RNAseq data. In each case the predicted intron (automatically assigned by the software used to generate the models) contains an in-frame termination codon that would render an unspliced transcript non-functional. In 3_660, the available RNAseq data for this locus do not indicate the presence of an intron. For 7_4410 the gene model is unsupported by any RNAseq reads, it lies in an intergenic region and there is no evidence of a start codon. For 15_19950, the predicted polypeptide sequence shows the N-terminus to lie only 14 residues before the SGIPAT “gate” sequence: however, a BLASTX analysis of the predicted 5'UTR sequence immediately prior to the candidate N-terminal methionine residue shows a more likely N-terminal coding region with significant similarity to that of PpPYL1 (Pp3c26_15240). Readthrough shows that these two sequences are out of frame, with two consecutive termination codons between the “UTR” and “coding” sequences. These are likely degenerated pseudogenes and should be excluded.

Similarly, the *M. polymorpha* model Mapoly0145s0004 is speculative, as there is very little RNAseq support for this locus.

For *Sphagnum fallax*, The sequence assembly used (based on scaffolds, rather than chromosomes) is not the most recent: the V1.1 locus identifiers are as follows:

V0.5 Sphfalx0046s0087 = V1.1 Sphfalx16G013800
V0.5 Sphfalx0031s0007 = V1.1 Sphfalx16G062700
V0.5 Sphfalx0002s0105 = V1.1 Sphfalx19G066800
V0.5 Sphfalx0024s0170 = V1.1 Sphfalx11G055600
V0.5 Sphfalx0310s0087 = V1.1 Sphfalx04G53400

Version 1:

Reviewer comments:

Reviewer #2

(Remarks to the Author)

The authors have revised their manuscript appropriately, and the revised version is much clearer in terms of wording and the description of the evolution of ABA receptors.

One point I would like the authors to further improve concerns the function of AtPYL13. Based on the reduced seed production of the quattuordecuple mutant (generated by introducing the *pyl13* and *pyl6* mutations into the duodecuple *pyl* mutant), the authors argue for a role of AtPYL13 in fertility and seed maturation. However, I find this evidence insufficient to support their claim for the following reasons. First, this observation does not provide direct insight into the specific role of AtPYL13 alone. Second, it does not clarify the function of AtPYL13 in seed maturation. Since the central argument of this work is that AtPYL13 plays a role in seed maturation (particularly in the desiccation stage), it is critical to discuss AtPYL13 function in this specific developmental context.

I note that mutations in seed maturation-specific genes result in seeds with desiccation intolerance or distinct morphological defects. In contrast, phenotypes related to fertility usually leads to the absence of embryonic/zygotic structures in seeds, or

as aberrant segregation ratios. If the reduced seed phenotype observed here is due to sterility (either male or female), it does not support the authors' main claim regarding a role for AtPYL13 in seed maturation.

I encourage the authors to provide clear phenotypic evidence supporting the function of AtPYL13 in seed maturation. Otherwise, I suggest removing this ambiguous observation and tone down their claim accordingly.

Reviewer #3

(Remarks to the Author)

In the revised manuscript, the authors appropriately answer my comments to the previous manuscript, and I have no further question or comment.

Reviewer #4

(Remarks to the Author)

First, my apologies for my tardiness in returning this review: I received the request to consider the revised manuscript immediately before taking a vacation, and it is only in the last week that I've had time to examine the revision.

I'm happy to say that so far as my own comments of the original manuscript are concerned, I think that the authors have addressed all of the comments I raised satisfactorily.

My comments below are principally the correction of some phrases to conform with standard English usage. However I have also made some suggestions where some alterations in the text add clarity: for example an alternative opening to the Abstract (immediately below)

I found the Abstract to lack clarity: in order to rectify this I have rewritten the opening 6 lines in a way that I think make it easier for the reader to understand (and hopefully, then proceed to read the entire paper!):

"Land plants have evolved strategies to survive water deficiency. Among these adaptations, a "drying without dying" strategy, regulated by the hormone Abscisic Acid (ABA), evolved in early land plants and is maintained in the desiccated seeds of angiosperms. This is regulated by a family of ABA receptors (the PYR/PYL/RCAR (PYL) family) able to bind protein phosphatases of the PP2CA family and suppress their inhibitory action on water stress responses. ABA-independent PYLs first appear in an aquatic algal lineage, however their origins and the mechanistic basis of ABA-independent PYL variants in land plants remain uncharacterized. Here we characterize.....etc."

The Introduction begins: Early land plants are mainly poikilohydric and adopt the "drying without dying" strategy, which relies on fundamental stress adaptation mechanisms including osmoregulation and enables plants to survive extreme conditions². Vascular land plants retain the "drying without dying" strategy in specific organs, such as seeds or pollen, but utilize the "avoiding drying" strategy for their vegetative stages, which depends on the activation of stress responses by the phytohormone ABA^{3,4}.

- It is important to realise that this is not a binary choice, and that both acquisition of dehydration tolerance and dehydration avoidance involve activation of multiple stress responses by ABA (for example promotion of expression of genes encoding LEA proteins as a prerequisite for achieving desiccation tolerance, and the implementation of stomatal closure as a means of drought avoidance).

I suggest rewriting the opening lines of the paragraph commencing at Line 41 as follows:

"ABA activation occurs through a protein phosphorylation cascade, in which Snf1-related protein kinase 2 (SnRK2) kinases have a central role. Upon ABA binding, the PYR/PYL/RCAR (PYL) ABA receptors.....etc."

L58-60: "In the nonvascular liverwort *Marchantia*, MpPYL1 shows ABA-enhanced activity to inhibit PP2Cs, although with high basal activity⁹, which triggers high basal stress signaling and actively represses plant growth¹⁵"

- Reference 15, whilst being a comprehensive review of the regulation of plant growth responses, does not make any reference to *Marchantia* or the basal activity of MpPYL1 (the focus is primarily on stress and growth regulation in angiosperms)

L61: potentially enabling broader range of ABA responses
To: potentially enabling a broader range of ABA responses

L63: were subsequently incorporated into plant lineage.
To: were subsequently incorporated into land plant lineages.

L81: The PYL genes from plants form monophyletic group
To: The PYL genes from plants form a monophyletic group

Results section

L81: The PYL genes from plants form monophyletic group
To: The PYL genes from plants form a monophyletic group

L182: This partially explains the weak interaction of PrPYL-ABI1 and strong interactions between algal PYLs and Arabidopsis PP2Cs as well as the constitutive activation of RD29B-LUC expression in pyl duodecuple mutant protoplasts by algal PYLs (Fig. 1, 2, and Supplementary Fig. 2a)

- I can see the interactions between algal PYLs and Arabidopsis PP2Cs, and the activation of RD29B-LUC in Figures 2 and Supplementary 2, but not in Figure 1.

L198-200: Since high basal stress signaling favored the terrestrialization of land plant progenitors, this critical arginine was not conserved during evolution to maintain high basal activity of PYLs.

To: Since high basal stress signaling would have favored the terrestrialization of land plant progenitors, we propose that this critical arginine was not conserved during evolution to maintain high basal activity of PYLs.

L280: several AtPYL13-like proteins differ from the ABA-responsive PYLs by one or two amino...
To: several AtPYL13-like proteins differ from the ABA-dependent PYLs by one or two amino...

L286: We evaluated the ABA responsiveness of these AtPYL13-like proteins
To: We evaluated the ABA dependence of these AtPYL13-like proteins

Discussion

L355: algae, the ABA receptor PYLs are not present until the last common ancestor of land plants
To: algae, the ABA receptor PYLs are not present until the last common ancestor of land plants

L357-8: Our results showed that PYL-like proteins are present in bacteria, might have gained PP2C binding ability in some soil bacteria. Plants acquired PYL genes horizontally...

To: Our results showed that PYL-like proteins are present in bacteria, and might have gained PP2C binding ability in some soil bacteria. Plants likely acquired PYL genes horizontally...

L365-6: Besides, the monomeric PYLs also evolved the strict ABA dependency in mosses
To: Independently, the monomeric PYLs also evolved strict ABA dependency in mosses

L366-7: The ABA-binding and -responsive properties of PYLs confer ABA-dependent or ABA-enhanced PP2C-inhibition activity, which enables land plants to change...
To: The ABA-binding properties of PYLs confer ABA-dependent or ABA-enhanced PP2C-inhibition activity, which enabled land plants to change....

Reviewer #1 (Remarks to the Author):

This study presents an intriguing hypothesis regarding the evolutionary origins of PYL genes, proposing to provide experimental evidence that suggests a bacterial PYL homolog that was transmitted to the plant lineage. However, the central claims of the paper, particularly as stated in the title, are not adequately supported by the presented data. Most of the experimental work essentially reproduces findings from previous studies rather than providing novel insights. Furthermore, the manuscript suffers from substantial clarity issues, with unsupported or misinterpreted claims in the introduction and a presentation of results that obfuscates key conclusions (see specific examples below).

Major Concerns

Query #1:

Functional Characterization of a Bacterial PYL Homolog

A central premise of this study is the identification of a bacterial PYL homolog with ABA-independent activity. However, the assays used fail to assess this function rigorously. The experimental systems either contain endogenous ABA (as in plant-based systems) or rely on overexpression, which can artificially drive protein-protein interactions independent of ligand binding. Notably, even well-characterized ABA-dependent receptors, such as PYL4, exhibit ABA-independent PP2C binding under similar conditions. A more appropriate *in vitro* approach, where protein concentrations are strictly controlled and ABA is entirely absent, is required to definitively assess ligand dependency.

Responses: We thank the reviewer for raising critical methodological considerations regarding ligand dependency assessment. To address these points, we conducted additional *in vitro* biochemical characterization under rigorously controlled conditions: 1) PP2C phosphatase activity assays using pNPP substrate (Fig. 1g, h; Supplementary Fig. 1f); 2) Isothermal Titration Calorimetry (ITC) (Supplementary Fig. 1g). Together with the SnRK2.6 kinase activity assays at previous version (Fig. 1e, f), these experiments employed purified recombinant proteins in systems with precise control of protein concentrations/ratios and ABA levels. Collectively, our data demonstrate that PrPYL exhibits ligand-independent activity, validating its functional characterization as a bacterial ABA-independent PYL homolog.

Concerning *in vivo* validation systems, we are actively developing ABA-deficient plant material

through CRISPR/Cas9-mediated knockout of *ABA2* in the *pyl* duodecuple mutant background. This approach aims to generate plants with combined defects in ABA biosynthesis and perception. While the developmental phenotypes associated with the parental mutant complicate line establishment, we are optimizing growth conditions to overcome these challenges.

Query #2:

Horizontal Gene Transfer Hypothesis Lacks Supporting Evidence

The claim of horizontal gene transfer (HGT) is presented without substantive support. Establishing HGT in a system that diverged over 600 million years ago is inherently difficult. Yet, the authors fail to explore/roll out alternative evolutionary scenarios, such as convergent evolution, which remain equally plausible. A robust phylogenetic analysis should provide statistical confidence measures and exclude other evolutionary possibilities. If the phylogenetic tree showed plant sequences embedded within bacterial lineages, it could suggest recent gene transfer, but the current analysis lacks the necessary rigor. The arbitrary nature of the sequence relationships as presented does not substantiate the proposed HGT event. While the reviewer is not a bioinformatician, the absence of statistical validation and failure to test alternative hypotheses significantly weaken this claim.

Responses: We appreciate the reviewer's engagement with the evolutionary context of this work. As our study fundamentally constitutes an experimental investigation of bacterial PYL functionality but not a phylogenetic analysis, the Cheng et al. (2019) reference primarily provides the evolutionary premise motivating our functional inquiry. Accordingly, we have revised the title and manuscript to frame HGT strictly as a hypothesis.

Our core contribution demonstrates cross-kingdom functional conservation: PrPYL exhibits PP2C-binding capabilities mirroring plant PYLs (Fig. 1d) and shows structural convergence with AtPYL1 (Fig. 1i-l). These results establish mechanistic conservation of PYL function, though definitive determination of evolutionary origins requires specialized genomic analyses beyond this study's scope.

Query #3:

Specific examples:

In the title "...in the seeds of land plants"

The relations described in the study extend beyond seed plants, as the *PYL* gene family is conserved across the entire lineage of land plants, not just in seed-bearing species. Also the title should explicitly state which functional aspect of *PYL* receptors has been retained throughout evolution—specifically, their ability to bind PP2C phosphatases.

Responses: We thank the reviewer for this constructive suggestion. Accordingly, we have revised the title to: “ABA-independent PP2C-binding in *PYL*s traces to bacterial origins and persists in land plants”.

Query #4:

Furthermore, the claim of horizontal gene transfer (HGT) remains a hypothesis rather than a demonstrated/conclusion. As such, it should either be removed from the title or clearly framed as a hypothesis rather than a definitive statement. The current phrasing in the title risks misleading readers by presenting an unverified assumption as an established finding.

Responses: We sincerely thank the reviewer for pointing this out. We agree that horizontal gene transfer (HGT) represents one plausible evolutionary scenario rather than a definitively proven mechanism for the origin of bacterial-derived *PYL*-like sequences in plants. To address this, we have revised the title to: “ABA-independent PP2C-binding in *PYL*s traces to bacterial origins and persists in land plants”. We have also clearly framed horizontal gene transfer of *PYL* genes as a hypothesis throughout the main text.

Query #5:

-Abstract: Has inaccuracies

Examples: Authors claim: “which requires constitutive activation of stress signaling, was mainly adopted by early land plants and persists in seeds.”

This statement is incorrect—ABA levels are crucial for seed dormancy and decrease to allow germination. During dormancy, ABA levels remain high to prevent premature germination. However, for germination to occur, ABA levels must decline (while gibberellins increase).

Responses: We sincerely thank the reviewer for identifying this critical inaccuracy regarding stress signaling in seeds. We have revised the abstract to: “Among these adaptations, ‘drying without dying’

strategy, which requires constitutive stress signaling activation, evolved in early land plants and is maintained in desiccated seeds.” The constitutive stress signaling activation in desiccated seeds is maintained primarily through ABA-dependent mechanism during dormancy, with ABA-independent mechanisms^{1,2}.

Query #6:

Authors claim: “Through evolution, the PYR/PYL/RCAR family (PYL) of ABA receptors gained several functions over the course of evolution, namely inhibition of protein phosphatases (PP2Cs), ABA-enhanced inhibition of PP2Cs, and ABA dependent inhibition of PP2Cs”

The only well-established physiological function of PYL receptors is their ability to inhibit PP2C phosphatases, which are central to ABA signaling. If the authors intended to suggest an additional function, they should clarify it and provide supporting evidence.

Responses: We sincerely thank the reviewer for pointing this out. Accordingly, we have revised the abstract to: “During evolution, the PYR/PYL/RCAR (PYL) family of ABA receptors acquired PP2C binding and inhibitory activities.”

Query #7:

Authors claim: “However, it remains unclear how the ABA-irresponsive PYLs originated and whether they exist in land plant. “

This statement is incorrect—it has been shown that PYL receptors in land plants likely originated from algal ancestors, and homologous proteins are present across all land plant lineages. This suggests that the PYL family is not exclusive to land plants but instead evolved from pre-existing proteins in algae before being co-opted into the ABA signaling pathway in terrestrial plants. See doi.org/10.1073/pnas.1914480116

Responses: We sincerely thank the reviewer for prompting critical clarification regarding PYL evolution. As requested by one reviewer, we changed “ABA-irresponsive” PYLs to “ABA-independent PYL-like proteins”. We have accordingly revised the abstract: “However, the origin of algal PYLs and the mechanistic basis of ABA-independent PYL variants in land plants remain uncharacterized.” We define these ABA-independent PYL-like proteins in the subsequent sentence as those “retain constitutive PP2C binding but lack ABA-enhanced inhibitory activity”.

Query #8:

Authors claim: “*AtPYL13* and *AtPYL13*-like genes were highly expressed in dicot seeds, suggesting the transformation from ABA receptors to ABA-irresponsive PYLs after expansion of the gene family”.

This statement misrepresents the actual expression profile of *PYL* in Arabidopsis seeds. While *PYL13* expression can be detected, public datasets (e.g., BAR ePlant) indicate that its expression is approximately 100-fold lower than *PYR1*, suggesting it plays a minor role in seed germination. Additionally, experimental evidence challenges the significance of *PYL13* in this context. For example, activating *PYR1* alone, using a specific agonist, was sufficient to inhibit germination. Dimeric multiple mutants exhibit an early germination phenotype, further suggesting that the primary regulators of germination are other *PYL* receptors. See Park 2009 DOI: 10.1126/science.1173041 and Okamoto 2011 <https://doi.org/10.1073/pnas.1305919110>

Given these findings, if *PYL13* has any role in germination, it is likely minimal or redundant, and its functional relevance in this process remains questionable.

Responses: We appreciate the reviewer’s insightful comments regarding the expression and functional relevance of *AtPYL13*. We agree that *AtPYL13* exhibits very low expression levels in most tissues and does not play a central role in seed germination. To clarify this point, we have revised the abstract as follows: “*AtPYL13*-like genes in dicots exhibited high expression during seed maturation and in desiccated seeds, suggesting a functional shift from canonical ABA receptors to ABA-independent *PYL*-like proteins following gene family expansion.” This statement aligns with the transcriptional analyses presented in our manuscript (Fig. 4d, e; Fig. 5a, b). Based on their distinct expression patterns, we propose that *AtPYL13*-like genes primarily contribute to seed vigor maintenance rather than seed germination regulation. We have also clarified this point in the Discussion.

Query #9:

-Introduction- Has inaccuracies

Some Examples:

Authors claim: “Terrestrialization is a pivotal event in the evolution of green plants, which requires mechanisms to survive under drought stress.”

The term "drought stress" is not appropriate to describe the primary challenges of terrestrialization (refers to the evolutionary transition from an aquatic to a terrestrial environment). Terrestrialization involves adaptation to a gradual shift from consistently wet conditions to environments with fluctuating water availability rather than exposure to extreme drought. Using "drought stress" oversimplifies the complexity of this transition, which encompasses not only water limitation but also factors such as desiccation tolerance, gas exchange regulation, light intensity, and structural adaptations to terrestrial life.

Response: We sincerely thank the reviewer for this insightful observation. We agree that “drought stress” does not fully capture the multifaceted challenges of terrestrialization. As suggested, we have revised the Introduction to more accurately reflect the complexity of this evolutionary transition: “Terrestrialization represents a pivotal event in green plant evolution, requiring adaptive mechanisms to overcome multiple environmental challenges, particularly fluctuating water availability.”

Query #10:

Authors claim: “Plants mainly utilize two strategies to survive under water deficiency: “drying without dying” or “without drying”.

These terms are not clear.

Responses: We appreciate the reviewer’s valuable comment regarding the clarity of these terms. In the revised manuscript, we have now explicitly defined these survival strategies as: “Plants employ two principal strategies to cope with water deficiency: “drying without dying (desiccation tolerance)” or “avoiding drying (drought avoidance)”³.

Query #11:

Authors claim: “The early land plants are mainly poikilohydric and adopt the “drying without dying” strategy, which requires constitutive activation of stress signaling and enables plants to get through extreme conditions” –Early land plants possess mechanisms for water regulation and stress adaptation that are independent of vascular systems, as described in the cited paper. A more relevant and evolutionarily conserved mechanism is osmoregulation, which is shared between mosses and seed plants.

Regarding the statement “PP2Cs can be ABA-dependent or ABA-enhanced”, this terminology is

unclear. PP2C inhibition occurs through both ABA-dependent and ABA-independent mechanisms, and the distinction between "ABA-enhanced" and "ABA-dependent" needs to be explicitly defined to avoid confusion.

Responses: We sincerely appreciate the reviewer's insightful comments regarding these important mechanistic distinctions. We have carefully revised the text to incorporate these points.

For the poikilohydric strategy discussion: "Early land plants are mainly poikilohydric and adopt the "drying without dying" strategy, which relies on fundamental stress adaptation mechanisms including osmoregulation and enables plants to survive extreme conditions."

For the PP2C terminology clarification: "The ABA-binding properties of PYLs vary significantly in seed plants; likewise, their PP2C inhibition mechanisms show distinct ABA-dependence patterns: (i) strictly ABA-dependent (no basal inhibition activity without ABA, e.g., the dimeric subfamily III PYLs in angiosperms)⁴, (ii) ABA-enhanced (basal inhibition activity potentiated by ABA)⁵, or (iii) ABA-independent (not modulated by ABA). The "ABA-independent PYL-like proteins" include: 1) constitutively active PYLs that activate stress responses regardless of ABA⁶; and 2) inactive PYLs incapable of activating stress responses."

Query #12:

Authors claim: "The entirely ABA dependent dimeric PYLs emerged in angiosperms and have nearly no basal activity"-This applies to other PYLs as well— not just dimeric forms. This should be considered as it's not the only mechanism of low basal activity (in Arabidopsis and other species), as indicated in DOI: 10.1016/j.cub.2024.12.043."

Authors claim: "which enables angiosperms to have a wider range of ABA responses and improved balance between growth and stress responses"-This is an assumption and should be stated as such.

Responses: We sincerely thank the reviewer for these insightful comments. We have revised the text as follows: "Strictly ABA-dependent PYLs with minimal basal activity emerged in Bryophytina mosses and angiosperms^{5,7}, potentially enabling broader range of ABA responses and improved balance between growth and stress responses."

Query #13:

Authors claim: “However, it remains unclear how the ABA-irresponsive PYLs originated.”

The focus should be more specific—there is evidence that land plant lineages and the common ancestor of *Zygnema circumcarinatum* possessed a nonresponsive PYL. What remains unclear is how this protein was ‘adopted’ into the plant lineage.”

Responses: We sincerely thank the reviewer for this insightful comment. We have revised the text as follows: “However, it remains unclear how ABA-independent PYL-like proteins emerged and were subsequently incorporated into plant lineage.”

Query #14:

-Specific points in the result section:

The section titled: “The bacterial PYL-homolog from *Paraburkholderia rhynchosiae* possesses PP2C-binding ability”

Authors claim: “It is proposed that the ancestral PYLs in plants were gained through horizontal gene transfer from soil bacteria based on phylogenetic analyses (Fig. 1a).” –The phylogenetic analysis does not support horizontal gene transfer occurring over 600 million years ago.

Responses: We sincerely thank the reviewer for this important observation. Our phylogenetic analysis indeed does not provide definitive support for horizontal gene transfer events occurring over 600 million years ago. Accordingly, we have revised the manuscript by properly citing Cheng et al. (2019)⁸ and repositioning Fig. 1a.

Query #15:

Authors claim: “Despite lacking ABA responsiveness”

This should be tested *in vitro* using assays that directly assess the involvement of ABA as a ligand, such as a PP2C inhibition assay, isothermal titration calorimetry, or any equivalent assay.

Responses: We thank the reviewer for this important methodological suggestion. We have now performed: 1) PP2C inhibition assays using pNPP substrate (Fig. 1g, h; Supplementary Fig. 1f); 2) Isothermal Titration Calorimetry (ITC) (Supplementary Fig. 1g). Combined with our previous SnRK2.6 kinase activity assays (Fig. 1e, f), these results confirm that PrPYL lacks ABA binding and responsiveness.

Query #16:

The authors identified the interaction between bacterial PrPYL and PP2C via a yeast two-hybrid assay. However, this interaction was not reproducible in planta nor when SnRK2 competed for the PP2C interface in an in vitro assay. Based on this, the authors concluded that there is an interaction with PP2C but no PP2C inhibition. As stated, 'These results suggested that the bacterial PrPYL possesses PP2C-binding but not PP2C inhibitory activity.' This conclusion is flawed for two reasons: 1. The interaction between PYL and PP2C typically blocks the active site of the PP2C enzyme. Therefore, a 'canonical interaction' should, by default, result also in inhibition. 2. To properly claim this, the interaction and PP2C activity must be evaluated through specific assays, not as a default by a negative result. At this point, it is an unsubstantiated guess.

An alternative explanation could be that, in yeast, the protein steady-state concentration is high, allowing for the monitoring of low-affinity interactions. *In vitro* and *in-planta*, however, are more challenging, as other proteins could compete for the PP2C interface. Consequently, it is unsurprising that the weak interaction with bacterial PrPYL is outcompeted by SnRK2s (whether from plants or purified protein), resulting in the absence of an interaction signal."

Responses: We sincerely thank the reviewer for this insightful critique and valuable hypothesis. Our experimental data demonstrate that: (1) PrPYL exhibits weak PP2C binding without inhibitory activity (Fig. 1); (2) PrPYL lacks the conserved Gate loop (S/TGLPA) (Supplementary Fig. 1a) that mimics SnRK2 activation loop in SnRK2-PP2C complex⁹; and (3) Molecular docking confirms PrPYL's inability to properly occupy ABI1's catalytic site (Response Fig. 1). These results indicate a non-canonical binding mechanism distinct from angiosperm PYLs. As suggested, we have incorporated the possibility that weak PrPYL-PP2C interactions may be outcompeted by endogenous proteins (e.g., plant PYLs and SnRK2s) into our main text. We have revised the text as follows: "The weak PrPYL-PP2C interactions may be outcompeted by endogenous proteins (e.g., plant PYLs and SnRK2s), thus generating very weak reconstituted LUC signals in planta. These results suggest that PrPYL represents a prototype ABA receptor."

Response Fig. 1. Molecular docking of AtPYL1 and PrPYL into catalytic active site of AtABI1.

Query #17:

The section titled “ABA-irresponsive PYLs exhibit PP2C inhibitory activity in Zygnematales”.

Sun et al. (2019) showed that algal PYL interacts with PP2C in an ABA-independent manner. In that study, Sun also demonstrated that ZcPYL8 cannot bind ABA (by isothermal calorimetry titration). In this current study, the authors aimed to test additional algal PYLs. They used an Arabidopsis *PYL* mutant with endogenous ABA levels, which were unsuitable for testing ABA-independent activity. The SnRK activity (following release) assay is sensitive to PYLs with basal ABA-independent activity but cannot distinguish if there is a mild ABA effect. These two assays do not support the claim that the 'PYL homologs gained ABA-independent PP2C inhibitory activity during the evolution of Zygnematales.' In vitro assays that directly test ABA binding would be better suited to scratch the surface of this hypothesis. I suggest starting by following the biochemical assays performed by Sun in 2019.

Response: We thank the reviewer for this critical methodological suggestion. To address these concerns, we have now conducted: 1) PP2C inhibition assays using pNPP substrate (Fig. 2g-j; Supplementary Fig. 2b); and 2) Isothermal Titration Calorimetry (ITC) to directly assess ABA binding (Supplementary Fig. 2c). Together with our SnRK2.6 kinase activity assays (Fig. 2e, f), these experiments confirm that MeenPYL3 functions in an ABA-independent manner.

Query #18:

The section titled “An ABA-irresponsive PYL exists in Arabidopsis”

The authors claimed that “the wild-type AtPYL13 could not activate *RD29B-LUC* with ABA treatment, while the expression of AtPYL13-Q38K/F71L restored the ABA-dependent activation of *RD29B-LUC* in the *pyl* duodecuple mutant protoplasts (Fig. 3c), indicating that AtPYL13 is otherwise functional and that the two natural sequence variations in the CL1/CL2 domains of AtPYL13 confer ABA irresponsiveness”

A paper from the Grill group addresses PYL13 activity and the specific mutation analyzed here: <https://doi.org/10.1073/pnas.1322085111>. These mutations were thoroughly tested in that study. I failed to see the added value of repeating them here."

Response: We sincerely appreciate the reviewer’s guidance in highlighting the foundational work by Grill lab. Our intent is not to challenge but to complement existing knowledge. While previous studies established AtPYL13’s ABA responsiveness using transient expression in ABA-deficient backgrounds (*aba2-1*), our study extends this observation by demonstrating in a genetic null background lacking endogenous ABA receptors (*pyl* duodecuple mutant) that: (i) wild-type AtPYL13 fails to activate ABA-responsive reporters (Fig. 3a), and (ii) AtPYL13-Q38K/F71L restores ABA-dependent signaling (Fig. 3c). This independent validation underscores the functional plasticity of PYL13’s ligand-binding domains under physiologically relevant receptor-free conditions.

Query #19:

The sections titled “Two invariant residues are crucial for PYL ABA receptors.”, “ABA-irresponsible PYLs exist in seed plants” and “ABI3 mediates seed-specific expression of *AtPYL13*”

I don’t understand the rationale behind the ectopic expression of impaired-active mutated *PYLs*. What can be learned from these experiments? This entire section seems unrelated to the main focus of the study. Furthermore, it is well-established that dimeric PYLs are key players in seed germination biology. This has been tested through agonist activation of specific PYLs and by examining expression levels (actual reads versus the relative expression tested here). In fact, *PYL13* expression is barely detectable."

Response: We thank the reviewer for raising these critical points. Our ectopic expression experiments serve two key purposes—mechanistic validation and functional essentiality. By introducing Q38K/F71L mutations into ABA-independent PYL13 (Fig. 3c-f), we demonstrate these ancestral residues sufficiently restore ABA responsiveness. Reciprocal mutagenesis in canonical ABA receptors (Fig. 3h-k) confirms these invariant residues are strictly required for ABA perception.

While dimeric PYLs dominate germination control, our transcriptional analysis (Fig. 5) reveals *AtPYL13*'s selective upregulation specifically during seed maturation—not germination—suggesting its ABA-insensitive activity may enable distinct developmental programming at this critical stage.

Reviewer #2 (Remarks to the Author):

Lu et al. report the evolutionary trajectory of PYL ABA receptors. Based on phylogenetic and functional analyses, they found that an ABA-irresponsive PYL in soil bacteria was horizontally transferred from soil bacteria, gained PP2C interaction in *Paraburkholderia rhynchosiae*, became capable of inhibiting PP2C in the common ancestor of Zygnematales and land plants, evolved into ABA receptors in liverworts, and then diversified to possess various affinities to ABA, including an irresponsive receptor exclusively expressed in seeds.

The presented work attracts a wide range of biologists who study protein evolution and hormone signaling. The work is comprehensive and based on solid functional analysis. The following are my comments to improve the manuscript:

Query #1:

[1] The title needs to be reconsidered:

Response: We thank the reviewer for this suggestion. The title has been revised to: “ABA-independent PP2C-binding in PYLs traces to bacterial origins and persists in land plants”.

Query #2:

(2) "Horizontally transferred" typically refers to genes, not proteins.

Response: We thank the reviewer for pointing this out to us. The text has been amended to refer to *PYL* genes (not proteins) when discussing horizontal transfer events.

Query #3:

(3) I would think that the ABA-irresponsive receptors that was horizontally transferred from soil bacteria and retained functions in seeds are qualitatively distinct. PrPYL is a prototype of ABA receptor, while AtPYL13 is a variant of ABA receptor. The current title sounds continuous ABA-irresponsive receptors.

Response: We thank the reviewer for this insightful clarification. The title has been revised to: “ABA-independent PP2C-binding in PYLs traces to bacterial origins and persists in land plants”. Additionally,

we have clarified in the Results that “PrPYL represents a prototype ABA receptor.”

As requested by one reviewer, we changed “ABA-irresponsive” PYLs to “ABA-independent PYL-like proteins”. We have also clarified the distinct stepwise transition within the Brassicaceae lineage (Supplementary Fig. 3a; steps i-iii): (i) initial divergence of the AtPYL11/12/13 clade¹⁰ (node marked with a purple asterisk), which has only been identified in eudicots; (ii) secondary divergence of AtPYL13 (gold-highlighted branch) from the AtPYL11/12 subclade; (iii) subsequent acquisition of lineage-specific variations within the AtPYL13 subclade, including one in *Camelina sativa* and two in the *Arabidopsis* lineage (*A. thaliana*, *A. lyrata*, and *A. halleri*; variation sites labelled in red). The stepwise progression represents reversion from ABA-enhanced to ABA-independent states, suggested that ABA-independence in AtPYL11/12/13 clade is a derived trait re-evolved through specific adaptations for seed desiccation tolerance (“drying without dying”).

Query #4:

(4) Related to this, the overall context appears to have two distinct parts. I recommend the authors revise the text to create a cohesive narrative.

Response: We sincerely thank the reviewer for this constructive suggestion. We have thoroughly revised the manuscript to enhance narrative cohesion, specifically streamlining the conceptual progression from bacterial prototypes to plant variants, strengthening transitional linkages between evolutionary and functional analyses, and emphasizing mechanistic continuity in ABA-independent PP2C binding.

Query #5:

[2] Co-evolution of PYL and PP2C:

The authors examined the function of ancient PYLs and their interaction with *Arabidopsis* PP2Cs. The evolution of PYLs during this period heavily depends on the nature of PP2C. The evolution/conservation of required amino acid residues in PP2Cs should be discussed. Ideally, PP2C residues required for PYL interaction or inhibition can be separately discussed. If possible. This is a core part of the story.

Response: Thank you for your valuable suggestions. Accordingly, we conducted preliminary *in silico* analyses of PP2C evolution and PP2C/PYL interactions. First, we obtained representative PYL/PP2C sequences from *Physcomitrella patens*, *Zygnema circumcarinatum*, *Mesotaenium endlicherianum*, *Marchantia polymorpha*, and *Paraburkholderia rhynchosiae*. These sequences are primarily based on previous work¹¹, with the exception of *Mesotaenium endlicherianum* and *Paraburkholderia rhynchosiae*. For these two species, we used BLAST-P to identify putative ABI1-homolog sequences.

PP2C is present in bacteria (e.g., *Bacillus subtilis* PrpC, RsbU, and RsbP). Thus, we performed BLAST-P searches in *Paraburkholderia rhynchosiae* using *B. subtilis* PrpC, RsbU, and RsbP as queries. We identified PrpC and RsbP homologs in this symbiotic bacterium. Hereafter, we refer to the *Paraburkholderia rhynchosiae* PrpC homolog as PrPP2C.

The structural basis for clade-A PP2C inhibition by PYLs involves docking of the PYL S/TGLPA gate loop (CL2) into the PP2C DGHG catalytic active site loop^{12,13}. Sequence alignment (Response Fig. 2a/b) shows the PP2C catalytic loop (D243-G246 in AtHAB1) is highly conserved, except in PrPP2C (single-residue variation). The S/TGLPA motif (CL2) is conserved in *Marchantia polymorpha* and *Arabidopsis*, while *Physcomitrella patens* has SSLPA, MeenPYL3 has TGIPG, ZcPYL8 has SGIPG (Response Fig. 2b), and PrPYL lacks this motif (retaining only S52, which mediates PrPYL–AtABI1 binding as shown in the main text). Another key PP2C residue is AtHAB1 W385 (AtABI1 W300), which points into the ABA-binding pocket and makes a water-mediated hydrogen bond to the ketone group of ABA. This interaction helps stabilize the ABA-PYL-PP2C triple-component complex. Notably, MeenABI1 and PrPP2C show variations at this residue (Response Fig. 2a), consistent with their ABA-independent functions.

We also used AlphaFold3 to predict PYL–PP2C complex structures (PrPYL–PrPP2C, ZcPYL8–ZcABI1Δ1-247, MeenPYL3–MeenABI1, MpPYL–MpABI, PpPYL–PpABI1A). Due to low confidence in some predictions, only PrPYL–PrPP2C and ZcPYL8–ZcABI1Δ1-247 were analyzed further. For plant-derived ZcPYL8–ZcABI1Δ1-247 (Response Fig. 2c), the predicted structure shows the ZcPYL8 gate loop packed against the PP2C catalytic loop (D84–G87), supporting its ABA-independent function. For PrPYL–PrPP2C (Response Fig. 2d), we predict interaction without inhibition for two reasons: (1) PrPYL lacks the SGLPA loop, and S52 is distal to the interface; (2) the PrPP2C catalytic loop is distant from potential binding sites, preventing active-site blockade.

PYL/PP2C co-evolution is indeed compelling. However, our study focuses on ABA-independent PYL evolution, biochemistry, and biological roles. Current limitations—specifically, the absence of multi-species PYL/PP2C complex structures and low-confidence *in silico* predictions—preclude

deeper analysis. We aim to address this question in future work.

Response Fig. 2. Preliminary in silico analysis of co-evolution of PYL/PP2C.

Query #6:

[3] ABA-irrespective PYLs:

The functionality of ABA receptors or PYLs can be assigned by their ability to inhibit PP2C and their ABA dependency, regardless of whether they act as monomers or dimers. ABA receptors evolved as PYLs with ABA-enhanced PP2C inhibition. The present work demonstrated that ABA-irrespective PYLs constitutively inhibit some PP2Cs, but their PP2C inhibition activity is not quantitatively compared with other receptors. The ability to induce downstream events (or inhibit PP2C activity) among PYL members is a critical aspect of their functionality, as they function in the same cells with other ABA-responsive receptors *in vivo*. This makes it difficult for readers to assess their role *in vivo*. The phenotype was primarily assessed by overexpressors. *In vitro* SnRK2 activation was quantified, but this quantification does not provide the degree of strength among family members. The authors discussed, “Although the *pyl* duodecuple mutant is viable and fertile, we cannot get any seeds for the *pyl* quattuordecuple mutant that has further mutated *AtPYL6* and *AtPYL13*, supporting the critical role of *AtPYL13* in fertility and seed maturation.” The multigenic mutant phenotypes are the most critical data to assess the role of *AtPYL13* among family members, especially in seeds where it is exclusively expressed. I encourage the authors to describe the multigenic mutant phenotypes in more detail (I suggest that the overexpressor phenotype be included in the supplemental materials).

Related to the above, I am curious if the two critical residues (Q and F) in ABA-irrespective PYLs are only characteristic conservations of PYL13-like proteins.

Response: We sincerely appreciate the reviewer’s insightful comments regarding the functional assessment of ABA-independent PYL-like proteins, particularly *AtPYL13*.

For quantitative comparison of PP2C inhibition, we agree that assessing inhibition strength is crucial. The PYL13 protein is difficult to purify, prone to degradation, and rapidly loses activity during *in vitro* assays. Relevant comparative data were reported in Zhao et al. (2013) (Figure 4B). These results demonstrate that PYL13 exhibits differential inhibitory potency towards specific PP2Cs. For example, PYL13 shows significantly stronger inhibition of PP2CA (a major seed-expressed PP2C) compared to PYR1 and PYL2 in the absence of ABA. This supports our conclusion that ABA-independent PYL-like proteins such as PYL13 can exert selective and potent effects on specific PP2C targets, particularly PP2CA, which is highly relevant to its seed-specific function.

We fully agree and have expanded the description of the multigenic mutant phenotypes. We have revised the text as follows: “Although the *pyl* duodecuple mutant is viable and fertile, we cannot obtain

any homozygous seeds for the *pyl* quattuordecuple mutant which has mutations in *AtPYL6* and *AtPYL13*¹⁴, supporting the critical role of *AtPYL13* in fertility and seed maturation. The *pyl* quattuordecuple mutant occasionally produces a few heterozygous seeds when grown in the same pot with Col-0 wild-type¹⁴, suggesting that the infertile phenotype is mainly caused by male-sterility or embryonic failure.”

Since we have added additional data in Supplementary Fig. 3 and it was already a large figure, we retained the overexpressor phenotype data in Fig. 3. Regarding the critical residues (Q and F), since we identified *PYL13*-like proteins based on these residues, this category of ABA-independent *PYL*-like proteins is defined by them. However, other unidentified ABA-independent *PYL*-like proteins may utilize distinct critical residues.

Reviewer #3 (Remarks to the Author):

The manuscript by Lu et al. reported an evolutionary model of ABA-irresponsive PYL-like proteins and proposed their roles in desiccation tolerance of angiosperms seeds. They demonstrated *in vitro* that a bacterial PYL-like protein can interact with but not inhibit the activity of Arabidopsis ABI1 in an ABA-independent manner, supporting a previous idea that plant *PYL* family is originated from soil bacteria. They also performed the same experiment set against *PYL*-like genes from Zygnematophyceae, showing that algal PYLs have the binding and inhibitory function against Arabidopsis PP2Cs in an ABA-independent manner. They focused on two important amino acids located at conserved loops 1 and 2 which are not conserved in ABA-irresponsive PYLs. The two amino acids were also not conserved in seed-specific ABA-irresponsive AtPYL13 and substitution of these amino acids to that of ABA-responsive PYLs were sufficient to confer ABA-responsiveness. They further proposed that *AtPYL13* was under the regulation of seed-specific ABI3 transcription factor, which is essential for the acquisition of seed desiccation tolerance.

It is a potentially interesting research paper to give insights into the evolution of PYLs and plant desiccation tolerance; however, the experimental evidence was not sufficient support their claims.

Query #1:

Moreover, I recommend the authors should discuss with an expert of evolutionarily research since the manuscript contains many flaws in evolutionary aspect. For example, “PYLs then gained ABA-binding and -responsive activities in liverworts,”. Liverworts are extant plant group and not ancestor of land plants.

Response: We sincerely thank the reviewer for identifying this critical error in our evolutionary interpretation and for the recommendation to consult an expert. We have revised the text as follows: “PYLs then acquired ABA-binding and -responsive functions in the last common ancestor of land plants.” We have also clarified other statements related to evolutionary aspects.

We extensively discussed these evolutionary aspects with Prof. Guanzhu Han, an expert in plant evolutionary biology. Based on his guidance and a careful re-examination of our evolutionary narrative throughout the manuscript, we revised the text to clarify the evolutionary trajectory of the *PYL* family. We are grateful for the reviewer's insight, which significantly improved the accuracy of our evolutionary narrative.

Query #2:

My major concern is that there is no *in vivo* experimental data showing that PYL13 is directly regulated by ABI3 and plays an essential role like *ABI3* in seed desiccation tolerance.

Response: We thank the reviewer for raising this important point. To address the direct transcriptional regulation, we performed ChIP-qPCR analysis using mature green seeds (15 days after flowering) of an *OE-ABI3-Flag* transgenic line. This experiment demonstrated enrichment of *AtPYL13* promoter fragments containing the RY motif by immunoprecipitated ABI3-Flag (Figure 5g), providing *in vivo* evidence that ABI3 directly binds the *AtPYL13* promoter and regulates its seed-specific expression.

Regarding the functional role in desiccation tolerance, we acknowledge that our data do not demonstrate that *PYL13* alone is essential in the same manner as the master regulator *ABI3*. *ABI3* orchestrates seed desiccation tolerance primarily by mediating the transcriptional reprogramming of a suite of maturation genes, including key effectors like *LEAs*¹⁵. We propose that *PYL13*, as an ABA-independent *PYL*-like protein whose expression is activated by *ABI3*, contributes alongside seed-expressed ABA receptors (such as *PYL11*, *PYL12*, *PYR1*, *PYL2*, *PYL5*, *PYL7* and *PYL9*, see Supplementary Fig. 5c) during late maturation. This collective activity of *PYLs* is likely important for fine-tuning processes like the acquisition of desiccation tolerance, working downstream or in parallel to the core *ABI3*-mediated program.

Query #3:

I was also confused about the “ABA-irresponsive” *PYLs*. What are the criteria for this word? For example, MeenPYL1/2 have no capacity but MeenPYL3 does have the capacity to activate *RD29B-LUC* in *pyl* protoplasts with or without ABA. These are all ABA-irresponsive *PYLs*? Probably it should be, since the authors describe *AtPYL13* as ABA-irresponsive, which cannot activate *RD29B-LUC* in *pyl* protoplasts like MeenPYL1/2.

Response: We thank the reviewer for raising this important point. As requested by one reviewer, we changed “ABA-irresponsive” *PYLs* to “ABA-independent *PYL*-like proteins”. We define “ABA-independent *PYL*-like proteins” as *PYL* proteins whose regulatory activity is not modulated by ABA. This category includes: 1) constitutively active *PYLs* (e.g., MeenPYL3) that activate stress responses (e.g., *RD29B-LUC*) regardless of ABA; 2) inactive *PYLs* (e.g., MeenPYL1, MeenPYL2, and *AtPYL13*) incapable of activating stress responses.

Thus, MeenPYL3 is correctly classified as an ABA-independent PYL-like protein because it functions constitutively without ABA binding. We have clarified this terminology in the revised manuscript to reflect that ABA-independent PYL-like proteins are not ligand-binding receptors.

Query #4:

Is ABA-irresponsive PYL re-evolved in angiosperms? What are the differences between ABA-irresponsive algal PYLs and ABA-irresponsive angiosperms PYLs.

Response: This is a very interesting question. While the fundamental ABA-independent basal activity represents the ancestral PYL state, our phylogenetic analysis reveals a distinct stepwise transition within the Brassicaceae lineage (Supplementary Fig. 3a; steps i-iii). We have revised the text as follows: “This progression is evidenced by: initial divergence of the AtPYL11/12/13 clade¹⁰ (node marked with a purple asterisk), which has only been identified in eudicots; (ii) secondary divergence of AtPYL13 (gold-highlighted branch) from the AtPYL11/12 subclade; (iii) subsequent acquisition of lineage-specific variations within the AtPYL13 subclade, including one in *Camelina sativa* and two in the *Arabidopsis* lineage (*A. thaliana*, *A. lyrata*, and *A. halleri*; variation sites labelled in red).” The stepwise progression represents reversion from ABA-dependent to ABA-independent states, suggested that ABA-independence in AtPYL11/12/13 clade is a derived trait re-evolved through specific adaptations for seed desiccation tolerance ("drying without dying").

The two natural amino acid variations in AtPYL13 confer ABA insensitivity, as indicated by the ABA-responsiveness of AtPYL13-Q38K/F71L (Fig. 3c-g). In contrast, mutation of the two natural amino acid variations in MeenPYL3 (R63K/I93L) did not alter ABA responsiveness of MeenPYL3; this result is shown in Supplementary Fig. 2d.

We have clarified this transition in the main text and modified Supplementary Fig. 3a and Supplementary Data 1 to highlight key evolutionary stages. Thus, the transformation describes recent reacquisition of ABA-independence within a derived eudicot lineage, consistent with the abstract's description of functional innovation after gene family expansion.

Query #5:

Comments

L88- and Fig.1a. What is the reason to add *SnRK2* and *ABF* in the *pyl* protoplast transient assay? Was

introduction of *AtPYL*(s) not sufficient to recover ABA response?

Response: Introduction of *PYL* receptors is sufficient to recover ABA response in the *pyl* duodecuple protoplast transient assay, while addition of *SnRK2* and *ABF* supports robust activation of luciferase (LUC) signals in the presence of *PYL* receptors. Therefore, we co-expressed *SnRK2* and *ABFs* to evaluate the ABA-responsiveness of *PYLs* in the *pyl* duodecuple protoplast transient assay. We have clarified it in Supplementary Methods.

Query #6:

L96- This experiment should include a positive control of ABA-dependent *PYL*.

Response: We thank the reviewer for this suggestion to strengthen the characterization of ABA-independent binding between *PrPYL* and *AtPP2Cs*. In the revised version, we performed additional ITC assays to evaluate ABA-responsiveness of *PrPYL* and found that *PrPYL* does not bind ABA (Supplementary Fig. 1g). Therefore, we did not further evaluate ABA-responsiveness for *PrPYL-PP2C* binding since *PrPYL* lacks ABA-binding ability.

Query #7:

L121- and Fig. 1k. This is also an extant bacterium and not a doner of *PYL*, therefor, misleading. The authors did not show that the target of bacterial *PYLs* is *PP2Cs*, and Fig. 1k should be deleted or revised.

Response: We sincerely thank the reviewer for identifying this critical error in our evolutionary interpretation. We have revised the text as follows: “Our results support the hypothesis that *PYLs* originated via horizontal gene transfer from bacteria, but acquired their *PP2C*-inhibitory and ABA-binding functions in plants.” We have deleted Figure 1k as suggested.

Query #8:

L122- The Figure layout should be arranged according to the order in the main text. There is no mention about Fig. 2a before Fig.2b.

Response: We thank the reviewer for this suggestion and have checked the figure order layout in main text, ensuring sequential referencing aligns with the figure layout. This adjustment clarifies the

presentation flow.

Query #9:

L155- Fig. 2e,f need quantification.

Response: Thanks. We have now included the relevant quantification bars directly to Fig. 2e,f at the bottom.

Query #10:

L162- Is this Ser the same with Ser112 described in L115? The numbers do not match each other.

Response: We thank the reviewer for this important observation. The serine residue within the "S/TGLPA" motif (line 162) is indeed functionally equivalent to Ser112 in AtPYL1 (line 115). This conserved residue resides in the gate loop (CL2 loop) that docks into the PP2C active site, where it forms critical hydrogen bonds stabilizing the PYL-PP2C complex. The differing numbering reflects sequence divergence among PYL orthologs.

We have revised the text as follows: "While Ser52 in PrPYL occupies a position structurally equivalent to Ser112 in AtPYL1 within the 'S/TGLPA' motif (CL2 loop) and is predicted to form a hydrogen bond with Gly180 in ABI1, the distance of 4.5 Å (instead of 3.0 Å and less) (Fig. 1j) could partially explain the weak interaction between plant PP2Cs and PrPYL."

Query #11:

L178- Reduction of high basal activity should be supported by an appropriate statistical analysis.

Response: Thanks. We have added statistical analyses comparing basal LUC signals between AtPYL4 and AtPYL4-K81R and revised the main text accordingly as follows: "To our surprise, the AtPYL4-K81R mutant lost its high basal activity in elevating *RD29B-LUC* expression (Supplementary Fig. 2d), suggesting that arginine is a better choice than lysine in CL1 to generate PYL ABA receptors with no basal activity."

Query #12:

L180-182 The description is not clear.

L181- What does “original land plants” mean?

Response: We thank the reviewer for pointing this out. Indeed, the previous description was unclear: “Since high basal stress signaling favors the terrestrialization of original land plants, this critical arginine was not reserved during evolution.”

Therefore, we have revised the text as follows: “Since high basal stress signaling favored the terrestrialization of land plant progenitors, this critical arginine was not conserved during evolution to maintain high basal activity of PYLs.”

Query #13:

L184- “Zygnematales” are extant plants.

L185- I mentioned above.

Response: Thanks. We have revised the text as follows: “The ABA-independent PYL-like proteins evolved before the last common ancestor of Zygnematophyceae and land plants, with some lineages subsequently gaining PP2C-inhibition capability. PYLs then acquired ABA-binding and -responsive functions in the last common ancestor of land plants^{6,7}.”

Query #14:

L186 and L196 The authors should describe the PYLs subfamilies with their function in the introduction.

Response: We thank the reviewer for this suggestion. We have revised the introduction as follows: “The ABA-binding properties of PYLs vary significantly in seed plants; likewise, their PP2C inhibition mechanisms show distinct ABA-dependence patterns: (i) strictly ABA-dependent (no basal inhibition activity without ABA, e.g., the dimeric subfamily III PYLs in angiosperms)⁴; (ii) ABA-enhanced (basal inhibition activity potentiated by ABA)⁵; or (iii) ABA-independent (not modulated by ABA). The “ABA-independent PYL-like proteins” include: 1) constitutively active PYLs that activate stress responses regardless of ABA⁶; and 2) inactive PYLs incapable of activating stress responses.”

Query #15:

L191- There is no data for AtPYL7. What is the criterion of “basal activity”? Is there any significant difference between AtPYR1 and AtPYL9 autoactivation?

Response: Thanks. We have now included the data for AtPYL7 (Fig. 3a). The autoactivation of PYR1, PYL1, and PYL7 was not significantly different from that of the control, while the autoactivation of PYL5-6 and PYL8-12 was significantly different from that of the control (Response Fig. 3).

Response Fig. 3. Autoactivation of *Arabidopsis* PYLs.

Query #16:

L193- Please describe more about the “stepwise transformation”. For me, it is not evident from the figures. It will be helpful to indicate where transformation occurred on the phylogenetic tree.

Response: We thank the reviewer for this critical insight. We have clarified the stepwise transition in the main text and modified Supplementary Fig. 3a to visually highlight key evolutionary stages: “Phylogenetic analysis of Brassicaceae PYLs revealed a stepwise transition from ABA-enhanced receptors to ABA-independent paralogs after gene family expansion (Supplementary Fig. 3a; steps i-iii). This progression is evidenced by: (i) initial divergence of the AtPYL11/12/13 clade¹⁰ (node marked with a purple asterisk), which has only been identified in eudicots; (ii) secondary divergence

of AtPYL13 (gold-highlighted branch) from the AtPYL11/12 subclade; (iii) subsequent acquisition of lineage-specific variations within the AtPYL13 subclade, including one in *Camelina sativa* and two in the *Arabidopsis* lineage (*A. thaliana*, *A. lyrata*, and *A. halleri*; variation sites labelled in red). These changes support the hypothesis that ABA-independent PYL-like proteins reemerged in the Brassicaceae family.”

Query #17:

L213- Which is not evident from Fig. 3c, as the authors described. Please explain why AtPYL13 failed to activate *RD29B* promoter. Is PP2CA seed-specific factor?

Response: This is an insightful question. The conflicting results between the *in vitro* kinase assay (Fig. 3d,e) and *in vivo* protoplast assay (Fig. 3a,c) likely due to interference by endogenous PP2Cs. Although PYL13 represses PP2CA and ABI1 to different extents (Fig. 3d,e), it may fail to inhibit certain clade A PP2Cs (e.g., HAB1/2 and HAI1) that are expressed in rosette leaves—the tissue source for protoplasts (*Arabidopsis* eFP Browser Developmental Map; **Response Fig. 4**).

Response Fig. 4. Expression profiles of clade A PP2Cs in *Arabidopsis* tissues.

Query #18:

L217- No PYR1 data in supplementary Fig.3b.

Response: We thank the reviewer for noting this oversight. We have corrected the statement as follows: “In contrast, AtPYL13-Q38K/F71L repressed PP2C phosphatase activity and released PP2C-mediated inhibition of SnRK2 activity in an ABA-enhanced manner (Fig. 3d, e, and Supplementary Fig. 3b, c).”

Query #19:

L234-236 Please explain what does this result indicate or suggest.

Response: We thank the reviewer for this insight. The observed effect may be due to interference by residual ABA dependence or endogenous PP2Cs or PYL6, whereas transient co-expression of ABI1 elevates ABI1 levels in protoplasts. We have revised the statement as follows: “Expression of *AtPYL8-K61Q/L87F* enabled weak ABA-enhanced activation of *RD29B-LUC* expression, likely due to residual ABA dependence. While co-transfection of wild-type *AtPYL8* with *ABI1* enabled ABA-induced *RD29B-LUC* expression, co-transfection of *AtPYL8-K61Q/L87F* with *ABI1* failed to induce expression even with ABA treatment. This suggests *AtPYL8-K61Q/L87F* lacks strong ABI1-inhibitory capacity.”

Query #20:

L249-250 Please clearly explain which are the putative ABA-irresponsive PYLs in Supplementary Data 1.

Response: We thank the reviewer for highlighting this need for clarification. We have revised Supplementary Data 1 to clearly identify putative ABA-independent PYL-like proteins by highlighting them and annotating them with nomenclature consistent with Figures 2a, 4b and Supplementary Fig. 3a.

Query #21:

L280- Why the authors used OsPYL13 for structure prediction? There is no functional data for OsPYL13 (Fig. 4c).

Response: We thank the reviewer for highlighting this inconsistency. OsPYL13 was initially selected due to its seed-specific expression; however, we recognized functional validation was limited. Therefore, we have: (1) Removed OsPYL13 structural predictions; (2) Focused analysis exclusively on OsPYL12, which has experimental validation (protoplast transient assays in Fig. 4c and ITC in Supplementary Fig. 4c); (3) Updated the text to: “The F76 variation in OsPYL12 (lavender) causes steric clash (atom distance $< 2 \text{ \AA}$) with ABA, explaining its ABA independence (Fig. 4h). ITC analysis confirmed OsPYL12 cannot bind ABA (Supplementary Fig. 4c).”

Query #22:

L309-310 Please explain why are these mutations required in this experiment briefly.

Response: We sought to validate ABI3's binding to the RY cis-element using a yeast one-hybrid (Y1H) assay. Initial attempts with full-length ABI3 failed due to yeast toxicity, and testing of a truncated ABI3-B3 domain failed to detect interaction.

Literature review revealed that specific ABI3-B3 domain mutations (R64K, R66K, P69S) enhance DNA binding affinity without compromising sequence specificity¹⁶. These mutations (ABI3-B3-KKS) enabled specific interaction with the PYL13 promoter's RY motif—but not with mutated controls—confirming motif specificity.

Reviewer #4 (Remarks to the Author):

QUERY #1:

Abscisic Acid (ABA) is arguably one of the most important biomolecules on the planet, as a consequence of its central importance in promoting plant dehydration tolerance: a state essential to the initial plant colonization of land ca. 500Ma that had planetary consequences: the photosynthetic activity of earliest land plants resulted in elevation of the oxygen content of the atmosphere to a level enabling the evolution of large land animals.

ABA, a small molecule derived from the breakdown of carotenoids and ubiquitous in Nature, acts as a ligand to the multimember family of ABA receptors (“PYLs”). These receptors are START-domain proteins that are structurally modified by ABA-binding so that they form molecular complexes with a sub-family of PP2C protein phosphatases blocking their dephosphorylation activity. This activates SnRK2 protein kinases that direct transcription factors to drive the expression of genes encoding a plethora of protective gene products that enables plants to survive dehydration.

This manuscript describes the functional evolution of the PYL receptor family that is thought to have been acquired by horizontal gene transfer from microbe(s) to an ancestral alga from which all land plants subsequently diverged and proliferated.

The paper comprises:

1. Examination of receptor-like functions of bacterial PYL homologs:

(i) Does a bacterial PYL-like gene complement the *Arabidopsis* 12x *pyl*- line?

- No: there is no activation of the *Rd29b* reporter gene either with or without added ABA

(ii) Does a bacterial PYL-like gene bind to an *Arabidopsis* PP2C?

- Yes, but only one of the bacterial proteins tested showed binding in Yeast-2-Hybrid and split luciferase assays. (*Paraburkholderia rhynchosiae* “Pr”)

(iii) Does the PrPYL activate SnRK2 phosphorylation in vitro?

- No. The bacteria tested were those identified in Reference 11 (which focused on algal progenitors in land plant evolution, but indicated these species to be taxonomically related potential HGT progenitors of plant *PYL* genes). Of these *Paraburkholderia* remains the leading contender.

(iv) Can structural modeling indicate whether PrPYL could interact with a plant PP2C protein?

- Up to a point: there are similarities between the modeled PrPYL structure and that of the crystallographically determined Arabidopsis PYL-ABA-PP2C co-complex, but the predictions indicate structural inconsistencies that would preclude ABA-interaction (not least, the lack of a “gate” motif in the bacterial sequence).

I think that these essentially negative findings are not unexpected, and one would not expect PYL homolog within an extant bacterium that has been independently evolving for the last 500 million years to necessarily be similar to whichever microbial sequence actually might have been passed into the genome of the algal ancestor of today’s land plants. But it is important that this has now been experimentally tested. While comparative genomics has been a powerful tool in developing our understanding of land plant evolution, there will always be some uncertainty in making accurate predictions.

Response: We sincerely appreciate the reviewer’s supportive and insightful comments.

QUERY #2:

2. Do algal PYL homologs within the Zygnematales have properties similar to angiosperm PYLs in activating ABA-mediated gene expression?

- Yes, but not all the properties of PYLs in modern land plants. PYL function has now been analysed extensively in angiosperms, and also to a lesser extent in early-diverging land plants (Bryophytes: Ref 12 and in a very recently published paper (Zimran et al. 2025. Current Biology 24: 818-830).

The results reported here showing ABA-independent “basal” activity in PYL-PP2C interactions are consistent with these recent papers. Whereas these two papers use direct analysis of PP2C inhibition in vitro, this manuscript nicely reinforces the evidence by showing PYL-binding-specific activation of SnRK2 kinase activity being promoted in an ABA-independent manner.

I suggest that the title of this section be altered to “Zygnematales PYLs exhibit PP2C inhibitory activity in an ABA-independent manner” as the current wording implies that the experiments had all been carried out in algae, rather than analysing the activity of PYLs from algae.

Response: We sincerely appreciate the reviewer’s supportive comments and constructive feedback, which have significantly improved our manuscript. In response, we have revised the section title and incorporated recent findings regarding PYL analyses in early-diverging land plants.

QUERY #3:

3. Characterizing ABA-independent PYL activity in Arabidopsis

- ABA-independent basal activity has been associated with PYL family members that are monomeric, with the dimeric angiosperm subfamily 3 PYLs being highly ABA-dependent. The authors have carried out complementation of the 12x *pyl*- mutant to identify Arabidopsis PYLs with basal activity and those without, confirming this.

They also state in Lines 192-194: “Phylogenetic analysis of PYLs from the Brassicaceae family exhibited the stepwise transformation from ABA receptors to ABA-irresponsible PYLs after expansion of the gene family”.

I do not follow this argument. The support for this is supposedly in Supplementary Figure 3a and Supplementary Data 1 but I cannot follow the logic (This may be my fault). I should have expected basal activity alone (ABA-independence) to be the ancestral state with ABA-dependence with some residual ABA-independence to follow and complete ABA dependence to be the latest acquisition. (Essentially this is what is described in the abstract in lines 17-19). I think the authors need to clarify this.

Response: This is a very interesting question. While the fundamental ABA-independent basal activity represents the ancestral PYL state, our phylogenetic analysis reveals a distinct stepwise transition within the Brassicaceae lineage (Supplementary Fig. 3a; steps i-iii). We have revised the main text as follows: “This progression is evidenced by: (i) initial divergence of the AtPYL11/12/13 clade¹⁰ (node marked with a purple asterisk), which has only been identified in eudicots; (ii) secondary divergence of AtPYL13 (gold-highlighted branch) from the AtPYL11/12 subclade; (iii) subsequent acquisition of lineage-specific variations within the AtPYL13 subclade, including one in *Camelina sativa* and two in the *Arabidopsis* lineage (*A. thaliana*, *A. lyrata*, and *A. halleri*; variation sites labelled in red).” The stepwise progression represents reversion from ABA-dependent to ABA-independent states, suggested that ABA-independence in AtPYL11/12/13 clade is a derived trait re-evolved through specific adaptations for seed desiccation tolerance ("drying without dying").

We have clarified this transition in the main text and modified Supplementary Fig. 3a and Supplementary Data 1 to highlight key evolutionary stages. Thus, the transformation describes recent reacquisition of ABA-independence within a derived eudicot lineage, consistent with the abstract's description of functional innovation after gene family expansion.

QUERY #4:

4. Do specific mutations identify residues required for ABA-dependence?

- The residues Q38 and F71 in PYL13 are identified as variant and required for ABA binding, and the PYL13 gene is characterized by basal activity rather than ABA-dependence using site-specific mutations to correct this, and to examine the requirement for Lysine and Leucine at the corresponding positions in other PYL genes.

This then leads on to analysis of PYLs in other plants that carry similar mutations, and that also exhibit principally basal activity in their regulation of SnRK2 phosphorylation.

Response: We sincerely appreciate the reviewer's supportive and insightful comments.

QUERY #5:

5. Seed specificity of *PYLs*

- A major outcome of PYL activation of the ABA response is in the regulation of seed development. The control of ABA-dependent acquisition of desiccation tolerance is one of the most important functions for plant reproduction, enabling the development of resilient reproductive propagules that can be dispersed in space, and as a result of ABA-induced dormancy, dispersed also in time. This property is also of fundamental importance for human social evolution, as the ability to store dry seeds from one year to the next is the basis of agriculture and transformed human societies from hunter-gathering tribes to settled communities leading to the advent of civilisation.

- The authors note that a number of PYLs are specifically expressed during seed development and are thus direct activators of the genetic programme that leads to desiccation tolerance in seeds. They additionally note that AtPYL13 is one of these seed specific PYLs, and that it (and likely the others) is expressed exclusively during this time, the transcription of the gene being directed by the ABI3 transcription factor – itself generally described as being seed-specific in its functions (actually, this is not quite correct: it is also active in bud dormancy).

- However, it is not clear to what extent the AtPYL13 is essential for this process, since seed desiccation tolerance is certainly principally ABA-dependent, and thus necessarily principally reliant on those other seed-expressed PYLs named in this section of the manuscript. I do not imagine that the basal activity of AtPYL13 contributes significantly, by comparison.

- This last section, while interesting, seems somewhat out of place by comparison with the foregoing information, and it might be better published separately in another journal.

Response: We appreciate this insightful comment regarding seed desiccation tolerance mechanisms. While ABA-dependent PYLs are central to this process, our data reveal a specialized role for ABA-independent PYLs in optimizing the “drying without dying” strategy.

As established in our phylogenetic analysis (Response to Query #3), ABA-independent activity in AtPYL13 represents a derived trait re-evolved within Brassicaceae – distinct from ancestral basal activity – suggesting strong selective pressure for seed adaptation.

AtPYL13 contributes alongside seed-expressed ABA receptors (PYL11, PYL12, PYR1, PYL2, PYL5, PYL7 and PYL9, see Supplementary Fig. 5c) during late maturation. While redundancy explains the lack of phenotype in *pyl13* mutants, genetic evidence reveals its essential role: 1) *pyl* duodecuple mutants (lacking 12 *PYLs*, excluding *AtPYL13*) remain viable; 2) Homozygous seeds of *pyl* quattuordecuple mutants (lacking 14 *PYLs* including *AtPYL13*) are unattainable; 3) Rare heterozygous seeds occur only when plants are grown with wild-type pollen donors¹⁴. This indicates AtPYL13 is critical for male fertility and/or embryo development when functionally combined with loss of key ABA receptors.

The “ABI3-AtPYL13” section is indispensable for the following reasons: (i) It answers the pivotal question: Why is ABA-independent *PYL*-like genes such as *AtPYL13* seed-specific? (ii) It identifies ABI3 as the direct transcriptional regulator, integrating this ABA-independent branch into the core desiccation tolerance network. Removing it would sever the critical triad: (i) The evolution of ABA-independent *PYLs* (Query #3), (ii) Their specific expression in seeds, and, (iii) The regulatory mechanism (ABI3) embedding them within the established desiccation tolerance program. This completes our narrative on how re-evolved ABA-independence optimizes seed fitness.

Additionally, we updated our description of ABI3 as a seed- and bud-specific transcription factor in the main text as suggested.

QUERY #6:

Recommendations

This is clearly a well-executed piece of work, but the authors should consider my comments above. I also have a few specific recommendations:

Response: We sincerely appreciate the reviewer’s recognition of our work and these constructive recommendations. We have implemented all suggested improvements throughout the manuscript: 1) Clarified evolutionary trajectories in response to Query #3 (Supplementary Fig. 3a); 2) Strengthened justification for the ABI3-PYL13 section (Fig. 5g); 3) Enhanced evaluation of ABA sensitivity and binding assays (Figs. 1g-h, 2g-j; Supplementary Figs. 1f-g, 2b-c, 3b-i). All modifications directly address your insightful comments and significantly improve the manuscript’s clarity. We are grateful for your expertise in strengthening this work.

QUERY #7:

1. The title et seq.

First, I do not think that the term “ABA-irresponsive” is appropriate as a description of the PYL receptors. First, no PYL is either “responsive” or “not responsive” to ABA. PYLs are receptors that become active through interacting with ABA as a ligand in order to enable ABA responsive gene expression.

It is the phosphatase activity and the subsequent pathway of gene activation that is responsive – not the PYL. In the case of PYL homologs that bind to PP2CA phosphatases without binding ABA, - defined as “basal activity” of these PYLs, it remains to be determined how much of a response is elicited.

For example:

Lines 82-83: “we evaluated the ABA responsiveness of PYLs in the Arabidopsis ABA-insensitive *pyl* duodecuple mutant”

- Actually, this isn’t testing the “ABA responsiveness” of the PYL, it’s measuring the ABA responsiveness of the *Rd29b-Luc* reporter when the 12x mutant is complemented by wild-type *PYLs* from *Arabidopsis*, *Marchantia* and *Physcomitrella*, and so-called bacterial PYLs (or more accurately, genes identified as having some similarity to bona fide PYLs).

I recommend the use of ABA-independent, rather than-ABA irresponsive, in the title:

“ABA-independent PYL-like proteins...”

...and throughout the manuscript.

Response: We thank the reviewer for this important clarification. We have implemented the following

changes throughout the manuscript.

First, we have revised the title to: “ABA-independent PP2C-binding in PYLs traces to bacterial origins and persists in land plants”.

Second, we have clarified PYL terminology in the revised main text as follows: “The ABA-binding properties of PYLs vary significantly in seed plants; likewise, their PP2C inhibition mechanisms show distinct ABA-dependence patterns: (i) strictly ABA-dependent (no basal inhibition activity without ABA, e.g., the dimeric subfamily III PYLs in angiosperms)⁴; (ii) ABA-enhanced (basal inhibition activity potentiated by ABA)⁵; or (iii) ABA-independent (not modulated by ABA). The “ABA-independent PYL-like proteins” include: 1) constitutively active PYLs that activate stress responses regardless of ABA⁶; and 2) inactive PYLs incapable of activating stress responses.”

QUERY #8:

2. Minor corrections

Lines 46-7 and 57-8: Use a standard font, not bold text. The use of bold text to emphasize a point is acceptable in a grant proposal, but not in a paper (unless it is a section heading).

Response: We thank the reviewer for noting this formatting convention. We have removed all non-heading bold text in the revised manuscript.

QUERY #9:

Line 49: “encoded by the algae Zygnema...” to: “encoded by the alga Zygnema...”

Response: Thanks. We have corrected it.

QUERY #10:

Lines 192-194: “Phylogenetic analysis of PYLs from the Brassicaceae family exhibited the stepwise transformation from ABA receptors to ABA-irresponsive PYLs after expansion of the gene family”

This implies that an ABA receptor with low basal activity arose first, and subsequently PYLs evolved with high basal activity. I’d have thought that basal activity was the ancestral state. Of course, this is being described in a relatively late-evolving plant group, in which there could have been prior loss of

members with basal activity prior to the re-emergence of PYLs with high basal activity.

Response: We thank the reviewer for this critical insight. This is a very interesting question. An alternative explanation involves the retention of ancestral ABA-independent PYL-like proteins in seed plants. However, we observe this PYL type only in specific lineages, where it diverges from ancestral PYLs. For example, the two natural amino acid variations in AtPYL13 confer ABA insensitivity, as demonstrated by the ABA-responsive phenotype of AtPYL13-Q38K/F71L (Figure 3c-g). Conversely, introducing equivalent mutations into MeenPYL3 (R63K/I93L) did not alter its ABA responsiveness (Supplementary Fig. 2d).

We have also clarified the stepwise transition in the main text and modified Supplementary Fig. 3a to visually highlight key evolutionary stages: “Phylogenetic analysis of Brassicaceae PYLs revealed a stepwise transition from ABA-enhanced receptors to ABA-independent paralogs after gene family expansion (Supplementary Fig. 3a; steps i-iii). This progression is evidenced by: (i) initial divergence of the AtPYL11/12/13 clade¹⁰ (node marked with a purple asterisk), which has only been identified in eudicots; (ii) secondary divergence of AtPYL13 (gold-highlighted branch) from the AtPYL11/12 subclade; (iii) subsequent acquisition of lineage-specific variations within the AtPYL13 subclade, including one in *Camelina sativa* and two in the *Arabidopsis* lineage (*A. thaliana*, *A. lyrata*, and *A. halleri*; variation sites labelled in red). These changes support the hypothesis that ABA-independent PYL-like proteins reemerged in the Brassicaceae family.”

Thus, the transformation describes recent reacquisition of ABA independence in a derived eudicot lineage, aligning with the abstract’s description of functional innovation after gene family expansion.

QUERY #11:

Supplementary dataset 1: Phylogenetic analysis of PYLs.

- The authors have fallen into the trap of failing to determine whether genes annotated as ABA receptors are indeed bona fide receptors. Thus the set of PYL homologs identified for *P. patens* identifies three gene models that were identified as encoding ABA receptors in a report by Sussmilch et al. (2017) Plant Signaling & Behavior 12: e1365210, but that are actually non-functional pseudogenes. These are Pp3c3_660, Pp3c7_4410 and Pp3c15_19950. The first two of these are predicted to contain a short intron that is unsupported by RNAseq data. In each case the predicted intron (automatically assigned by the software used to generate the models) contains an in-frame termination codon that would render an unspliced transcript non-functional. In 3_660, the available

RNAseq data for this locus do not indicate the presence of an intron. For 7_4410 the gene model is unsupported by any RNAseq reads, it lies in an intergenic region and there is no evidence of a start codon. For 15_19950, the predicted polypeptide sequence shows the N-terminus to lie only 14 residues before the SGIPAT “gate” sequence: however, a BLASTX analysis of the predicted 5’UTR sequence immediately prior to the candidate N-terminal methionine residue shows a more likely N-terminal coding region with significant similarity to that of PpPYL1 (Pp3c26_15240). Readthrough shows that these two sequences are out of frame, with two consecutive termination codons between the “UTR” and “coding” sequences. These are likely degenerated pseudogenes and should be excluded.

Similarly, the *M. polymorpha* model Mapoly0145s0004 is speculative, as there is very little RNAseq support for this locus.

For *Sphagnum fallax*, The sequence assembly used (based on scaffolds, rather than chromosomes) is not the most recent: the V1.1 locus identifiers are as follows:

V0.5 Sphfalx0046s0087 = V1.1 Sphfalx16G013800

V0.5 Sphfalx0031s0007 = V1.1 Sphfalx16G062700

V0.5 Sphfalx0002s0105 = V1.1 Sphfalx19G066800

V0.5 Sphfalx0024s0170 = V1.1 Sphfalx11G055600

V0.5 Sphfalx0310s0087 = V1.1 Sphfalx04G53400

Response: We thank the reviewer for highlighting these critical annotation issues. Supplementary Data 1 comprehensively surveys putative PYL-related sequences - including functional receptors, PYL-like proteins, and pseudo-proteins - across representative streptophyte algae and land plants. This inclusive approach allows systematic tracking of gene family evolution, even where functionality is uncertain.

We have implemented the following revisions based on your feedback: 1) *P. patens* pseudogenes: Pp3c3_660, Pp3c7_4410, and Pp3c15_19950 are now explicitly annotated as “non-functional pseudogenes” (supported by RNAseq evidence and termination codon analysis); 2) *M. polymorpha* annotation: Mapoly0145s0004 is flagged as “speculative (limited RNAseq support)”; 3) *S. fallax* assemblies: All locus identifiers updated to V1.1: V1.1 Sphfalx16G013800, V1.1 Sphfalx16G062700, V1.1 Sphfalx11G055600, V1.1 Sphfalx04G53400.

References:

1. Chen, K. *et al.* Abscisic acid dynamics, signaling, and functions in plants. *J Integr Plant Biol* **62**, 25-54 (2020).
2. Yuan, X.-P. & Zhao, Y. SnRK2 kinases sense molecular crowding and form condensates to disrupt ABI1 inhibition. *Sci Adv* **11**, eadr8250 (2025).
3. Gupta, A., Rico-Medina, A. & Cano-Delgado, A.I. The physiology of plant responses to drought. *Science* **368**, 266-269 (2020).
4. Dupeux, F. *et al.* A thermodynamic switch modulates abscisic acid receptor sensitivity. *EMBO J* **30**, 4171-84 (2011).
5. Hao, Q. *et al.* The molecular basis of ABA-independent inhibition of PP2Cs by a subclass of PYL proteins. *Mol Cell* **42**, 662-72 (2011).
6. Sun, Y. *et al.* A ligand-independent origin of abscisic acid perception. *Proc Natl Acad Sci USA* **116**, 24892-24899 (2019).
7. Zimran, G. *et al.* Abscisic acid receptors functionally converge across 500 million years of land plant evolution. *Curr Biol* **35**, 818-830.e4 (2025).
8. Cheng, S. *et al.* Genomes of subaerial Zygnematophyceae provide insights into land plant evolution. *Cell* **179**, 1057-1067.e14 (2019).
9. Soon, F.F. *et al.* Molecular mimicry regulates ABA signaling by SnRK2 kinases and PP2C phosphatases. *Science* **335**, 85-8 (2012).
10. Yang, J.F., Chen, M.X., Zhang, J.H., Hao, G.F. & Yang, G.F. Genome-wide phylogenetic and structural analysis reveals the molecular evolution of the ABA receptor gene family. *J Exp Bot* **71**, 1322-1336 (2020).
11. Sun, Y. *et al.* A ligand-independent origin of abscisic acid perception. *Proc Natl Acad Sci U S A* **116**, 24892-24899 (2019).
12. Melcher, K. *et al.* A gate-latch-lock mechanism for hormone signalling by abscisic acid receptors. *Nature* **462**, 602-8 (2009).
13. Yin, P. *et al.* Structural insights into the mechanism of abscisic acid signaling by PYL proteins. *Nature Structural and Molecular Biology* **16**, 1230-6 (2009).
14. Zhao, Y. *et al.* *Arabidopsis* duodecuple mutant of PYL ABA receptors reveals PYL repression of ABA-independent SnRK2 activity. *Cell Rep* **23**, 3340-3351 (2018).
15. Mönke, G. *et al.* Toward the identification and regulation of the *Arabidopsis thaliana* ABI3 regulon. *Nucleic Acids Res* **40**, 8240-8254 (2012).
16. Jia, H., Suzuki, M. & McCarty, D.R. Structural variation affecting DNA backbone interactions underlies adaptation of B3 DNA binding domains to constraints imposed by protein architecture. *Nucleic Acids Research* **49**, 4989-5002 (2021).

Reviewer #2 (Remarks to the Author):

The authors have revised their manuscript appropriately, and the revised version is much clearer in terms of wording and the description of the evolution of ABA receptors.

Query #1:

One point I would like the authors to further improve concerns the function of *AtPYL13*. Based on the reduced seed production of the quattuordecuple mutant (generated by introducing the *pyl13* and *pyl6* mutations into the duodecuple *pyl* mutant), the authors argue for a role of *AtPYL13* in fertility and seed maturation. However, I find this evidence insufficient to support their claim for the following reasons. First, this observation does not provide direct insight into the specific role of *AtPYL13* alone. Second, it does not clarify the function of *AtPYL13* in seed maturation. Since the central argument of this work is that *AtPYL13* plays a role in seed maturation (particularly in the desiccation stage), it is critical to discuss *AtPYL13* function in this specific developmental context.

I note that mutations in seed maturation-specific genes result in seeds with desiccation intolerance or distinct morphological defects. In contrast, phenotypes related to fertility usually leads to the absence of embryonic/zygotic structures in seeds, or as aberrant segregation ratios. If the reduced seed phenotype observed here is due to sterility (either male or female), it does not support the authors' main claim regarding a role for *AtPYL13* in seed maturation.

I encourage the authors to provide clear phenotypic evidence supporting the function of *AtPYL13* in seed maturation. Otherwise, I suggest removing this ambiguous observation and tone down their claim accordingly.

Response: We thank the reviewer for pointing out the overinterpretation in this point. We agree that fertility, seed maturation, and desiccation tolerance represent distinct physiological processes. In an independent unpublished study, we assessed transcriptional reprogramming during seed maturation and observed that the vast majority of transcriptional changes are completed prior to the desiccation phase. Thus, the *ABI3*-mediated seed-specific expression of *AtPYL11*, *AtPYL12*, and *AtPYL13* suggests their potential involvement in seed maturation and desiccation tolerance.

Both ABA-dependent and ABA-independent PYLs employ SnRK2s and ABFs to mediate transcriptional reprogramming and act redundantly in maintaining seed viability¹⁻³. Plants accumulate high levels of ABA during seed maturation, and ABA concentrations gradually decline during vernalization⁴. Therefore, elevated basal stress signaling mediated by ABA-enhanced and ABA-independent PYLs is likely necessary for seed viability at stages characterized by low ABA levels, such as following vernalization.

As we were unable to obtain a *pyl* tredecuple mutant harboring the *PYL13* mutation, we have removed the contested observations related to the quattuordecuple mutant and revised our conclusions accordingly to reflect a more cautious interpretation. The text

now focuses on correlative evidence from ABI3-mediated seed-specific expression (Fig. 5) to propose a possible, rather than definitive role for AtPYL13 during seed maturation.

Query #2:

Additional comments regarding the response to Reviewer #1:

Reviewer 1 provided the most extensive and detailed comments throughout the manuscript. Thus, the authors focused their revision efforts primarily on addressing Reviewer 1's concerns. I believe the authors did a good job in responding to Query #1. The two in vitro experiments and one in vivo experiment they conducted are considered gold-standard experiments, and the results support their claims convincingly. Regarding Query #2, I find the authors' explanation reasonable. They clarified that the focus of this work is on the functional aspects of evolutionary divergence, distinguishing it from Cheng et al., 2019. Compared to previous studies that discussed the evolution of PYL proteins from algae to land plants, this manuscript covers a much broader evolutionary timeline. Overall, Reviewer 1's comments were constructive, and the authors responded appropriately. As I am not an evolutionary biologist, I do not have further comments beyond this.

Response: We sincerely appreciate the reviewer's supportive feedback and valuable perspective. To further strengthen the evolutionary analysis in our study, we have engaged in extensive discussion with Professor Guanzhu Han, an expert in plant evolutionary biology, which has helped us to better clarify the evolutionary trajectory of the *PYL* gene family in the previous revised version.

Reviewer #4 (Remarks to the Author):

First, my apologies for my tardiness in returning this review: I received the request to consider the revised manuscript immediately before taking a vacation, and it is only in the last week that I've had time to examine the revision.

I'm happy to say that so far as my own comments of the original manuscript are concerned, I think that the authors have addressed all of the comments I raised satisfactorily.

Query #1:

My comments below are principally the correction of some phrases to conform with standard English usage. However I have also made some suggestions where some alterations in the text add clarity: for example an alternative opening to the Abstract (immediately below)

I found the Abstract to lack clarity: in order to rectify this I have rewritten the opening 6 lines in a way that I think make it easier for the reader to understand (and hopefully, then proceed to read the entire paper!):

"Land plants have evolved strategies to survive water deficiency. Among these adaptations, a "drying without dying" strategy, regulated by the hormone Abscisic Acid (ABA), evolved in early land plants and is maintained in the desiccated seeds of angiosperms. This is regulated by a family of ABA receptors (the PYR/PYL/RCAR (PYL) family) able to bind protein phosphatases of the PP2CA family and suppress their inhibitory action on water stress responses. ABA-independent PYLs first appear in an aquatic algal lineage, however their origins and the mechanistic basis of ABA-independent PYL variants in land plants remain uncharacterized. Here we characterize.....etc."

Response: We are grateful to the reviewer for the constructive feedback and for the time taken to refine the wording of our abstract. We have adopted the proposed opening lines and agree that they have greatly improved the clarity of our introduction.

Query #2:

The Introduction begins: Early land plants are mainly poikilohydric and adopt the "drying without dying" strategy, which relies on fundamental stress adaptation mechanisms including osmoregulation and enables plants to survive extreme conditions². Vascular land plants retain the "drying without dying" strategy in specific organs, such as seeds or pollen, but utilize the "avoiding drying" strategy for their vegetative stages, which depends on the activation of stress responses by the phytohormone ABA^{3,4}.

- It is important to realise that this is not a binary choice, and that both acquisition of dehydration tolerance and dehydration avoidance involve activation of multiple stress responses by ABA (for example promotion of expression of genes encoding LEA proteins as a prerequisite for achieving desiccation tolerance, and the implementation of stomatal closure as a means of drought avoidance).

Response: We are grateful to the reviewer for this constructive suggestion regarding the non-binary nature of these adaptive strategies. We have revised the manuscript to reflect that both dehydration tolerance and avoidance involve ABA-mediated stress responses.

Query #3:

I suggest rewriting the opening lines of the paragraph commencing at Line 41 as follows:

"ABA activation occurs through a protein phosphorylation cascade, in which Snf1-related protein kinase 2 (SnRK2) kinases have a central role. Upon ABA binding, the PYR/PYL/RCAR (PYL) ABA receptors.....etc."

Response: We thank the reviewer for the suggested improvement and have revised the text at the beginning of the paragraph on Line 41 as indicated.

Query #4:

L58-60: "In the nonvascular liverwort *Marchantia*, MpPYL1 shows ABA-enhanced activity to inhibit PP2Cs, although with high basal activity⁹, which triggers high basal stress signaling and actively represses plant growth¹⁵"

- Reference 15, whilst being a comprehensive review of the regulation of plant growth responses, does not make any reference to *Marchantia* or the basal activity of MpPYL1 (the focus is primarily on stress and growth regulation in angiosperms)

Response: We thank the reviewer for pointing this out. We have corrected the citation by replacing Reference 15 with the primary research article by Jahan et al. (2019), which specifically documents the growth repression by *MpPYL1* in *Marchantia*.

Query #5:

L61: potentially enabling broader range of ABA responses

To: potentially enabling a broader range of ABA responses

L63: were subsequently incorporated into plant lineage.

To: were subsequently incorporated into land plant lineages.

L81: The PYL genes from plants form monophyletic group

To: The PYL genes from plants form a monophyletic group

Response: We thank the reviewer for these helpful corrections. The manuscript has been revised as suggested.

Query #6:

Results section

L182: This partially explains the weak interaction of PrPYL-ABI1 and strong interactions between algal PYLs and Arabidopsis PP2Cs as well as the constitutive

activation of RD29B-LUC expression in pyl duodecuple mutant protoplasts by algal PYLs (Fig. 1, 2, and Supplementary Fig. 2a)

- I can see the interactions between algal PYLs and Arabidopsis PP2Cs, and the activation of *RD29B-LUC* in Figures 2 and Supplementary 2, but not in Figure 1.

Response: We are grateful to the reviewer for the careful reading and for identifying this citation error. We have revised the text to accurately reference the figures. The Y2H assay in Figure 1d shows the interaction of PrPYL with PP2Cs, and when contrasted with the stronger interaction of MeenPYL3 in Figure 2c, it supports our conclusion regarding the differential activities of these PYLs.

Query #7:

L198-200: Since high basal stress signaling favored the terrestrialization of land plant progenitors, this critical arginine was not conserved during evolution to maintain high basal activity of PYLs.

To: Since high basal stress signaling would have favored the terrestrialization of land plant progenitors, we propose that this critical arginine was not conserved during evolution to maintain high basal activity of PYLs.

L280: several AtPYL13-like proteins differ from the ABA-responsive PYLs by one or two amino...

To: several AtPYL13-like proteins differ from the ABA-dependent PYLs by one or two amino...

L286: We evaluated the ABA responsiveness of these AtPYL13-like proteins

To: We evaluated the ABA dependence of these AtPYL13-like proteins

Discussion

L355: algae, the ABA receptor PYLs are not present until the last common ancestor of land plants

To: algae, the ABA receptor PYLs are not present until the last common ancestor of land plants

L357-8: Our results showed that PYL-like proteins are present in bacteria, might have gained PP2C binding ability in some soil bacteria. Plants acquired PYL genes horizontally...

To: Our results showed that PYL-like proteins are present in bacteria, and might have gained PP2C binding ability in some soil bacteria. Plants likely acquired PYL genes horizontally...

L365-6: Besides, the monomeric PYLs also evolved the strict ABA dependency in mosses

To: Independently, the monomeric PYLs also evolved strict ABA dependency in mosses

L366-7: The ABA-binding and -responsive properties of PYLs confer ABA-dependent or ABA-enhanced PP2C-inhibition activity, which enables land plants to change...

To: The ABA-binding properties of PYLs confer ABA-dependent or ABA-enhanced PP2C-inhibition activity, which enabled land plants to change....

Response: We thank the reviewer for these helpful corrections. The manuscript has been revised as suggested.

References:

1. Nakashima, K. *et al.* Three Arabidopsis SnRK2 protein kinases, SRK2D/SnRK2.2, SRK2E/SnRK2.6/OST1 and SRK2I/SnRK2.3, involved in ABA signaling are essential for the control of seed development and dormancy. *Plant Cell Physiol* **50**, 1345-63 (2009).
2. Zhao, Y. *et al.* Arabidopsis duodecuple mutant of PYL ABA receptors reveals PYL repression of ABA-independent SnRK2 activity. *Cell Rep* **23**, 3340-3351 (2018).
3. Yoshida, T. *et al.* Four Arabidopsis AREB/ABF transcription factors function predominantly in gene expression downstream of SnRK2 kinases in abscisic acid signalling in response to osmotic stress. *Plant Cell Environ* (2014).
4. Chen, K. *et al.* Abscisic acid dynamics, signaling, and functions in plants. *J Integr Plant Biol* **62**, 25-54 (2020).